# Ice crystal number concentration estimates from lidar-radar satellite remote sensing. Part 1: Method and evaluation

Odran Sourdeval[1], Edward Gryspeerdt[2], Martina Krämer[3], Tom Goren[1], Julien Delanoë[4], Armin Afchine[3], Friederike Hemmer[5], and Johannes Quaas[1]

[1]Institute for Meteorology, Universität Leipzig, Leipzig, Germany
[2]Space and Atmospheric Physics Group, Imperial College London, London, UK
[3]Forschungszentrum Jülich, Institut für Energie und Klimaforschung (IEK-7), Jülich, Germany
[4]LATMOS/UVSQ/IPSL/CNRS, Guyancourt, France
[5]Laboratoire d'Optique Atmosphérique, Université Lille1, Villeneuve d'Ascq, France

**Correspondence:** Odran Sourdeval (odran.sourdeval@uni-leipzig.de)

**Abstract.** The number concentration of cloud particles is a key quantity for understanding aerosol-cloud interactions and describing clouds in climate and numerical weather prediction models. In contrast with recent advances for liquid clouds, few observational constraints exist on the ice crystal number concentration ($N_i$). This study investigates how combined lidar-radar measurements can be used to provide satellite estimates of $N_i$, using a methodology that constrains moments of a parameterized particle size distribution (PSD). The operational liDAR-raDAR (DARDAR) product serves as an existing base for this method, which focuses on ice clouds with temperatures $T_c < -30°C$.

Theoretical considerations demonstrate the capability for accurate retrievals of $N_i$, apart from a possible bias in the concentration in small crystals when $T_c \gtrsim -50°C$, due to the assumption of a monomodal PSD shape in the current method. This is verified by comparing satellite estimates to co-incident in situ measurements, which additionally demonstrates the sufficient sensitivity of lidar-radar observations to $N_i$. Following these results, satellite estimates of $N_i$ are evaluated in the context of a case study and a preliminary climatological analysis based on 10 years of global data. Despite of a lack of other large-scale references, this evaluation shows a reasonable physical consistency in $N_i$ spatial distribution patterns. Notably, increases in $N_i$ are found towards cold temperatures and, more significantly, in the presence of strong updraughts, such as those related to convective or orographic uplifts. Further evaluation and improvements of this method are necessary but these results already constitute a first encouraging step towards large-scale observational constraints for $N_i$. Part two of this series uses this new dataset to examine the controls on $N_i$.

## 1 Introduction

Clouds play a major role in the climate system and are essential components of the Earth-atmosphere radiation balance (Stephens, 2005). A precise understanding of their properties and processes therefore is necessary to properly address current uncertainties of climate change estimates (Boucher et al., 2013). In particular, the impact of ice clouds on the Earth's radiation budget is recognized as being substantial (e.g. Liou, 1986; Stephens et al., 1990) but still remains difficult to quantify

due to the large variability and complexity of their radiative, macro- and micro-physical properties (Zhang et al., 1999; Baran, 2009).

Because of their high spatial and temporal coverage, satellite observations are excellent tools to answer these questions (Lohmann et al., 2007). The A-Train satellite constellation offers a unique synergy between a wide range of active and passive instruments (Stephens et al., 2002), such that numerous methods now exist to provide spaceborne retrievals of ice cloud properties. For instance, bi-spectral approaches based passive visible (Nakajima and King, 1990) or thermal infrared (Inoue, 1985) measurements are often used to directly infer the cloud optical depth ($\tau_c$) and ice crystal effective radius ($r_{eff}$) (e.g. King et al., 1998; Sourdeval et al., 2013). Direct retrievals of the vertically integrated ice water content (IWC) - the ice water path (IWP) - can also be obtained from these channels (Guignard et al., 2012; Sourdeval et al., 2015), passive microwave sensors (Gong and Wu, 2014), or a synergy of both (Holl et al., 2014). Vertical profiles of the cloud visible extinction ($\alpha_{ext}$), IWC and ice crystal $r_{eff}$ are commonly provided using lidar and/or radar measurements (e.g. Vaughan et al., 2009; Austin et al., 2009; Delanoë and Hogan, 2010). However, only few developments have to date focused on the ice crystal number concentration ($N_i$).

A lack of $N_i$ retrievals from satellite contrasts with the importance of this quantity for understanding and describing ice cloud processes (Comstock et al., 2008). Along with the mass concentration, the number concentration is often used as a prognostic variable in two-moment bulk microphysics schemes that predict the evolution of clouds in recent climate and numerical weather prediction models (Khain et al., 2000; Seifert and Beheng, 2006). An absence of global observational constraints therefore limits the evaluation of model predictions to sparser in situ measurements (e.g. Jensen et al., 1994; Zhang et al., 2013; Farrington et al., 2016). Moreover, $N_i$ appears as a particularly useful metric to quantify aerosol-cloud interactions due its potentially close link with the aerosol concentration (Kärcher and Ström, 2003; Kay and Wood, 2008; Hendricks et al., 2011). Consequently, while numerous studies have for these reasons used satellite estimates of the cloud droplet number concentration ($N_d$) to evaluate the indirect aerosol radiative forcing (Boers et al., 2006; Quaas et al., 2006, 2008; Gryspeerdt et al., 2016), the contribution of ice clouds to this effect remains largely unknown (Heyn et al., 2017).

One reason for this absence of a global $N_i$ dataset lies in the difficulty to directly link this quantity to other commonly retrieved cloud properties. For liquid clouds, $N_d$ can for instance be inferred through relationships between satellite retrievals of $\tau_c$ and the droplet $r_{eff}$ (Han et al., 1998; Brenguier et al., 2000). These relationships rely on strong assumptions that have shortcomings (Grosvenor et al., 2018) but nonetheless provide $N_d$ values that compare well against in situ observations (Painemal and Zuidema, 2011) and can be used to establish climatologies (Bennartz and Rausch, 2017) and study aerosol-cloud interactions (Han et al., 2002; Quaas et al., 2008). Such relationships are less trivial for ice clouds due to the high complexity and variability of ice nucleation processes (Kärcher and Lohmann, 2002, 2003; Ickes et al., 2015). Recent attempts have been made, e.g. by Mitchell et al. (2016, 2018) who linked (providing additional lidar information) the absorption $\tau_c$ and ice crystal $r_{eff}$ to $N_i$ for thin single-layer ice clouds, but rigorous validation remains necessary.

An alternative approach to estimate $N_i$ has arisen with the emergence of retrieval methods aiming at directly constraining parameters of particle size distributions (PSDs) from remote sensing observations. Indeed, provided that a PSD is properly estimated, the corresponding number concentration (the zeroth moment of the PSD, or $\mathcal{M}_0$) can be extracted. Important

developments on applying these methods to satellite observations can be attributed to Austin and Stephens (2001) who, through an elaborate variational scheme, used the sensitivity of radar reflectivity ($Z_e$) and $\tau_c$ to other moments, namely $\mathcal{M}_6$ and $\mathcal{M}_2$, respectively, to constrain PSD shape parameters. This method, initially dedicated to liquid clouds, allowed retrieving profiles of droplet geometric mean radius and a vertically homogeneous $N_d$. This work was later extended to ice clouds by Benedetti

et al. (2003) and further improved by Austin et al. (2009) to perform retrievals of $N_i$ profiles using better *a priori* assumptions. These developments are now implemented in the operational CloudSat 2B-CWC-RO product, which has extensively been used to study the IWC (e.g. Wu et al., 2009; Waliser et al., 2009; Eliasson et al., 2011) but its $N_i$ product remains to be thoroughly evaluated. Notably, Protat et al. (2010) highlighted through comparisons to ground-based lidar-radar cloud products the need to improve these $N_i$ retrievals prior to quantitative use. The authors argued that radar-only methods lack sensitivity to $N_i$ profiles

due to the dominant contribution of small ice particles to the total number concentration, whereas the combined use of a lidar extinction backscatter coefficient ($\beta_{\text{ext}}$) would help to further constrain the amount of small particles. However, no operational estimation of $N_i$ from satellite lidar-radar observations has to date been attempted.

Based on these early developments, this study aims at investigating the capabilities of lidar-radar methods to estimate $N_i$ by producing and evaluating a new dataset based on the operational liDAR-raDAR (DARDAR, Delanoë and Hogan, 2010) prod-

uct. DARDAR retrieves profiles of ice cloud properties by combining measurements from the CloudSat Cloud Profiling Radar (CPR) and the Cloud-Aerosol Lidar with Orthogonal Polarization (CALIOP). Although DARDAR does not operationally provide $N_i$, and has not been tested for this purpose, its retrieval framework that aims at constraining parameters of a PSD parameterization (Delanoë et al., 2005, hereinafter D05) makes it a suitable candidate to estimate this quantity. Nevertheless, a careful evaluation remains necessary to determine if the D05 parameterization is theoretically capable of predicting $N_i$ and if

lidar and/or radar measurements can provide sufficient information to properly constrain it. Therefore, a threefold evaluation is performed here to investigate the quality of these lidar-radar $N_i$ estimates based on comparisons to in situ observations, a case study and a brief climatological analysis.

The paper is structured as follow: Section 2 presents the methodology used to estimate $N_i$ from current DARDAR products. Sec. 3 describes the data utilized in this study. The ability of DARDAR to retrieve in situ measurements of $N_i$ is investigated in

Sec. 4. Then, Sec. 5 discusses the vertical structure of $N_i$ estimates along a short orbit and Sec. 6 proposes a brief analysis of $N_i$ climatologies. Finally, Sec. 7 concludes this study. Algorithmic limitations and uncertainties are discussed in an Appendix. The second part of this series (Gryspeerdt et al., 2018b) will use this new dataset to investigate the processes controlling $N_i$.

## 2 Methodology

### 2.1 Representation of the size distribution

$N_i$ can be expressed as the integral of a given ice particle size distribution $N(D)$,

$$N_i = \mathcal{M}_0 = \int_0^\infty N(D)dD, \tag{1}$$

with $D$ the particle dimension (hereinafter the maximum diameter). Hence, $N_i$ corresponds to the moment zero of the PSD, noted $\mathcal{M}_0$. Other moments also relate to various cloud properties (e.g. to the IWC, through mass-dimension relations, or to $D_{\text{eff}} = \mathcal{M}_3/\mathcal{M}_2$) and to remote sensing measurements (e.g. $\beta_{\text{ext}}$ relates to $\mathcal{M}_2$ and $Z_e$ to $\mathcal{M}_6$), demonstrating that PSDs act as crucial links between physical parameters and observations. However, the lidar extinction $\beta_{\text{ext}}$ and the radar reflectivity $Z_e$ each provide information on a single moment of the PSD and so, assuming a pair of single-wavelength measurements, their combination is not sufficient to fully constrain every aspect of a complex PSD; simplifications are necessary.

Parameterizing PSDs is a challenging task due to the large variability of their shapes on a global scale or even within a cloud layer (e.g. Mitchell et al., 2011; Krämer et al., 2016). Nevertheless, D05 and Field et al. (2005) showed that two-moment normalization methods can be used to reasonably approximate a wide range of measured size distributions to a single shape function, noted $F$, referred to as a "universal" or "normalized" PSD. By normalization it is meant that the dimension and concentration axes are carefully scaled in order to make $F$ independent of parameters that strongly influence the shape of the original PSD.

This study will focus on the D05 parameterization, which is used in DARDAR (see Sec. 3.1) to relate lidar/radar measurements to ice cloud properties. In D05, a normalization factor noted $N_0^*$ and the ice crystal mean volume-weighted diameter $D_{\text{m}}$ (defined as $\mathcal{M}_4$ / $\mathcal{M}_3$) serve as scaling parameters to the concentration and dimension axes, respectively. The normalization process can then be summarized as

$$F(D_{\text{eq}}/D_{\text{m}}) = N(D_{\text{eq}})/N_0^*, \tag{2}$$

where $D_{\text{m}}$ and $N_0^*$ are specifically set to make $F$ independent of the IWC and $D_{\text{m}}$ of the original PSD (i.e. they become constant after normalization). The ice crystal size is represented by the equivalent-melted diameter, $D_{\text{eq}}$, which relates to $D$ through

$$D_{\text{eq}} = \left[\frac{6m(D)}{\pi\rho_w}\right]^{\frac{1}{3}}, \tag{3}$$

where $\rho_w = 1000\,\text{kg m}^{-3}$ is the density of liquid water and $m(D)$ is a given mass-dimension (m-D) relationship. DARDAR uses the empirical m-D formulas by Brown and Francis (1995) when $D > 300\,\mu\text{m}$ and by Mitchell (1996) otherwise. D05 demonstrated with in situ measurements that this approach allows the accurate prediction of $\mathcal{M}_2$ and $\mathcal{M}_6$. Inversely, $\beta_{\text{ext}}$ and $Z_e$ can be used to constrain these moments, infer the associated scaling parameters and reproduce the original PSD using Eq. (2).

D05 further concluded that a four-parameter gamma-modified distribution,

$$N(D_{\text{eq}}) = N_0 D_{\text{eq}}^{\alpha} \exp\{-k D_{\text{eq}}^{\beta}\}, \tag{4}$$

allows the parameterization to properly fit in situ measurements from mid-latitude and tropical regions. In DARDAR, $\alpha$ and $\beta$ are two fixed parameters that were chosen to best fit these measurements ($\alpha = -1$ and $\beta = 3$), whereas $N_0$ and $k$ are iteratively adjusted during the retrieval process to fit observational constraints through their relations to the scaling parameters, as shown in Sec. 2.2.

## 2.2 Extracting $N_i$ from DARDAR

Considering the gamma-modified function in Eq. (4) to describe the shape of $N(D)$ in Eq. (1), and because the total number of particles is independent of the choice of a dimensional variable, $N_i$ in DARDAR corresponds to

$$N_i = \int_0^{+\infty} N_0 D_{eq}^{\alpha} \exp\{-k D_{eq}^{\beta}\} dD_{eq}. \tag{5}$$

Because $\alpha$ and $\beta$ are fixed, $N_i$ can be computed given a knowledge of $N_0$ and $k$. These two parameters are not part of the operational products but can be deduced from their link to other retrieved properties. It is here demonstrated how retrievals of IWC and $N_0^*$ can be used to determine $D_m$, deduce $N_0$ and $k$, and subsequently estimate $N_i$. A strict consistency with the current version of DARDAR is respected to ensure that $N_i$ estimates are meaningful; possible improvements, such as proposed by Delanoë et al. (2014, hereinafter D14) for future DARDAR versions, are not included at this stage.

As mentioned in Sec. 2.1, the scaling parameters $N_0^*$ and $D_m$ are defined so that $N(D_{eq})$ becomes independent of IWC and $D_m$ after normalization. Using the definition of $N(D_{eq})$ from Eq. (2) to rewrite $D_m$, the latter condition leads to $\mathcal{M}_4^F = \mathcal{M}_3^F$, with $\mathcal{M}_n^F$ the $n^{\text{th}}$ moment of the normalized PSD $F(D_{eq}/D_m)$. Subsequently, $\mathcal{M}_3^F$ and $\mathcal{M}_4^F$ must be equal to an arbitrary constant, which was set by D05 to $\Gamma(4)/4^4$. By inserting Eq. (4) into Eq. (2), and after simplification of the definite integral, $\mathcal{M}_n^F$ becomes

$$\mathcal{M}_n^F = \frac{1}{\beta} \Gamma\left(\frac{\alpha+n+1}{\beta}\right) \frac{N_0}{N_0^*} D_m^{-(n+1)} k^{-\frac{\alpha+n+1}{\beta}}. \tag{6}$$

Based on this equation, the conditions $\mathcal{M}_4^F = \mathcal{M}_3^F$ and $\mathcal{M}_3^F = \Gamma(4)/4^4$ lead to two unique relationships between the PSD parameters $k$ and $N_0$ and the scaling variables $N_0^*$ and $D_m$:

$$k = \left[ \frac{1}{D_m} \frac{\Gamma\left(\frac{\alpha+5}{\beta}\right)}{\Gamma\left(\frac{\alpha+4}{\beta}\right)} \right]^{\beta} \tag{7}$$

and

$$N_0 = N_0^* D_m^{-\alpha} \frac{\Gamma(4)}{4^4} \beta \frac{\Gamma\left(\frac{\alpha+5}{\beta}\right)^{\alpha+4}}{\Gamma\left(\frac{\alpha+4}{\beta}\right)^{\alpha+5}}. \tag{8}$$

Hence, providing $N_0^*$ and $D_m$, Eq. (7) and (8) can be inserted into Eq. (5) to compute $N_i$.

   The scaling parameter $N_0^*$ is provided in DARDAR, whereas $D_m$ can be deduced from other cloud properties. For instance, considering that IWC $= \frac{\pi \rho_w}{6} \mathcal{M}_3$ for equivalent-melted spheres, and by using Eq. (2) to demonstrate that $\mathcal{M}_3 = N_0^* D_m^4 \mathcal{M}_3^F$, $D_m$ relates to IWC and $N_0^*$ following

$$D_m = 4 \left[ \frac{1}{\pi \rho_w} \frac{\text{IWC}}{N_0^*} \right]^{\frac{1}{4}}. \tag{9}$$

## 3 Data description

### 3.1 Satellite retrievals

Global DARDAR retrievals of IWC and $N_0^*$ are used to compute $N_i$, following the methodology described in Sec. 2. This section only provides a brief introduction to this algorithm; the reader is invited to refer to Delanoë and Hogan (2008, 2010) for further details.

DARDAR (currently v2.1.1) uses a variational method that merges measurements from CALIOP ($\beta_{\text{ext}}$) and CPR ($Z_e$) to constrain the scaling parameters of D05 and infer profiles of various ice cloud properties such as $\alpha_{\text{ext}}$, IWC and $r_{\text{eff}}$. DARDAR retrievals are provided with a vertical resolution of 60 m along the CloudSat footprint (about 1.7 km of horizontal resolution), i.e. globally with equator crossings around 1:30 am/pm local time.

The position and thermodynamic phase of cloud layers are determined prior to the retrieval process (Ceccaldi et al., 2013) by merging satellite observations with reanalyses from the European Centre for Medium-Range Weather Forecasts (ECMWF). To avoid possibly strong uncertainties in retrievals of the cloud phase and/or properties, only purely ice clouds that are not situated below supercooled or liquid layers are considered in this study (i.e. layers identified as supercooled or mixed-phase are ignored).

DARDAR has extensively been used for improving our understanding of clouds and precipitations (e.g. Battaglia and Delanoë, 2013; Protat et al., 2014; Feofilov et al., 2015; Massie et al., 2016) . It has also been evaluated against products from a similar lidar-radar method (Deng et al., 2010) and in situ observations. Notably, Deng et al. (2012) found good agreements between the retrievals of IWC, $r_{\text{eff}}$ and $\alpha_{\text{ext}}$ from both methods, which also compared well against co-incident in situ observations, despite a small overestimation noted for IWCs retrieved by DARDAR in lidar-only conditions.

DARDAR retrievals from 2006 to 2016 are here used to produce a 10-year $N_i$ dataset. It can be noted that DARDAR products are not continuously available throughout this period due to gaps in the CloudSat measurements. Such discontinuities should however not affect the following conclusions as precise analyses of $N_i$ patterns (e.g. trends or diurnal cycles) are not intended in this study. To avoid possible confusion with the operational product, the research-level $N_i$ dataset obtained here will be referred to as DARDAR-LIM (DARDAR - Leipzig Institute for Meteorology).

It can be noted that, as any retrieval algorithm, DARDAR depends on assumptions made on non-retrieved parameters used in its forward model to simulate lidar and radar measurements. Furthermore, additional hypotheses are needed in the absence of information from one instrument, as discussed in Sec. A1. Thanks to its use of a statistical approach, DARDAR is able to rigorously propagate assumed errors on non-retrieved forward model parameters or any other *a priori* assumptions on its retrievals. A propagation of the errors attached to IWC and $N_0^*$ on $N_i$ shows relative uncertainties from about 20 to 50% on this parameter (see Sec. A2). Expectedly, these uncertainties are the lowest when lidar and radar measurements are together available. However, these numbers do not provide a complete estimation of the accuracy of $N_i$ as DARDAR does not rigorously account for uncertainties related to assumptions on the PSD shape. A preliminary sensitivity study showed that strong deviations from the assumed $\alpha$ and $\beta$ parameters could reasonably lead to errors up to 50% on $N_i$ (see Sec. A3). The overall uncertainties on $N_i$

due to instrumental sensitivity and physical assumptions is therefore difficult to quantify based on DARDAR products alone. This study instead aims at evaluating the quality of these satellite $N_i$ estimates through comparisons to in situ measurements.

## 3.2 In situ measurements

In situ PSD measurements from mid-latitude and tropical ice clouds are required to evaluate the satellite estimates of $N_i$. This evaluation must determine if (i) the PSD parameterization used in DARDAR (i.e. D05) is capable of predicting $\mathcal{M}_0$, and (ii) there is enough sensitivity in lidar-radar measurements to properly constrain $N_i$. A few conditions are thus set for this evaluation. To answer (i), it is preferable that the measurements used in this evaluation are independent of the ones utilized by D05 to build the PSD parameterization. Answering (ii) additionally requires measurements from flights that are coincident with the CloudSat overpass. Finally, (i) and (ii) require usable measurements of the concentration in small ice crystals (i.e. $D < 100\,\mu$m), which highly contribute to $N_i$. This implies that possible phenomena of ice crystal shattering on the probe tips and inlets (Korolev et al., 2011, 2013) must be accounted for to a reasonable extent, through combined specific instrumental design and post-processing (Field et al., 2006; Korolev and Field, 2015).

### 3.2.1 Airborne instruments and campaigns

**Table 1.** Description of the in situ campaigns. The numbers correspond to PSDs averaged over 10-s periods and for ice clouds with $T_c < -30$°C.

| Campaign | Instrument(s) | $\overline{\text{TAS}}$ | #PSDs / Eq. sampling time |
|---|---|---|---|
| COALESC 2011 | NIXE-CAPS | 168 m.s$^{-1}$ | 3459 / 9.6 h |
| ML-CIRRUS 2014 | NIXE-CAPS | 207 m.s$^{-1}$ | 5954 / 16.5 h |
| ACRIDICON-CHUVA 2014 | NIXE-CAPS | 209 m.s$^{-1}$ | 4166 / 11.6 h |
| SPARTICUS 2010 | 2D-S | 174 m.s$^{-1}$ | 13121 / 36.4 h |
| ATTREX 2014 | FCDP / 2D-S | 157 m.s$^{-1}$ | 11465 / 31.8 h |

Measurements from five recent airborne campaigns are used during this evaluation process. Three campaigns are described in the 'Cirrus Guide Part I' by Krämer et al. (2016), namely COALESC 2011 (Combined Observation of the Atmospheric boundary Layer to study the Evolution of Strato-Cumulus; Osborne et al., 2014), ML-CIRRUS 2014 (Mid-Latitude CIRRUS; Voigt et al., 2016) ACRIDICON-CHUVA 2014 (Aerosol, Cloud, Precipitation, and Radiation Interactions and Dynamics of Convective Cloud Systems; Wendisch et al., 2016). Another two campaigns took place over the US and tropical Pacific: SPARTICUS 2010 (Small PARTicles In CirrUS; Mace et al., 2009) and ATTREX 2014 (Airborne Tropical TRopopause EXperiment-2014; Jensen et al., 2015). A detailed description of these field campaigns and of their involved instrumentation can be found in the above-mentioned references and is therefore not repeated here. However, a brief summary of the information relevant to this evaluation is provided below and in Table 1.

The COALESC campaign involved 16 flights performed by the BAe-146 aircraft of the Facility for Airborne Atmospheric Measurements over the South-East coast of England and Wales, during February and March 2011. Despite that the main objectives of COALESC focused on stratocumulus clouds, numerous flights also involved direct measurements of PSDs within mixed-phase and cirrus clouds. The instrumentation for cloud particle measurements notably involved the NIXE-CAPS (Novel Ice Experiment - Cloud-Aerosol Spectrometer) (Meyer, 2012; Luebke et al., 2016), which provides distributions of the number concentration of particles with sizes from 0.6 to 937 $\mu$m. This instrument consists of a combination of the CAS-DPol probe for particles smaller than 50 $\mu$m and the Cloud Imaging Probe (CIPg) for particles larger than 15 $\mu$m. The in-cloud PSDs are combined from CAS-Dpol (3.0 to 20 $\mu$m) and CIPg ($> 20\,\mu$m). It should be noted that the NIXE-CAPS inlets have been designed to limit the occurrence of shattering effects, which are further reduced through the use of post-processing by inter-arrival time algorithms. Flight details and additional information regarding the NIXE-CAPS instrument and its uncertainties are provided in Costa et al. (2017) and Meyer (2012), respectively.

ML-CIRRUS took place in March and April 2014 over Europe and the North Atlantic. This campaign aimed at investigating nucleation and life cycle processes in cirrus, as well as their impact on climate. The High Altitude and LOng range (HALO) aircraft flew a total of 16 flights, including 40 h dedicated to the remote sensing or in situ measurement of cirrus. Similar to COALESC, cloud particle measurements were performed by the NIXE-CAPS probe. The reader can refer to Luebke et al. (2016) for further details on these measurements during ML-CIRRUS.

ACRIDICON-CHUVA took place in September 2014 over the Amazonian forest with the primary goal to study the role of anthropogenic aerosols on the life cycle of deep convective clouds and precipitation. This campaign involved the HALO aircraft, which performed 13 research flights for a total of 96 h. The cloud particle measurements were performed by the NIXE-CAPS probe. The algorithms to remove shattered ice fragments were not automatically applied to avoid possible erroneous removal of small droplets in warm and mixed-phase clouds. However, applying the inter-arrival time algorithms generally only negligibly change the cirrus ice particle concentrations, since in cold cirrus the crystals in most cases does not grow to sizes that are subject to shattering. Further details on the use of NIXE-CAPS during ACRIDICON-CHUVA can be found in Costa et al. (2017).

The ATTREX-2014 mission took place between February and March 2014 over the West tropical Pacific. Six flights were performed by the NASA Global Hawk aircraft, for a total of 34 h of measurements inside cirrus within the tropical tropopause layer (TTL, i.e. from an altitude of about 14 to 19 km). Concentrations in small- to moderate-size particles were measured by two instruments: a Two-Dimension Stereo (2D-S) probe (Lawson et al., 2006) for particle sizes between 5 and 1280 $\mu$m (extended to 3205 $\mu$m using the time dimension) with a maximal bin resolution of 10 $\mu$m, and a Fast Cloud Droplet Probe (FCDP) for sizes from 1 to 50 $\mu$m. Different processing methods, noted $M_i$, are available to determine particle concentrations and sizes from the 2D-S (Lawson, 2011). PSDs used here have been processed with $M_1$ or $M_7$ when available. Erfani and Mitchell (2016) have shown no significant differences in the concentration of small particles from these two methods. The 2D-S was specifically developed to limit ice shattering through probe inlet design and is combined with a post-processing treatment based on an inter-arrival time algorithm (Lawson, 2011). However, it should be noted that concentrations measured in its first 2 bins (i.e., for particles smaller than 25 $\mu$m) may suffer from large uncertainties (Jensen et al., 2013b; Gurganus

and Lawson, 2018), which must be considered when selecting a minimal size threshold for computing $N_i$ (see Sec. 3.3). The FCDP also is considered to be efficient at removing shattered particles (McFarquhar et al., 2007). These two instruments are therefore combined here to improve the description of small particles in PSD measurements. The FCDP is used to provide the concentration of particles from 3 to 24 $\mu$m (i.e. 10 bins) and the 2D-S is used from 25 (i.e. its 3rd bin) to 3205$\mu$m. The 1 $\mu$m gap is accounted for by scaling the concentration of the last FCDP bin. More information on 2DS and FCDP measurements during ATTREX-2014 can be found in Thornberry et al. (2017).

SPARTICUS was operated as part of the Atmospheric Measurement and Radiation (ARM) aerial program (Schmid et al., 2013) to reach a better understanding of small ice particles in clouds. This mission took place between January and June 2010 over Central USA and involved a Learjet-25 aircraft that performed 200 h of scientific flights in synoptic and convective ice clouds. Its instrumentation involved the 2D-S probe for particle size measurement. A Forward Scattering Spectrometer Probe (FSSP) was also available during the campaign but its measurements are not included here due to likely contamination by shattering (Field et al., 2003; McFarquhar et al., 2007; Jackson et al., 2015). The SPARTICUS data used here was treated with a combination of the $M_1$ method for D > 365 $\mu$m and $M_4$ otherwise (Lawson, 2011), which allows for a more accurate treatment of out-of-focus particles (Korolev, 2007). An advantage of SPARTICUS for this study is that it contains numerous coincident flights with the A-Train, as detailed in Deng et al. (2012).

### 3.2.2 Data processing

In order to ensure an optimal consistency between the PSD measurements from each airborne campaign, an identical post-processing procedure has been followed to treat 1-Hz measurements from the 2DS, FCDP-2DS and NIXE-CAPS. This section discusses the most important details regarding the treatment of these measurements.

First, the 1-Hz measurements have been averaged over 10-s periods to improve the statistical reliability of cloud sampling by in situ probes. This averaging also allows for a better comparability with cloud volumes sampled by CloudSat (and therefore DARDAR products), which has an along- and across-track horizontal resolution of 1.7 and 1.4 km, respectively. Considering the average true air speeds (TAS) for each campaign (see Table 1), 10-s PSDs are representative of flight legs from about 1.6 to 2.1 km.

Furthermore, to avoid possible ambiguities and uncertainties related to satellite retrievals and in situ measurements in mixed-phase clouds, this study focuses purely ice clouds, i.e. with a temperature $T_c$ < -40°C. However, to allow for additional flexibility in the evaluation, all in situ measurements obtained when $T_c$ < -30°C are considered. Possible contamination by liquid drops are expected to be negligible at these temperatures (Costa et al., 2017).

Finally, the IWC corresponding to each in situ PSD is required to obtain predictions by D05. Bulk measurements are available for SPARTICUS and ATTREX but the bulk IWC was not measured for ACRIDICON-CHUVA, COALESC and ML-CIRRUS. Alternatively, and consistently with Krämer et al. (2016), the m-D relation by Luebke et al. (2016), noted $m_{L16}(D)$, can instead be utilized to estimate IWCs from the NIXE-CAPS PSD measurements. $m_{L16}(D)$ is based on a m-D relation by Mitchell et al. (2010), which was slightly modified to improve the representativeness of the mass concentration for small ice crystals. The validity of this type of approach, and of $m_{L16}(D)$ in particular, was recently consolidated by Erfani and Mitchell (2016) and

Afchine et al. (2018), who demonstrated their accuracy and generalizability for all types of ice clouds from $T_c < -20°C$. Afchine et al. (2018) has in particular shown that this relation should be applicable to tropical clouds and that the influence of different m-D relations on IWC is small in the temperature range of cirrus. Considering this, and because Mitchell et al. (2010) and Erfani and Mitchell (2016) developed and tested their m-D relation using 2D-S measurements from tropical and mid-latitude campaigns (including SPARTICUS), $m_{L16}(D)$ should as well be applicable to SPARTICUS and ATTREX2014. For the sake of consistency, $m_{L16}(D)$ is here utilized to estimate the IWC for all campaigns. The uncertainties arising from using a m-D relation are discussed in Erfani and Mitchell (2016) and appear reasonable in the context of this evaluation due to the relatively small sensitivity of D05 predictions to IWC, as discussed in Sec. 4.1.

Overall, about 40 000 10-s PSDs, or 106 h of equivalent cloud sampling, are used for the evaluation presented in this study. These numbers are summarized in Table 1 and the distribution of temperatures sampled during each campaign is indicated in Fig. S1 of the supplementary materials.

## 3.3 Choice of a minimum integration size

To ensure a consistency with DARDAR when inferring $N_i$ from Eq. (5), the PSD parameters $\alpha$ and $\beta$ are set to -1 and 3, respectively. However, $\alpha = -1$ implies a discontinuity in $N(D_{eq})$ when the diameter equals zero. An analytic solution for $N_i$ can therefore only be obtained by considering a minimum diameter, $D_{min}$, for the integral. This threshold must here be chosen within the validity range of the in situ measurements used for the evaluation.

As mentioned in Sec. 3.2.1, the 2DS, FCDP and NIXE-CAPS have a different sensitivity to small particles. The former instrument measures ice crystals with sizes down to about $5\,\mu m$, whereas the two latter can detect particles down to $1\,\mu m$. For consistency reasons, and to avoid possible contamination by aerosols, only ice crystals larger than $5\,\mu m$ will be considered when computing $N_i$ from each probe. The same threshold is thus applied when computing $N_i$ from DARDAR and the following results will focus on concentrations in crystals larger than $D_{min} = 5\,\mu m$, noted $N_i^{5\,\mu m}$.

In situ measurements of $N_i^{5\,\mu m}$ can still be associated with large uncertainties that are difficult to quantify. In particular, measurements from the first two size bins of the 2D-S (5 to $25\,\mu m$) are known to suffer from uncertainties due to the instrumental response time and depth-of-field (Jensen et al., 2013b; Gurganus and Lawson, 2018). Also, despite being minimized, a contamination by ice shattering events cannot be excluded. These effects are typically associated with an overestimation of $N_i^{5\,\mu m}$. Therefore, the concentration of particles larger than 25 and $100\,\mu m$ ($N_i^{25\,\mu m}$ and $N_i^{100\,\mu m}$, respectively) will also be used during this evaluation. The $D_{min} = 25\,\mu m$ threshold allows to represent ice crystals of moderate sizes for which in situ measurements can be considered of higher confidence. The $D_{min} = 100\,\mu m$ threshold typically involves concentrations for which in situ measurements are the most accurate and the D05 parameterization is expected to perform well.

It is worth mentioning that different physical processes are likely to influence $N_i$ depending on the threshold choice. For instance, small particles that are nucleated through homogeneous freezing should dominate $N_i^{5\,\mu m}$, whereas large particles resulting from aggregation processes are likely to influence $N_i^{100\,\mu m}$.

## 4  In situ evaluation

### 4.1  Optimal predictability of $N_i$ by D05

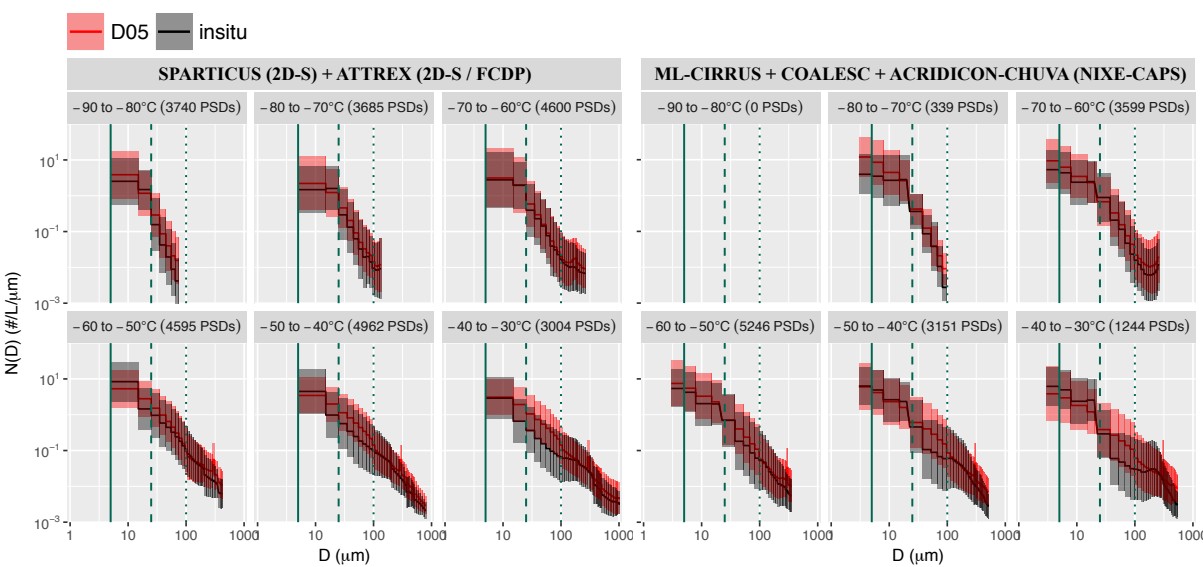

**Figure 1.** Left: Mean PSDs measured (black lines) during ATTREX and SPARTICUS, averaged per 10°C temperature bins (from -90 to -30°C). Black contours indicate one standard deviation around the mean. The mean and spread of one-to-one predictions by the D05 parameterization are similarly indicated in red. The total amount of PSDs in each $T_c$ bin is indicated in the legend and the relative contributions from each campaign can be deduced from Fig. S1. Vertical plain, dashed and dotted green lines indicate D = 5, 25 and 100 $\mu$m, respectively. The SPARTICUS data with $T_c$ < -60°C is here ignored to avoid contaminating FCDP measurements with uncertainties arising from the first first size bins of 2D-S. Right: Similar, for the ACRIDICON-CHUVA, ML-CIRRUS and COALESC campaigns.

The ability of D05 to predict $N_i$ is now investigated. It is reminded that this parameterization is designed to predict $\mathcal{M}_2$ and $\mathcal{M}_6$ and so its representation of the distribution in small particles remains to be tested. PSDs and $N_i$ predictions by D05 are here computed on the basis of IWC and $N_0^*$ values from each of the 40 000 PSDs composing the dataset described in Sec. 3.2. Comparing these predictions back to the original in situ PSDs and $N_i$ measurements should provide insights on the abilities and limitations of D05 to predict $N_i$ assuming that IWC and $N_0^*$ are perfectly constrained (i.e. if DARDAR retrievals of these parameters were optimal).

### 4.1.1  Reproducibility of the PSDs

As indicated in Sec. 2.1, the D05 parameterization predicts PSDs based on the assumption of a "universal" size distribution shape and the knowledge of two scaling parameters. Following this formalism, a PSD prediction by D05 can be obtained given the $D_m$ and $N_0^*$ values corresponding to each measured PSD. $D_m$ can directly be extracted from in situ PSDs, as it corresponds to the ratio of $\mathcal{M}_4$ to $\mathcal{M}_3$. $N_0^*$ can be indirectly estimated from $D_m$ and the IWC, using Eq. (9). It can be noted that $N_0^*$ is

proportional to $IWC \times D_m^{-4}$, which means that predictions by D05 are much more sensitive to $D_m$ than to the IWC. This point makes the use of $m_{L16}(D)$ to estimate IWC a reasonable approximation for the purposes of this evaluation. The size dimension of PSDs predicted by D05 has been converted from $D_{eq}$ to $D$ to improve the clarity of the following comparisons.

Comparisons between the PSD measurements obtained during ATTREX2014 and SPARTICUS and corresponding predictions by D05 are shown in Fig. 1 (left panels). The black and red lines respectively indicate the mean measured and predicted PSDs within 10°C temperature bins. The use of measurements from mid-latitude (SPARTICUS) and TTL ice clouds (ATTREX2014) allows a high statistical significance to be reached (over 3000 PSDs) in each $T_c$ bin from -90 to -30°C. The colored contours indicate one standard deviation around that mean. It can be noted that the measured and predicted concentrations in the FCDP bins have been averaged within each of the two first 2D-S bins in order to conveniently display the means in Fig. 1. This figure clearly shows a very good overall agreement between D05 predictions and the in situ measurements. The mean as well as the spread of the 2D-S and FCDP measurements are well represented by D05. The agreement is especially good for -90°C < $T_c$ < -50°C, where the in situ distribution tends to be mono-modal with very few large particles. A small overestimation by D05 of the concentration of crystals smaller than 25 $\mu$m is still noted when $T_c$ < -70°C. An overestimation of the number of particles with $D$ < 100 $\mu$m is also noted for D05 from $T_c$ > -50°C, where a second mode appears for large aggregated particles. Such features and temperature dependency of PSD shapes have already been widely reported in the literature (e.g. Mitchell et al., 2011; Mishra et al., 2014; Luebke et al., 2016). In the occurrence of a bi-modal shape in in situ measurements, the D05 parameterization naturally tends to reproduce the concentration of large particles due to their strong weight on $D_m$ and IWC. Because a monomodal shape is assumed to describe the PSD in D05, an erroneous extrapolation of the concentration of small particles leads to the observed overestimation when $T_c$ > -50°C. However, this overestimation appears to mainly concern particles from 25 to 100 $\mu$m, as the concentration of ice particles smaller than about 15 $\mu$m seems accurately predicted when $T_c$ > -70°C (keeping in mind that measurements for such small particles can be highly uncertain).

These results are supported by the evaluation of the D05 predictions of NIXE-CAPS measurements during ACRIDICON-CHUVA, COALESC and ML-CIRRUS, which are similarly shown on the right-hand side of Fig. 1. Despite much fewer measurements of ice clouds with $T_c$ < -70°C, very good agreements are found in the mean and the spread predicted by D05 for -50 < $T_c$ < -70°C. A small overestimation of the concentration of small ice crystals (D < 25 $\mu$m) is found for $T_c$ < -60°, where concentrations measured by the NIXE-CAPS only slightly increase. This feature should be carefully accounted for due to the lack of measurements in the coldest temperature bin, but could again indicate a too steep increase of $N_i$ towards small particles in D05 (i.e., a too negative $\alpha$) at very low temperatures. Consistent with the previous results, the D05 predictions are less accurate towards higher temperatures as bi-modal structures tend to appear in the in situ measurements above -50°C. Moreover, comparing all campaigns shows a very good overall agreement between the NIXE-CAPS and 2D-S/FCDP measurements, which points towards the generalization of these conclusions. It should be mentioned that these analyses are not repeated by explicitly discriminating between cloud types (e.g. synoptic cirrus/anvil or liquid/ice origin) for reasons of brevity. The overall agreements observed in Fig. 1 for each are considered satisfactory in this evaluation, especially since DARDAR does not discriminate between cloud types and the normalized size distribution used in D05 is expected to perform equally for all cloud types.

### 4.1.2 Consequences on $N_i$ predictions

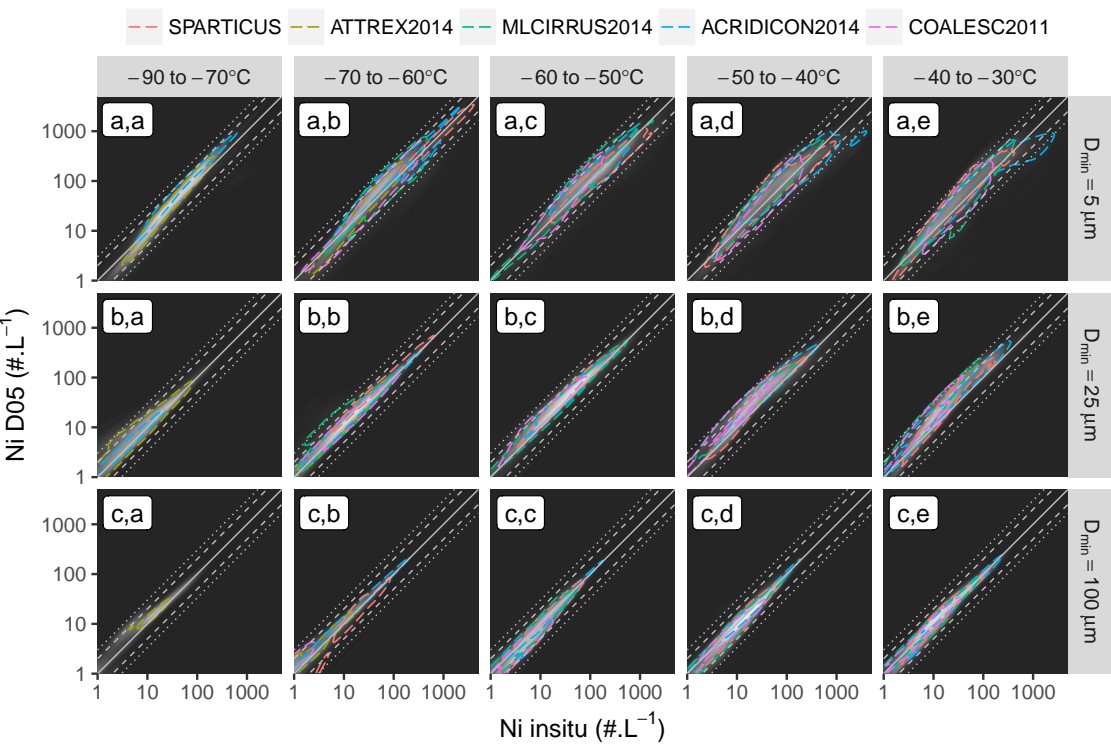

**Figure 2.** Density scatterplot showing $N_i$ theoretically estimated by D05 as function of corresponding in situ measurements (from white to black indicates high to low frequency of occurrence). Color isolines indicate the 68% (one standard deviation) confidence interval for each campaign. Density and confidence intervals are provided per 10°C temperature bins from -80 to -30°C (first to fifth column) and lower integration threshold for $N_i$ (5, 25 and 100 $\mu$m in the first, second and third row, respectively). The identity line and a factor of 2 and 3 around it are indicated by grey plain, dashed and dotted lines, respectively.

$N_i$ obtained from direct integrations of the measured and predicted PSDs are now compared. Fig. 2 shows a density scatterplot of one-to-one comparisons between the in situ measurements (x-axis) and corresponding D05 predictions (y-axis), obtained by integrating the corresponding PSDs from $D_{min}$ = 5, 25 and 100 $\mu$m (first to third row, respectively). The background color indicates the overall density and isolines are provided to show the 68% confidence levels (i.e. from which all values inside fall within one standard deviation $\sigma$ from the mean) for each campaign. These results are shown per 10°C temperature bins from -80°C to -30 °C (first to fifth column, respectively).

Fig. 2(c,a-e) show that the prediction of $N_i$ for ice particles larger than 100 $\mu$m is very consistent with the in situ reference, with an agreement close to the one-to-one line for all campaigns and temperatures. $N_i^{100\,\mu m}$ values ranging between about 1 and

$100\,\text{L}^{-1}$ are observed. This good agreement was expected from Sec. 4.1.1, and because D05 should in principle perform best at reproducing the concentration in large particles. Fig. 2(b,a-e) also indicates an accurate prediction of $N_i^{25\,\mu\text{m}}$, well within a factor of 2 (dashed lines), from -80 to -50°C. At higher temperatures, $N_i^{25\,\mu\text{m}}$ predictions by D05 can be overestimated by a factor of 2 to 3 (dotted lines) for most field campaigns. These results also hold for $N_i^{5\,\mu\text{m}}$, as indicated in Fig. 2(a,a-e), despite a larger spread within and between the campaigns in this case. It can be noted that the overestimation is particularly strong for SPARTICUS (red isolines) but is less clear for other campaigns. The overestimation is also not as clear as for $N_i^{25\,\mu\text{m}}$, as the concentration in particles smaller than $25\,\mu\text{m}$ appears more properly predicted by D05 (see Fig. 1). At $T_c < $-50°C, the D05 predictions are more consistent with the in situ measurements, with maximal $N_i^{5\,\mu\text{m}}$ values of about $300\,\text{L}^{-1}$ but that can also reach up to $1000\,\text{L}^{-1}$ for several field campaigns. A small overestimation, by a factor less than 2, can nevertheless be observed in $N_i^{5\,\mu\text{m}}$ predictions by D05 when $T_c < $-60°C. This is consistent with its steeper increase of the concentration in small ice crystals, noted in Fig. 1.

## 4.2 Satellite estimates vs. co-incident measurements

Section 4.1 demonstrated the ability of the D05 parameterization to predict $N_i$ measurements from numerous airborne campaigns. However, these conclusions only reflect ideal cases where the input parameters of D05 are perfectly constrained, since IWC and $N_0^*$ were extracted from the in situ data. It is now necessary to investigate if enough information is contained in lidar and radar measurements to sufficiently constrain these two parameters and estimate $N_i$.

This question is investigated by comparing the DARDAR-LIM $N_i$ to measurements from co-incident flights. These flights are selected under the condition that they are within a maximum distance of $5\,\text{km}$ and a 30-min time period from the CloudSat/CALIPSO overpass. Among the campaigns described in Sec. 3.2.1, co-incident flights with the A-Train track were intended during ACRIDICON-CHUVA, ML-CIRRUS and SPARTICUS. Unfortunately, none of the 3 co-incident flights during ACRIDICON-CHUVA could be selected here, due to the absence of CALIOP measurements (12 Sept. 2014) or flights that do not fulfill the above conditions (about $3\,\text{h}$ late or $350\,\text{km}$ West from the overpass track on 21 and 23 Sept. 2014, respectively). Also, technical issues occurred during the ML-CIRRUS co-incident flights (04 April 2014), making PSD measurements uncertain and not usable for this evaluation. However, numerous flights successfully achieved a close spatial and temporal co-incidence with the A-Train during SPARTICUS. A list and description of all these flights can be found in Deng et al. (2012). Overall, about 1750 PSDs were here found to match the above conditions and are considered in this evaluation. The co-incident DARDAR-LIM $N_i$ are obtained by selecting the closest pixel (based on a great-circle distance) at the altitude of the airplane.

Figure 3 compares, similarly to Fig. 1, PSDs measured by the 2D-S along A-Train overpasses (black) to corresponding predictions by D05 (red). PSDs predicted by D05 on the basis of co-incident DARDAR IWC and $N_0^*$ retrievals are additionally shown in blue. These correspond to the PSDs that are integrated to compute $N_i$ in DARDAR-LIM. All PSDs are averaged per 10°C bins (columns) and by instrumental conditions met for DARDAR retrievals (rows). In agreement with results from Sec. 4.1.1, the theoretical predictions by D05 are in better agreement with 2D-S observations when the latter display monomodal shapes. This is mainly observed towards low temperatures but also when retrievals are obtained in lidar-only condition, i.e. for thin cirrus or in regions near cloud-top. The 2D-S PSDs feature a stronger bi-modality in lidar-radar

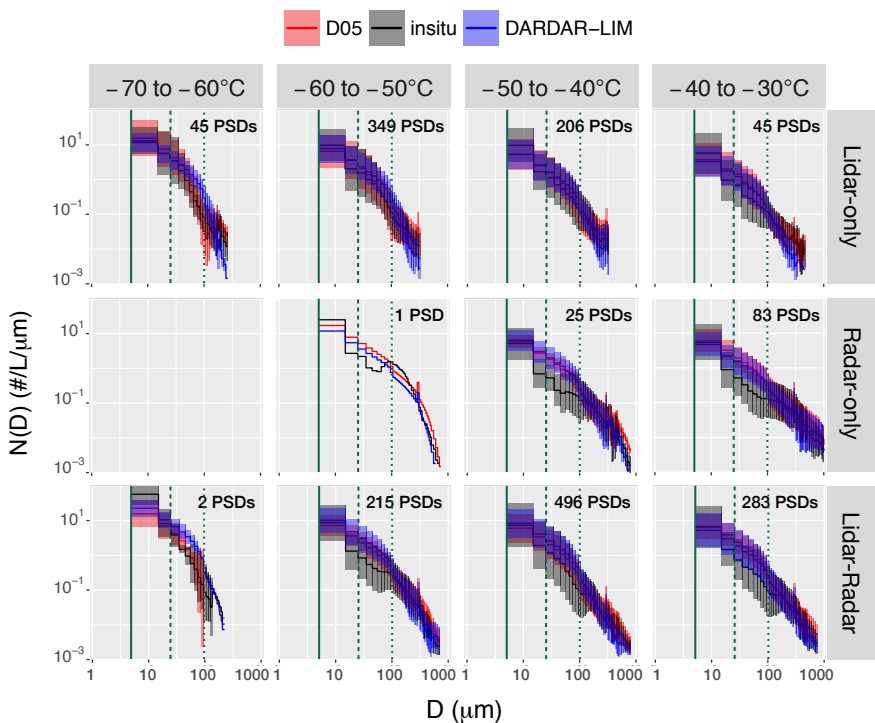

**Figure 3.** Similar to Fig. 1 but for SPARTICUS flights co-incident with the A-Train overpass. PSDs estimated on the basis of DARDAR IWC and $N_0^*$ retrievals (i.e. corresponding to the DARDAR-LIM $N_i$) is shown in blue. All PSDs are averaged per temperature bin (columns) and instrumental conditions met during DARDAR retrievals (rows).

and radar-only regions, where larger crystals resulting from aggregation or complex heterogeneous nucleation processes are likely to appear. D05 is by construction unable to reproduce this behaviour but, despite disagreements with the 2D-S, PSDs retrieved by DARDAR-LIM agree well with D05 predictions based on in situ measurements. This indicates that $N_0^*$ and IWC are sufficiently retrieved and that errors on $N_i$ are likely to be dominated by assumptions on the PSD shape. That is especially
5    true in lidar-radar conditions where $\beta_{\mathrm{ext}}$ and $Z_e$ both provide information on the concentration of small and large particles, respectively. As expected, the concentrations in particles with $D > 100\,\mu m$ are not well constrained in lidar-only conditions. Inversely, radar-only retrievals poorly constrain concentrations in particles smaller than $100\,\mu m$. However, Fig. 3 shows that, despite fewer available constraints under lidar-only conditions, reasonable $N_i^{5\,\mu m}$ and $N_i^{25\,\mu m}$ estimates are obtained due to simpler PSD shapes.
10      In order to avoid problems related with one-to-one comparisons of satellite and airplane measurements, a statistical comparison is presented in Fig. 4. This figure shows histograms of $N_i$ for the 2D-S (black) and DARDAR-LIM (blue) per temperature bin and $D_{\mathrm{min}}$ threshold. Theoretical predictions by D05 are indicated in red to provide an idea of the optimal expectations for DARDAR-LIM. Plain, dotted and dashed lines indicate satellite estimates that correspond to lidar-radar, lidar-only, radar-only conditions, respectively. These histograms are individually shown in Fig. S6 for a better clarity. Mean $N_i$ values for 2D-S,

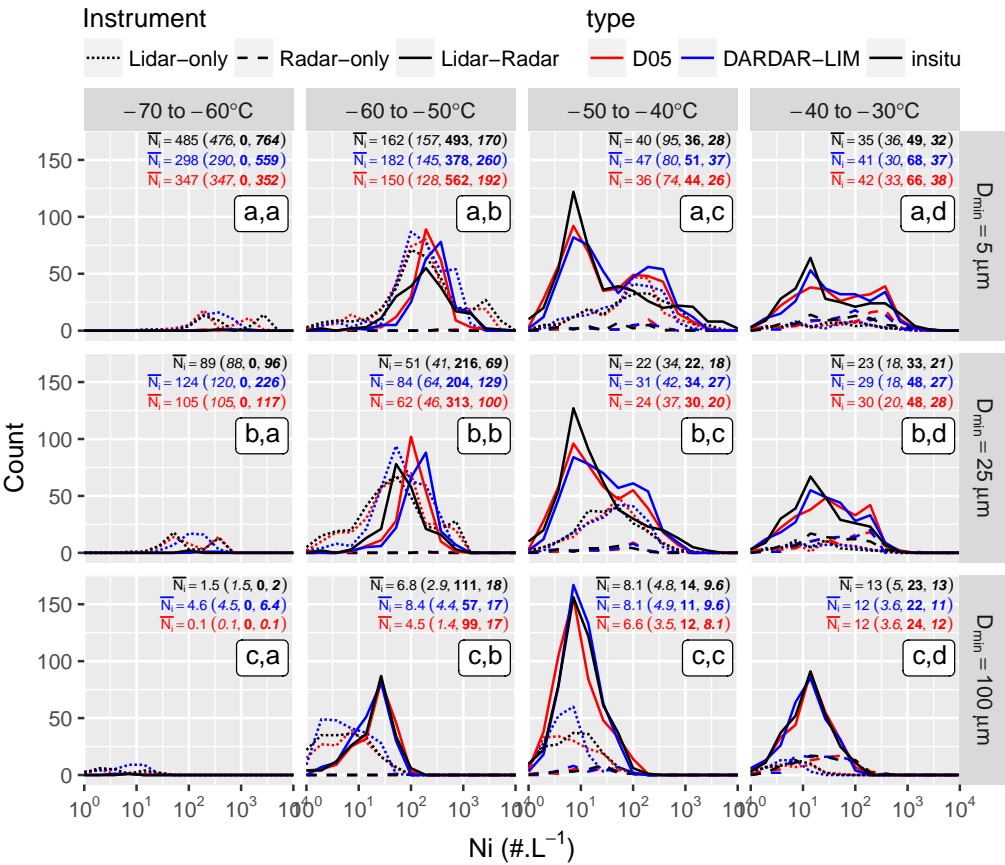

**Figure 4.** Histograms of $N_i$ measured during SPARTICUS (black), theoretically estimated by D05 (red) and retrieved by DARDAR-LIM (blue). Plain, dotted and dashed lines indicate that DARDAR retrievals were obtained using the lidar-only, radar-only and lidar-radar, respectively. Panels indicate the temperature and integration threshold, similarly to Fig. 2. Geometric means of $N_i$ (in $L^{-1}$) are shown in each panel for 2D-S, DARDAR-LIM and D05. The overall mean is first indicated, followed in brackets by values for the lidar-only (italic), radar-only (bold) and lidar-radar (bold-italic) subsets.

DARDAR-LIM and D05 are indicated in each panel. Very good agreements are seen between the satellite estimates and in situ observations of $N_i^{100\,\mu m}$ at all temperature ranges above -60°C. The distributions and mean values of $N_i^{100\,\mu m}$ estimated by DARDAR-LIM are perfectly consistent with the 2D-S for all instrumental conditions. Deviation of radar-only values in the -60 to -50°C bin (Fig. 4(c,b)) can be discarded as only representative of 1 PSD. The satellite estimates of $N_i^{25\,\mu m}$ also agree with the 2D-S, however with the expected overestimation of $N_i^{25\,\mu m}$ in D05 and DARDAR-LIM due to limited PSD shape assumptions. Overestimations by about 10 to 30% and 20 to 60% are found in the mean values of $N_i^{25\,\mu m}$ by D05 and DARDAR-LIM, respectively. This overestimation is less in lidar-only conditions, consistently with the weaker bi-modality of PSDs (see Fig. 3). Similar observations can be made for $N_i^{5\,\mu m}$, with a slightly smaller overestimation of the mean values by DARDAR-LIM due to good agreements between DARDAR-LIM and 2D-S for $D < 15\,\mu m$ noted in Fig. 3. However, it is

reminded that uncertainties in the 2 first bins of the 2D-S can contaminate its estimations of $N_i^{5\,\mu m}$ and so $N_i^{25\,\mu m}$ represents a more trustworthy estimate of $N_i$ from this instrument. Finally, it clearly appears from Fig. 4 that, even when the distributions from DARDAR-LIM and the 2D-S do not perfectly agree, the satellite estimates remain close to the D05 predictions. This again indicates that errors in $N_i$ estimates by DARDAR-LIM are dominated by assumptions made on the PSD shape rather than by a lack of instrumental sensitivity.

Figues 3 and 4 therefore demonstrate that DARDAR-LIM is capable of statistically reproducing 2D-S measurements of $N_i^{100\,\mu m}$, $N_i^{25\,\mu m}$ and $N_i^{5\,\mu m}$, with an overestimation up to about a factor of 2 that can be expected in the mean $N_i^{25\,\mu m}$ and $N_i^{5\,\mu m}$ values due to a misrepresentation of the PSD shape by D05 at warm temperatures. An analysis of one-to-one comparisons between DARDAR-LIM/D05 and 2D-S (see Fig. S4 and S5) also support these conclusions.

## 5   Case study

A first examination of $N_i$ profiles by DARDAR-LIM is performed here in the context of a case study corresponding to a frontal ice cloud structure observed on 03 February 2010 around 20:00 UTC over South Central USA. This case is of particular interest as it contains a leg of high spatial and temporal coincidence between the A-Train and the Learjet-25 aircraft involved during SPARTICUS.

### 5.1   Overall context

The cloud structure analyzed here is part of a mature cyclonic system that has reached an occluded stage, as featured by the brightness temperature snapshot shown in Fig. 5(a). Further analyses of the weather conditions (not shown here for brevity reasons) indicated that this system originated from a mid-tropospheric wave pattern that crossed the US and supported a surface low pressure area over North Central Mexico. The storm then moved northwesterly, eventually reaching Northeastern USA on 05 February as a major blizzard.

The CloudSat track crossed the cyclone from south to north around 19:55 UTC (dashed green line). The section of interest for this case study (plain green line) captures a frontal cloud associated with the ascending southern moist warm air flow atop cold continental air mass. The corresponding CloudSat $Z_e$ and CALIOP $\beta_{ext}$ profiles, shown in Fig. 6(a-b), typically hint to high water contents and precipitation toward the center of the cyclone and to thin ice clouds as the A-Train moves northwards toward its periphery.

The Learjet-25 performed in situ measurements in a cirrus at the edge of the cyclone. The flight track is shown by black lines in Fig. 5(a-b) and Fig. 6. These figures indicate that the aircraft approached from the west at an altitude of about 10.7 km (near cloud top) and descended to about 8 km (near cloud base) before reaching the overpass. The aircraft then closely followed the A-Train while ascending to cloud top and finally descended back to cloud base in a spiral. Optimal comparisons between the A-Train and Learjet-25 measurements are expected within the ascending leg from about 39.5 to 40.7°N, where the time and space coincidences are well within 15 min and 10 km, respectively.

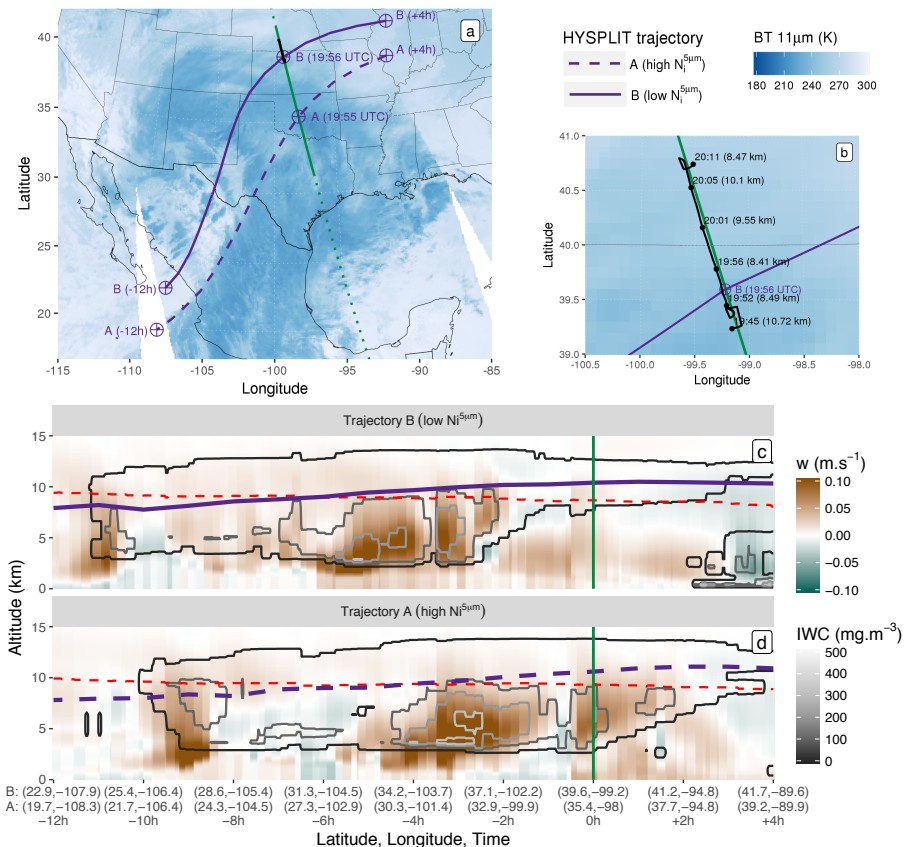

**Figure 5.** (a): Map summarizing the observations during the case study. The shaded blue background corresponds to MODIS/Aqua 11 $\mu$m brightness temperature measurements around the A-Train overpass. The CloudSat track is shown by a dotted green line and a plain green line highlights the region of interest. The Learjet-25 flight track is shown in black. Dashed and plain purple lines represent HYSPLIT trajectories computed for 2 air parcels of high (A) and low (B) $N_i^{5\,\mu m}$, respectively, at 11 km along the overpass. (b): Similar to (a) but zoomed in around the flight area. The aircraft UTC times and altitudes are indicated in black. (c-d): Vertical cross sections of $w$ (background color) and IWC (grey contours) predicted by NARR along the B and A trajectories, respectively. The positions of the corresponding air parcel are indicated in purple. A dashed red line shows the -40 °C isotherm. The overpass time is highlighted by a vertical green line.

## 5.2 Vertical structure along the overpass

Profiles of $r_{eff}$, IWC, $N_i^{5\,\mu m}$, $N_i^{25\,\mu m}$ and $N_i^{100\,\mu m}$ along the selected A-Train overpass are respectively shown in Fig. 6(d-h). As expected, high IWC values are retrieved between altitudes of 2.5 and 7.5 km along the southern half of the track (31.2-35.4°N), i.e. towards the center of the cyclone. Retrievals of $r_{eff}$ indicate small crystals (about 30 $\mu$m or less) above the 40°C isoline (dashed red line) and particles larger than 100 $\mu$m below 8 km. The clear cut below 2.5 km, associated with high $Z_e$, corresponds to pixels classified as rain by the DARDAR mask. Fig. 6(f) shows that high $N_i^{5\,\mu m}$ values are found towards cloud top, ranging from 250 to more than 1000 L$^{-1}$. $N_i^{5\,\mu m}$ increases above the -40°C isoline, consistently with the probable

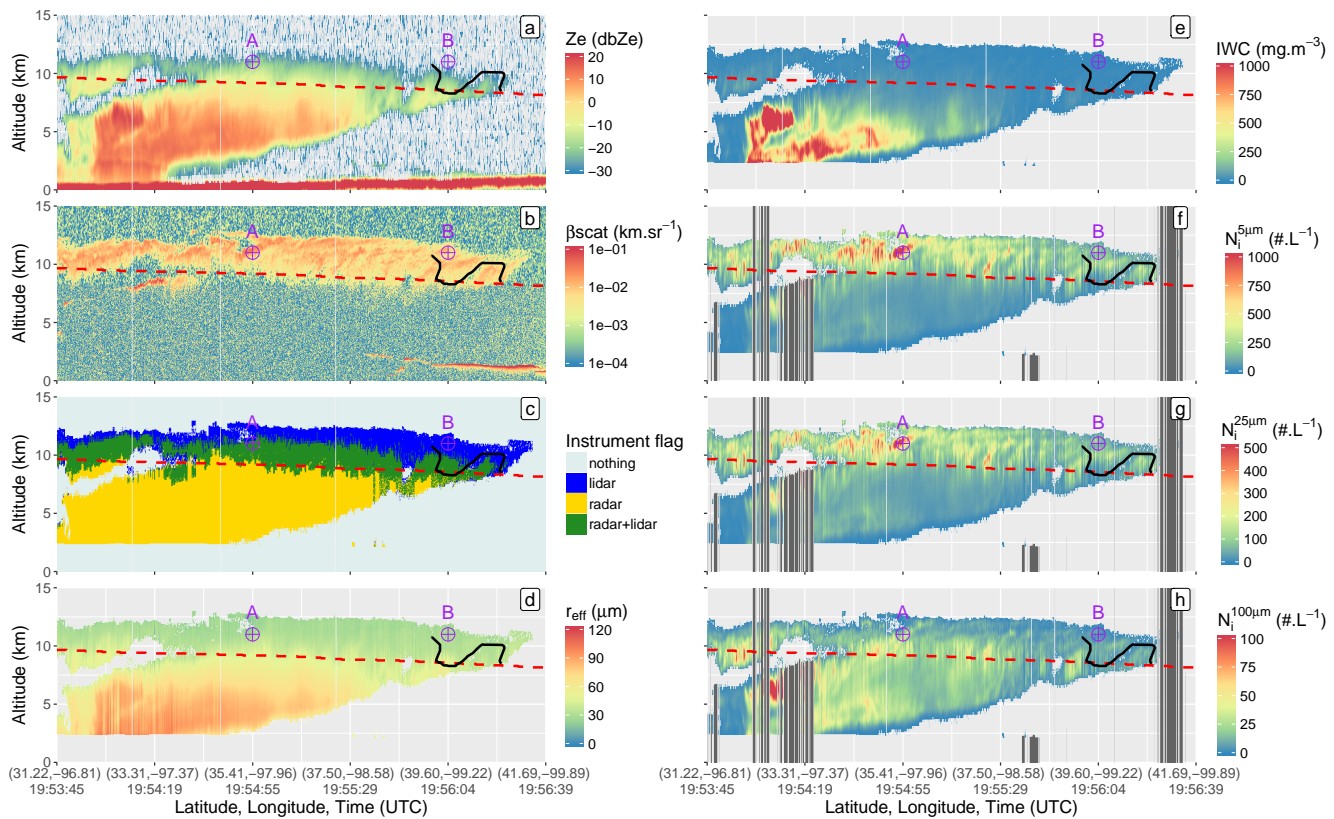

**Figure 6.** Vertical profiles of the (a) CloudSat reflectivity factor, (b) CALIOP backscatter coefficient, (c) DARDAR instrument flag, (d) $r_{eff}$, (e) IWC, (f-h) DARDAR-LIM $N_i^{5\,\mu m}$, $N_i^{25\,\mu m}$ and $N_i^{100\,\mu m}$, respectively, along the selected A-Train overpass (plain green line in Fig. 5). Dark-grey shaded areas in (f-h) indicate a rejection of the retrievals (insufficient $n_{iter}$ or the below a supercooled layer; see Sec. 3.1). The -40°C isotherm is in dashed red, the location of the A and B air parcels are indicated in purple, and the Learjet-25 track is shown in black.

occurrence of homogeneous nucleation below that temperature threshold. However, the increase in $N_i^{5\,\mu m}$ when $T_c < $ -40°C is not spatially homogeneous. Indeed, very high values (reaching $1000\,L^{-1}$) are observed between 33.3 and 36.5°N, where the cloud-base heights starts increasing in relation with the frontal system. However, lower $N_i^{5\,\mu m}$ with values, typically between 250 and $500\,L^{-1}$, are observed above -40°C toward the cyclone periphery (36.5 to 41.5°N). This distribution of $N_i^{5\,\mu m}$ could be
5   consistent with expectations of stronger updraughts and vertical transport of moisture near the center of the cyclone. This could result from high homogeneous freezing rates of aqueous aerosols that occur on top of existing ice crystals heterogeneously formed from liquid droplets (the area of high backscatter around 33.3°N in Fig. 6(b) indicates the presence of supercooled liquid water). This hypothesis will further be discussed in Sec. 5.3.

The vertical distribution of $N_i^{25\,\mu m}$ in Fig. 6(g) shows patterns that are similar to $N_i^{5\,\mu m}$, with absolute values that are about
10   50% lower. Profiles of $N_i^{100\,\mu m}$ shown in Fig. 6(h) indicate that areas of high $N_i^{100\,\mu m}$ are located deeper in the cloud than where high $N_i^{5\,\mu m}$ and $N_i^{25\,\mu m}$ are found. $N_i^{100\,\mu m}$ tends to increase below regions of high $N_i^{5\,\mu m}$, which is coherent with

possible aggregation processes, and remains constant or decreases slightly towards cloud base before precipitating from the lowermost layers. Very high $N_i^{100\,\mu m}$ values appear around 33.0°N, coincident with high IWC values. However, this area is subject to supercooled layers, where retrievals are highly uncertain. It is worth noting that the $N_i^{100\,\mu m}$ distribution does not necessarily follows that of the IWC.

Finally, it can be noted when comparing the $N_i$ profiles in Fig. 6(f-h) with the corresponding instrumental flags shown in Fig. 6(c) that no clear bias is observed within transition areas between the cloud properties obtained in lidar-only, radar-only and lidar-radar conditions, consistent with conclusions from Sec. 4.2.

A detailed analysis of 2D-S measurements along the co-incident flight leg (see Fig. S7) showed good overal agreements with DARDAR-LIM estimates of $N_i^{5\,\mu m}$, $N_i^{25\,\mu m}$ and $N_i^{100\,\mu m}$. Despite few discrepancies, the overall variations of $N_i$ appear well captured by DARDAR-LIM, in particular along the ascending leg, which has the highest time and space coincidence with the satellite overpass.

## 5.3 Trajectory analysis of $N_i^{5\,\mu m}$ patterns

Thorough investigations of nucleation processes based on DARDAR-LIM are not in the scope of this paper and will further be discussed in the part two of this series. Nevertheless, a qualitative analysis is here presented to provide further explanation to the $N_i^{5\,\mu m}$ patterns observed in this case study.

To achieve this, back- and forward-trajectories were computed from the points A and B indicated in Fig. 6, using the Hybrid Single Particle Lagrangian Integrated Trajectory Model (HYSPLIT; Stein et al., 2015) coupled with the North American Regional Reanalysis (NARR; Mesinger et al., 2006) model. A and B correspond to air parcels associated with high and low $N_i^{5\,\mu m}$, respectively, when $T_c < $ -40°C. Their trajectories (from -12 to +4 h starting at the overpass time) are shown as dashed and plain purple lines, respectively, in Fig. 5(a). The corresponding altitudes are similarly indicated as function of time in Fig. 5(c-d). Complementarily, the vertical cross-sections of vertical wind velocities ($w$) predicted by NARR along the trajectories are indicated by a color background in Fig. 5(c-d). Contours of the NARR IWC are shown in grey to serve as rough indicators of the presence of ice cloud layers in the model. It can be noted that NARR provides reanalyses with an horizontal resolution of 32 km, 29 vertical levels and with 3-hourly outputs; the closest output times and grid-points were therefore selected.

It is observed that, in agreement with expectations, both air parcels originated from warm moist air over the Pacific and slowly ascended atop the cold continental air following northwesterly trajectories associated with slow vertical motions (light brown colors). Parcel B ascended from about 8 km at 08:00 UTC to 11 km at the overpass time (20:00 UTC) and then remained at a constant altitude. The -40°C isoline (dashed red line) is crossed around 14:00 UTC and this parcel then appears to belong to an anvil-like maturing cloud layer from 19:00 UTC, supported by small $w$ (less than 1 cm.s$^{-1}$) observed around the overpass time. On the contrary, parcel A, which also started ascending from an altitude around 7.5 km, met stronger uplifts later during the day, around 16:00 UTC. Consistently, the -40°C isoline is also crossed about 2 h later than for parcel B. This parcel keeps ascending upon meeting with the overpass, where updraughts stronger than 5 cm.s$^{-1}$ are found.

These observations are in agreement with the high $N_i^{5\,\mu m}$ values observed at A, thus likely to be caused by strong and recent updraughts. The high sensitivity to $w$ could indeed relate to high in situ homogeneous freezing event of aqueous aerosols,

occurring on top of ice crystals formed, probably heterogeneously, from liquid droplets (Kärcher and Ström, 2003; Kärcher et al., 2006; Krämer et al., 2016). On the contrary, B corresponds to an air parcel within a mature cloud, where $w$ is too small to cause further ice nucleation and small ice crystals have already started to sublimate or aggregate. It is worth mentioning that these variations of $N_i$ with $w$ are in good agreement with findings by Kärcher et al. (2006) based on a physically-based ice nucleation scheme. An exact comparison of absolute numbers is however difficult as Kärcher et al. (2006) also shows a dependence on background ice nuclei concentrations, which are unknown for this case study.

Keeping in mind possible uncertainties associated with the NARR reanalyses and HYSPLIT trajectories, this analysis still provides comforting arguments as to physical meaningfulness of $N_i^{5\,\mu m}$ and $N_i^{25\,\mu m}$ patterns in DARDAR-LIM.

## 6 Global $N_i$ climatologies

Spatial distributions of $N_i^{5\,\mu m}$, $N_i^{25\,\mu m}$ and $N_i^{100\,\mu m}$ corresponding to 10 years of DARDAR-LIM products are now analysed. A thorough evaluation of these distributions remains difficult due to a lack of other reference for such climatological data; preliminary results are thus discussed here to assess the overall coherence of the observed $N_i$ patterns with general expectations. The interpretation of these distributions in terms of evidence of controls on $N_i$ are here only briefly addressed as they will be the focus of part two.

### 6.1 Geographical distributions

Figures 7(a,a-e) show the spatial distribution of $N_i^{5\,\mu m}$ averaged in a $2\times2°$ lat-lon grid and subset into 10°C bins from -80 to -30°C. Corresponding pixel counts are shown in Fig. S8. The $N_i^{5\,\mu m}$ shows a strong temperature dependence, with higher $N_i^{5\,\mu m}$ values being observed at colder $T_c$ globally (Fig. 7(a,a-e)). This $T_c$ dependence is particularly important over tropical land regions and in regions experiencing strong convection (the tropical warm pool, intertropical convergence zone). This is consistent with the strong updraughts in convective regions producing high supersaturations and so higher nucleation rates, causing these increased $N_i^{5\,\mu m}$ values (Kärcher and Lohmann, 2002; Krämer et al., 2016).

Low $N_i^{5\,\mu m}$, below $100\,L^{-1}$, are observed in subsidence regions, where thin cirrus are typically met. Nevertheless, it should be kept in mind that these regions contain only few ice clouds, most likely remnants of jets stream or tropical anvils (see. Fig. S8). On the contrary, maximum mean $N_i^{5\,\mu m}$ values, between 200 and $300\,L^{-1}$, appear at very low temperatures and in deep convective regions. It can be noted that these numbers are about 2-3 times higher than the concentrations reported by Jensen et al. (2013a, 2016) for TTL cirrus, which are more consistent with the $100\,L^{-1}$ that are here found in subsidence regions. This could hint at an overestimation of $N_i^{5\,\mu m}$ by DARDAR-LIM at these temperatures, possibly related to the too steep increase of the concentration towards small ice crystals noted for D05 in Fig. 1 and 2. However, it is difficult based on Fig. 7 to disentangle the contributions from different cloud types or meteorological conditions to $N_i$, which would be required to properly compare these $N_i$ climatologies to specific in situ measurements. This point will be further discussed in part two.

There is also a strong $T_c$ relationship in orographic regions, but it is prominent at warmer temperatures, with a large increase in $N_i^{5\,\mu m}$ being observed in the Himalayas, the Rockies, Southern Andes and the Antarctic Peninsula, as well as the edge of

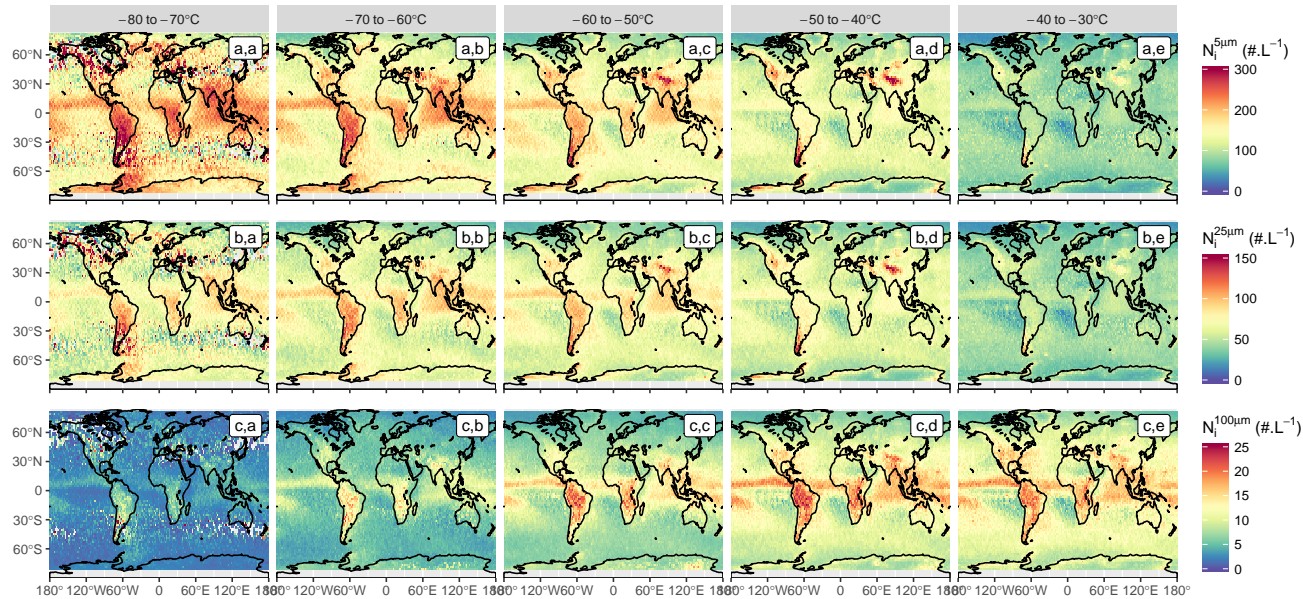

**Figure 7.** Spatial distribution of $N_i^{5\,\mu m}$ (a,a-e), $N_i^{25\,\mu m}$ (b,a-e) and $N_i^{100\,\mu m}$ (c,a-e) from 2006 to 2016, averaged in a $2 \times 2°$ lat-lon grid and per 10°C temperature bin from -80 to -30°C.

the East Antarctic ice sheet (Fig. 7(a,d)). These higher $N_i^{5\,\mu m}$ values are typically found in the mid-latitudes, where higher windspeeds provide stronger orographic uplifts (Gryspeerdt et al., 2018a). Consequently, such features are less likely in the tropics, where the atmosphere is barotropic. This is for instance clearly noted in the Andes, where no high $N_i^{5\,\mu m}$ values appear at the northern end.

5 Similar features are seen in the distributions of $N_i^{25\,\mu m}$, with absolute values that are about 50% smaller than $N_i^{5\,\mu m}$. Maximum $N_i^{25\,\mu m}$ values, found at low $T_c$ in the tropics or in orographic regions, are therefore of about $100\,L^{-1}$. It is reminded that $N_i^{25\,\mu m}$ is relatively robust to PSD shape assumptions (see sec. A3) and was found to agree well with in situ measurements for $T_c < -60°C$. Another notable difference with $N_i^{5\,\mu m}$ is the weaker temperature dependence of $N_i^{25\,\mu m}$ below -50°C, particularly in convective regions. This could be due to ice crystals larger than $25\,\mu m$ being less directly related to homogeneous freezing 10 events.

The spatial distribution of the $N_i^{100\,\mu m}$ shown in Fig. 7(c,a-e) is noticeably different, with a significantly reduced $N_i^{100\,\mu m}$ at lower temperatures. This might be expected through the reduced efficiency of the aggregation and deposition processes needed to generate larger crystals at colder $T_c$, along with the size-sorting of ice crystals in cirrus clouds. At all the temperatures examined, higher $N_i^{100\,\mu m}$ values are observed in convective regions, where updraughts are sufficient to transport large particles 15 to the upper troposphere, but also where high $N_i^{5\,\mu m}$ and $N_i^{25\,\mu m}$ at colder temperatures subsequently lead to high $N_i^{100\,\mu m}$ as the clouds mature.

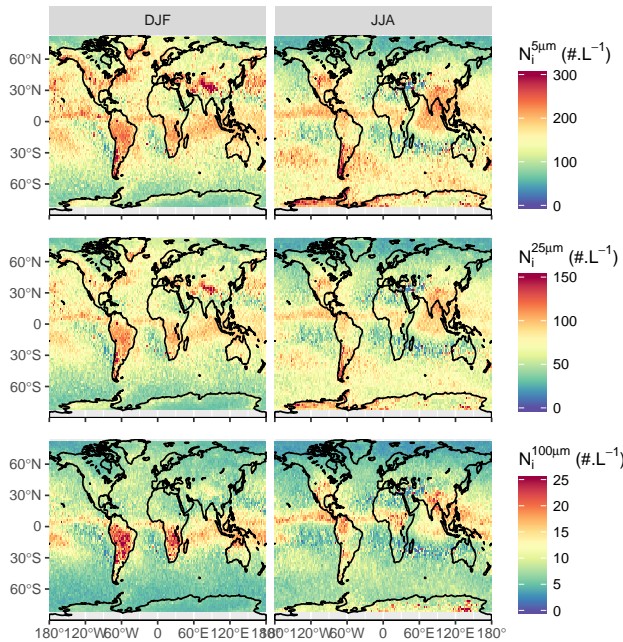

**Figure 8.** Spatial distribution in $N_i^{5\,\mu m}$, $N_i^{25\,\mu m}$ and $N_i^{100\,\mu m}$ between -60 and -50°C , during northern hemisphere winter (DJF; a-b,a) and summer (JJA; a-b,b) seasons.

The seasonal variability of $N_i$ spatial distributions in the -60 to -50°C bin is shown in Fig. 8. Strong variations are found in the tropics and along the ITCZ, where large cloud structures are convectively driven. High $N_i$ values are noted in these regions during summer seasons, therefore strengthening their link to freezing events associated with deep convective structures. These values typically decrease by a factor of 2-3 during winter seasons. Inversely, $N_i^{5\,\mu m}$ and $N_i^{25\,\mu m}$ in the mid-latitude storm tracks and orographic regions are found to be higher during winter months, consistent with stronger jets (Gryspeerdt et al., 2018a).

Comparisons of these results with recent findings by Mitchell et al. (2016, 2018) based on thermal-infrared and lidar measurements show good consistency in mid-latitude regions (increases in orographic and storm track regions). Lower absolute $N_i$ values are found here, possibly due to the use of a $D_{min}$ threshold. These two studies also identify a strong decrease of $N_i$ in the tropics between -55°C and -45°C that do not clearly appear in Fig. 7(a,c-d). A possible explanation for this could be the different cloud sampling between the two methods. Lidar and thermal-infrared measurements indeed only provide the concentrations of thin-to-moderately thick cirrus or near cloud-top found at this temperature range, whereas Fig. 7 also indicates $N_i$ within deep convective clouds, where high values can be expected (Paukert et al., 2017). More consistent comparisons with these studies would therefore require to look at the spatial distributions of DARDAR-LIM $N_i$ at cloud top. These will be analysed in part two.

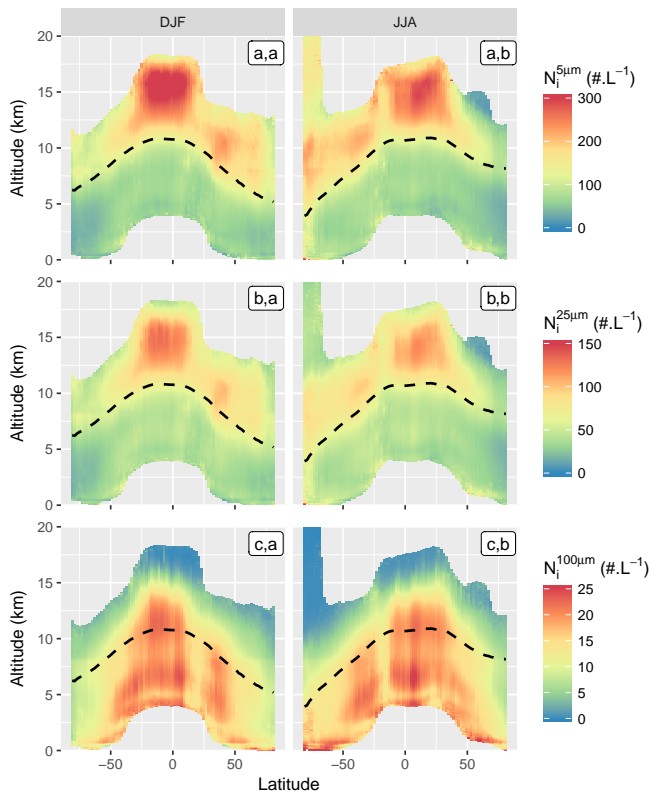

**Figure 9.** Zonal profiles of $N_i^{5\,\mu m}$ (a,a-b), $N_i^{25\,\mu m}$ (b,a-b) and $N_i^{100\,\mu m}$ (c,a-b), during northern hemisphere winter (DJF; a-b,a) and summer (JJA; a-b,b) seasons. The -40°C isoline is shown as a dashed black line.

## 6.2 Zonal profile distributions

Subsets of zonal profiles distributions of $N_i^{5\,\mu m}$, $N_i^{25\,\mu m}$ and $N_i^{100\,\mu m}$ corresponding to the northern hemisphere winter (DJF) and summer (JJA) seasons are presented in Fig. 9. Corresponding pixels counts are provided in Fig. S9.

As with the global maps in Fig. 7, a dependency between $N_i$ and the temperature is observed. It is clear from these zonal plots
5  that this relationship is not linear, as $N_i^{5\,\mu m}$ and $N_i^{25\,\mu m}$ nearly double their values upon crossing the -40°C isoline (dashed line). Because homogeneous nucleation rates become significant at colder temperatures (Koop et al., 2000), this suggests that the $N_i^{5\,\mu m}$ at temperatures colder than -40°C is strongly influenced by homogeneous freezing. The increase in $N_i^{5\,\mu m}$ is particularly strong in the tropics, where convective updraughts may be able to generate the high supersaturations required for homogeneous nucleation.

10   It is also observed that $N_i^{25\,\mu m}$, and $N_i^{5\,\mu m}$ to a lesser extent, decrease at very low temperatures towards the tropopause. This could indicate lower $N_i^{25\,\mu m}$ and $N_i^{100\,\mu m}$ at cloud top, but should be carefully considered due to the low statistical significance of retrievals in the high troposphere (see Fig. S9). This feature also appears in convective regions and could hint at lower $N_i$ in

very cold TTL cirrus by comparison to the high values found within convective structures below the freezing level. It can be noted that the vertical structure of $N_i^{5\,\mu m}$ and $N_i^{25\,\mu m}$ observed here in the tropics is consistent with recent simulations of deep convective clouds by Paukert et al. (2017). Finally, it should be mentioned that the $N_i^{5\,\mu m}$ values of about $150\,L^{-1}$ observed above 12 km over Antarctica during the winter season are highly uncertain due to likely issues in the cloud mask and retrievals in this region.

Consistent with the global maps in Fig. 7(c,a-e), there is a strong decrease in $N_i^{100\,\mu m}$ with decreasing temperature. This temperature dependence is much stronger for $T_c < $ -40°C, becoming much weaker at warmer temperatures. Despite being lower than $N_i^{5\,\mu m}$, at warmer temperatures, the $N_i^{100\,\mu m}$ reaches values higher than $20\,L^{-1}$, such that large crystals comprise a significant fraction of the total $N_i^{5\,\mu m}$ (50-100 $L^{-1}$). The temperature dependence will be further investigated in part 2.

## 7  Summary and conclusions

A novel approach to estimate $N_i$ from combined CALIPSO-CloudSat measurements, called DARDAR-LIM, is here presented and evaluated against in situ measurements and in the context of a case study and a preliminary climatological analysis.

Based on about 40 000 PSD measurements from five recent in situ campaigns, it is demonstrated that $N_i$ can be predicted by constraining the moments of normalized PSDs using $\beta_{ext}$ and $Z_e$ measurements. The D05 parameterization appears capable of predicting reasonably well the measured concentration of particles from different minimum size thresholds and for ice clouds with $T_c$ spanning from -90 to -30°C, demonstrating good predictions of $N_i^{5\,\mu m}$, $N_i^{25\,\mu m}$ and $N_i^{100\,\mu m}$ (Fig. 2). A possible bias in $N_i^{5\,\mu m}$ and $N_i^{25\,\mu m}$ predictions is nevertheless noted when $T_c \gtrsim$ -50°C and is explained by a misrepresentation in D05 of the bi-modality observed in the measured PSDs (Fig. 1). A slight overestimation of $N_i^{5\,\mu m}$ by D05, by a factor less than 2, is also noted when $T_c < $ -60°C due to its too steep representation of the concentration in particles smaller than $25\,\mu m$. $N_i^{25\,\mu m}$ does not seem significantly impacted, as it is less sensitive to PSD shape assumptions (Sec. A3).

Following these results, it is verified that $N_i$ estimates inferred from IWC and $N_0^*$ retrievals of DARDAR, which uses D05, are also in good agreement with the in situ measurements from co-incident flights (Fig. 3-4). Good agreements are observed between DARDAR-LIM and co-incident D05 predictions based on the co-incident in situ measurements, thus demonstrating the sufficient sensitivity in $\beta_{ext}$ and $Z_e$ to constrain $N_i$. Observed differences between DARDAR-LIM and the in situ data are consistent with the aforementioned expectations and are explained by limited PSD shape assumptions in D05. It was noted that lidar-only estimates of $N_i$ are possible because, despite being associated with fewer observational constraints, they correspond to PSDs observed at cloud top or in thin cirrus that tend to follow a monomodal shape. Due to the strong bi-modality of PSDs associated with radar-only retrievals, corresponding estimates of $N_i^{5\,\mu m}$ and $N_i^{25\,\mu m}$ are more difficult. Finally, it should be noted that these analyses of co-incident flights are only based on 2D-S measurements, which can be uncertain for sizes between 5 and $25\,\mu m$. More in situ comparisons will therefore be necessary to further evaluate $N_i^{5\,\mu m}$ from DARDAR-LIM.

Reasonable physical consistency is also found in the vertical distribution of $N_i$ estimates analysed along a short orbital track in the context of a case study representative of an occluded frontal system. $N_i^{5\,\mu m}$ and $N_i^{25\,\mu m}$ are found to increase below -40°C (Fig. 6(f-g)), in conformity with higher homogeneous nucleation rates. Large $N_i^{100\,\mu m}$ values are found deeper in thick cloud

layers. As expected, very good consistency is observed between estimates obtained in lidar-only, radar-only and lidar-radar conditions. Based on a quantitative analysis of the trajectory of two air masses, it is observed that regions that are subject to stronger updraughts and therefore supplied moisture show peaks in $N_i^{5\,\mu m}$, whereas regions representative of mature cloud parcels do not. Direct comparisons to aircraft measurements that are co-incident with the satellite track again confirm that
DARDAR-LIM reproduces well the overall values and spatial variability measured in situ.

Finally, global distributions of $N_i^{5\,\mu m}$ and $N_i^{100\,\mu m}$ are analysed on the basis of a 10-year climatology (Fig. 7-9). An overall increase of $N_i^{5\,\mu m}$ with decreasing temperature is observed but its rate is regionally dependent. A global increase is observed as $T_c$ reaches -40°C, consistent with a strong temperature dependency of the homogeneous nucleation rate. However, steep increases when $T_c < $ -50°C are only observed in regions where uplifts are sustained by convection or orography, in agreement
with expectations of high sensitivity of the $N_i$ to updraughts.

The lidar-radar $N_i$ estimates introduced in this study thus constitute a first and very encouraging basis to provide global observational constraints of this quantity, which open the door to a better understanding of cloud processes and their evaluation in climate and numerical weather prediction models. Improvements of the method remain necessary to reduce the uncertainties related to these $N_i$ estimates. In particular, the use of a PSD parameterization that is better fitted to retrieving $N_i$. This includes a
better representation of bi-modality and possibly a less steep increase of the concentration in small particles at low temperatures. Further comparisons to in situ measurements as well as modeling are also intended to continue to evaluation of this new $N_i$ product. A detailed investigation of the controls on $N_i$ based on the DARDAR-LIM dataset is presented in part two of this series.

## Appendix A: Expected uncertainties and limitations

### A1   $N_i$ under lidar- and radar-only conditions

The simultaneous use of lidar and radar observations makes DARDAR sensitive to a wide variety of ice clouds, with IWPs spanning from about $0.01\,\mathrm{g.m^{-2}}$ to $5\,\mathrm{kg.m^{-2}}$ (Sourdeval et al., 2016). Nevertheless, it should be kept in mind that the information provided by these two instruments does not always overlap, leading to lidar- and radar-only regions in the vertical profiles where cloud layers are optically very thin or thick, respectively. Fewer constraints are applied on D05 in this partial or complete
absence of information from one instrument and so retrievals may be more uncertain.

Under such conditions, DARDAR relies on *a priori* information provided by an empirical relation between $N_0^*$, $\alpha_{\mathrm{ext}}$ and the layer temperature $T_c$ (Hogan et al., 2006). This relation is further constrained in the occurence of lidar-radar conditions within the same column to improve the physical consistency of the *a priori* constraints (Hogan, 2007). The exact weight of these constraints on lidar- and radar-only retrievals is nevertheless difficult to quantify from the operational products only.
The number of iteration is here used as a proxy to avoid any strong influence of *a priori* assumptions on the retrievals; Cloud products associated with $n_{\mathrm{iter}} < 2$ are excluded from this study.

Following these considerations, it can be expected that DARDAR retrievals are optimal when both lidar and radar measurements are available. However, predicting the consequence of lidar-only or radar-only conditions on $N_i$ is not trivial. Reasonable

$N_i$ estimates should be possible in lidar-only regions due to $\beta_{\text{ext}}$ being sensitive to small particles (to $\mathcal{M}_2$), which largely contribute to $N_i$ ($\mathcal{M}_0$). PSDs observed in lidar-only conditions, i.e. towards cloud-top, are also likely to be monomodal and therefore easier to represent for D05. Radar-only estimates may be more difficult due to $Z_e$ being mainly sensitive to large particles (to $\mathcal{M}_6$) and will therefore depend on the capability of D05 to extrapolate the concentration in small particles. These questions will be investigated and discussed throughout the manuscript (see in particular Sec. 4.2).

## A2   Propagation of DARDAR operational errors

A qualitative estimation of the errors expected on $N_i$ can be obtained based on the uncertainties associated to DARDAR IWC and $N_0^*$ retrievals. These account for instrumental errors and uncertainties attached to non-retrieved parameters used to simulate the lidar and radar measurements. A direct propagation of these errors on $N_i$ can be reached from $\sigma_{N_i}^2 = \left(\frac{\partial N_i}{\partial \text{IWC}}\right)^2 \sigma_{\text{IWC}}^2 + \left(\frac{\partial N_i}{\partial N_0^*}\right)^2 \sigma_{N_0^*}^2$. The variances $\sigma_{\text{IWC}}^2$ and $\sigma_{N_0^*}^2$ are provided in DARDAR and the partial derivatives can be solved from the equations provided in Sec. 2.2. The relative uncertainties $\sigma_{N_i}$ / $N_i$ typically increase from about 25% in lidar-radar conditions to 50% in lidar-only or radar-only (see Figure S2), which confirms expectations of lidar-radar estimates being more precise due to a higher information content. However, these numbers only provide rough estimates of the actual uncertainties expected on $N_i$ as no rigorous errors are associated with non-retrieved parameters that are critical for its accuracy. For instance, the errors associated with PSD shape assumptions are only represented by fixed errors on $\beta_{\text{ext}}$ and $Z_e$ simulations instead of being computed from the exact sensitivity of these measurements to $\alpha$ and $\beta$ to the atmospheric state at each iteration.

## A3   Influence of PSD shape assumptions on $N_i$

Important uncertainties on $N_i$ can be expected from the choice of $\alpha$ and $\beta$ in Eq. (4). The former parameter is especially critical as it controls the steepness of the PSD towards small particles and strongly varies depending on the dominating nucleation processes (Mitchell, 1991). The normalization approach by D05 should in principle account for such variations in the PSD shape by adjusting the scaling parameters $N_0^*$ and $D_m$, but large deviations from the assumed $\alpha$ value due to unusual conditions could still have consequences on subsequent $N_i$ estimates. For instance, an overestimation of $\alpha$ (i.e., a less negative $\alpha$) in D05 will lead to an underestimation of $N_i$. This could occur in the presence of very high homogeneous nucleation rates (e.g., related strong orographic updraft) where $\alpha$ is very negative. Reciprocally, an underestimation of $\alpha$ is possible in case of highly dominant aggregation processes and will lead to an overestimation of $N_i$. The variability of $\alpha$ and $\beta$ between numerous airborne campaigns has been investigated by D14 in order to propose an updated version of the PSD parameterization for future DARDAR versions. Based on the shape parameters extracted in this study, it could be estimated that uncertainties within about 50% (usually an overestimation) can reasonably be expected on $N_i$ as results of variations from the PSD shape assumed in DARDAR (see Fig. S3 for details). Lower uncertainties should nevertheless be expected if the PSD is not too broad (i.e., in cases where homogeneous nucleation dominates) and if very small particles are discarded by a high integration threshold. Consequently, $N_i^{25\,\mu\text{m}}$ is less sensitive to errors on $\alpha$ and $\beta$ than $N_i^{5\,\mu\text{m}}$. These numbers remain preliminary as the average $N_0^*$ and IWC values used here may not be fully representative of each couple of $\alpha$ and $\beta$. Finally, it can be noted that the

coefficients chosen by D14 for future DARDAR versions, noted "all (DARDAR)" in Fig. S3, should lead to smaller $N_i^{5\,\mu m}$ and $N_i^{25\,\mu m}$ values due to a less negative $\alpha$.

*Data availability.* The DARDAR-LIM data created in this study will promptly be available via the ICARE data center (http://www.icare.univ-lille1.fr) or can be obtained from the corresponding author upon request. The DARDAR product was retrieved via ICARE and the MODIS

MYD09CMG product via the NASA Land Processes Distributed Active Archive Center (https://e4ftl01.cr.usgs.gov). All in situ data is available through mission-based databases and can be accessed after signing a data agreement. ACRIDICON-CHUVA and ML-CIRRUS are available via the DLR HALO database (https://halo-db.pa.op.dlr.de), COALESC via the MetOffice FAAM database (http://data.ceda.ac.uk/badc/faam), SPARTICUS via the ARM data discovery center (https://www.archive.arm.gov/discovery/) and ATTREX via the NASA Earth Science Project Office (https://espoarchive.nasa.gov).

*Acknowledgements.* The authors gratefully acknowledge the science teams involved in collecting and providing the airborne measurements used in this study. The authors are grateful to P. Lawson and S. Woods for helpful discussions regarding the processing of 2DS and FCDP data from these two campaigns. The NOAA Air Resources Laboratory and National Center for Atmospheric Prediction is acknowledged for the provision of HYSPLIT trajectories and NARR reanalyses. We are grateful to the Deutsches Klimarechenzentrum (DKRZ) for providing computational resources necessary for this study. This work was funded by the European Research Council (Grant 306284 "QUAERERE"),

by the Federal Ministry for Education and Research in Germany (Bundesministerium für Bildung und Forschung, BMBF) in the "HD(CP)[2]" project (FKZ 01LK1210D, 01LK1503A and 01LK1505E) and by the German Research Foundation (Deutsche Forschungsgemeinschaft, DFG) in Priority Programme SPP 1294 "HALO", project QU 311/14-1 ("FLASH"). TG received funding from the European Union Horizon 2020 research and innovation programme under the Marie Sklodowska-Curie Grant Agreement No. 703880. EG was supported by and Imperial College London Junior Research Fellowship. We acknowledge support from the German Research Foundation (DFG) and Leipzig

University within the program of Open Access Publishing.

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
