# Peer review of "Ice crystal number concentration estimates from lidar-radar satellite remote sensing. Part 1: Method and evaluation"

_Atmospheric Chemistry and Physics, 2018_

## Referee Comment (RC1) · Anonymous Referee #3 · 27 Feb 2018

The article presents a method to be used as an operational retrieval to derive ice crystal number concentrations of pure ice clouds, Ni, (T < -30°C) from combined spaceborne lidar-radar measurements (CALIPSO-CloudSat) and a thorough evaluation using in situ data from five airborne campaigns. An example of application is shown via a case study, including Lagrangian transport modelling. An interesting result is that regions with stronger updraughts show peaks in Ni with particle sizes > 5micron in contrast to regions of mature cloud, as one would expect. At the end, geographical maps and zonal profiles of 10 years of Ni are presented and discussed for particles with sizes > 5micron and >100micron. A follow-up paper will use these data in the framework of aerosol-cloud interactions. Ni is an essential microphysical parameter, which is re-

cently used as a prognostic variable in climate models, and therefore it is important to have global observational constraints. The variable is also important for process studies. The combination of lidar and radar measurements, being part of the A-Train, allows to determine the vertical structure of clouds such as top and base of the clouds, cloud layering, as well as ice water content and effective ice crystal diameter. The attempt to derive ice crystal number concentration is relatively recent, as its determination depends on several assumptions (in particular a gamma-modified particle size distribution (PSD) and a specific ice crystal mass - maximal diameter relationship). The presented method is based on a direct constraint of the shape of normalized particle size distributions using lidar extinction and radar reflectivity from the operational liDAR-raDAR (DARDAR) products. 40000 in situ PSD's are used for an evaluation, investigating results separately for ice crystal sizes > 5, 25 and 100 micron, first for the prediction of PSD from N0* and Dm and then for retrieved Ni. The article is generally well structured and well written. I strongly recommend the publication of this article, after minor revisions.

Minor Comments

1) The methodology section 2 gains in clarity by integrating the content of section 2.1 into sections 2.3 and section 3.1, in particular as DARDAR products are data and the retrieved variables such as beta-ext, Ze and beta-ext are not defined. In that way the section on the representation of the size distribution gets section 2.1, in which the advantage of using scaled PSD's is described as well as the necessity to assume a certain m-D relationship and a certain shape of PSD. The new section 2.2 (Extracting Ni from DARDAR products) goes then further into detail how to extract Ni from the DARDAR products N0* and IWC. It should be clearly stated in the beginning that from N0* and IWC from DARDAR one deduces Dm and finally Ni. New Section 2.1: Be careful of replacing DARDAR by 'DARDAR retrieval (see section 3.1)'. Then a short description of the DARDAR products (like in initial sect 2.1) should be integrated into section 3.1. P4,l17-18 define beta-ext as (lidar) extinction and Ze as (radar) reflectivity

2) p 6, l22 it is stated that DARDAR retrievals of pure ice clouds for which the iterative retrieval converged too quickly are ignored. How many of these retrievals are these and can you explain which category of cases these are?

3) The evaluation of the prediction of PSD's and Ni (using all field campaigns) and later for retrieved Ni (using coincident SPARTICUS measurements)is shown separately for different temperature intervals, which is important as ice crystal particles shapes differ with temperature. It would be very interesting to separate also anvils and synoptic cirrus, as m-D relations might be different. Is there enough statistics of the collocated SPARTICUS campaign measurements to compare Ni distributions of Fig. 5 for anvils and synoptic cirrus?

4) section 3.2.2: One specific ice crystal mass – maximum diameter (m-D) relationship is used to determine IWC from the PSD. Indeed, Delanoë et al. 2014 show that the uncertainty to the m-D relationship for the normalized PSD is less important when minimizing using lidar extinction and radar reflectivity. The uncertainty seems to increase if only the lidar extinction is used for the minimization (Fig. 9). As both measurements are complementary, there are clouds for which only the first (thin cirrus) or the latter (towards the base of thick cirrus) are available. We also know that the shape of crystals changes with temperature and Heymsfield et al. 2010 showed that the m-D relation for anvil ice clouds yield masses about a factor of 2 larger than for synoptic ice clouds. Erfani and Mitchell 2016 cite this result in their paper and write that their results showing a similarity in m-D expressions between these two cloud types might be an artefact if the ice particle masses for a given projected area are quite different between these types. The L16 m-D relationship was developed for midlatitude cirrus. So for tropical anvils the computed IWC might be biased. Did you test the IWC computed with the L16 m-D relationship with the measured IWC for tropical anvils (using SPARTICUS and ATTREX) ?

5) Figs. 6 c and d of the case study present the trajectories as function of UTC. The relevant variable is the time difference which you show in brackets, and then the position on the map in Fig. 6a. If it is not too complicated, it might be clearer to present instead of UTC longitude.

6) concerning Fig. 5, is it possible to get also De from DARDAR for this cloud ?

7) The long descriptive text of the case study is sometimes difficult to follow. I suggest for example to move the analysis of the collocated air track comparison (Fig. 8) to a supplement.

8) I would rename section 6 'Presentation of global Ni climatologies' and 6.1 'Geographical distributions'. P21l5: 'considered with caution' instead of 'cautiously considered'

Additional references

p 2, l 8: an IR spectral approach should also be mentioned: Guignard, A. C. J. Stubenrauch, A. J. Baran, and R. Armante, 2012: Bulk microphysical properties of semi-transparent cirrus from AIRS: a six year global climatology and statistical analysis in synergy with geometrical profiling data from CloudSat-CALIPSO, Atmos. Chem. Phys., 12, 503-525, doi: 10.5194/acp-12-503-2012

p 2, l 25: for liquid clouds one should also cite Han, Q., W. B. Rossow, J. Chou, and R. M. Welch, 1998: Global variation of column droplet concentration in low-level clouds, Geophys. Res. Lett., 25, 9, 1419-1422, doi :10.1029/98GL01095. Aerosol-Cloud Interactions with this dataset have been studied in : Han, Q., W. B. Rossow, J. Zeng, and R. M. Welch, 2002: Three Different Behaviors of Liquid Water Path of Water Clouds in Aerosol–Cloud Interactions, J. Atmos. Sci., 59, 726-735.

Typos: p 2, l 6: based on passive p 3, l 16: 'a suitable' instead of 'an suitable' p 6, l 8 : 'to demonstrate' instead of 'to demonstrated' p 15, l 12-13: 'a statistical comparison' instead of 'a statistical comparisons' p 16, l 2: 'is also indicated' instead of 'also is indicated' p 16, l 7: 'is observed' instead of 'tends is observed'

This comment is also available in a more readable format as attached pdf file.

Please also note the supplement to this comment:
https://www.atmos-chem-phys-discuss.net/acp-2018-20/acp-2018-20-RC1-
supplement.pdf

---

## Referee Comment (RC2) · Anonymous Referee #1 · 2 Mar 2018

General Comments:

This paper presents a new method for retrieving the ice particle number concentration $N_i$ for glaciated clouds, which should be useful for understanding aerosol interactions with ice clouds and the contribution of homogeneous vs. heterogeneous ice nucleation in cirrus clouds. A satellite remote sensing scheme for $N_i$ is needed since field campaigns cannot adequately inform us how $N_i$ varies with latitude and the seasons. The paper is well organized and well written, and usually cites the relevant literature. The quality of the figures is good. The methods developed in Sec. 5 for testing the retrieval are especially creative and effective.

[Figure]

A critical limitation of the retrieval algorithm is the use of a normalized universal ice particle size distribution, or PSD (Delanöe et al., 2005), where it is assumed that all PSD in nature conform to this normalized PSD shape. This normalized PSD is based on a four-parameter gamma function (Eq. 4) where parameters No and k can be deduced through their link with other operationally retrieved properties (IWC and No*) while PSD parameters $\alpha$ and $\beta$ need to be fixed as constants. This is of little consequence regarding $\beta$, which affects the largest ice particles having the lowest concentrations. But this is of major consequence regarding $\alpha$, which strongly influences the smallest ice particles that govern Ni. This is not mentioned in the paper. The small end of the PSD is sensitive to the rate of ice nucleation which is sensitive to the cloud updraft w (with higher w making $\alpha$ more negative, and Ni higher), as well as the aggregation rate that removes smaller ice particles having higher concentration (Herzegh and Hobbs, 1985, QJRMS; Mitchell, 1991, JAS). Thus, some discussion on this topic is warranted, especially on the errors that may result from "non-standard" conditions where atypical updrafts are common (such as over steep orography).

The lead author gave a nice talk about this retrieval at the A-Train Symposium in 2017. Henceforth, Ni refers to Ni for ice particle maximum dimension D > 5 $\mu$m. Slide 20 of this presentation, showing global distributions of Ni for 10 °C intervals, appears almost identical to Fig. 9 of this paper for T < -30 °C, except that the Ni legends differ. The Ni values reported in the presentation are higher by a factor of $\sim$ 1.7 relative to the Ni reported in Fig. 9 of this paper. What is the reason for this difference?

A major finding from this study was a strong global temperature dependence for Ni, with Ni increasing with decreasing temperature (e.g. Fig. 9). Ni as observed from many field campaigns are reported in Krämer et al. (2009, ACP) where the middle value for Ni shows little temperature dependence. Although the SPARTICUS Ni measurements shown in Muhlbauer et al. (2014) show a temperature dependence, the study by Krämer et al. is based on many field campaigns. Please comment on this to help readers understand this apparent discrepancy.

Other important issues are discussed below.

Major Comments:

1. Page 8, line 25: The 2DS photodiode array length is 1280 $\mu$m, which should be noted. Evidently the "time dimension" is used to size particles up to 3205 $\mu$m; please indicate the particle selection criteria used to size and count particles.

2. Figure 5 and Sec. 4.2: For T > -50 °C, by what factor is Ni (Dmin = 5 $\mu$m) overestimated, on average? For T $\leq$ -50 °C?

3. Page 21, lines 9-12: The strong temperature dependence of Ni mentioned here appears at variance with the in situ measurements reported in Krämer et al. (2009). Please mention this.

4. Figure 9 and Sec. 6.1: For T > -50 °C, Ni tends to be lower over regions characterized by extensive marine stratus, like off the west coasts of South America and Africa (from equator and southwards). Is this result real, or is it an artefact of the retrieval? If the latter is true, please explain.

5. Page 21, lines 14-19: A similar finding was reported in Mitchell et al. (2016, ACPD), where the highest Ni were associated with mountainous terrain. (Although this paper was rejected since the editor felt the retrieved Ni values were too high, and therefore could not be used to infer nucleation modes, no arguments cast doubt on the spatial and temporal relative differences in Ni, which still appear meaningful.)

6. Page 22, lines 7-9: It is more meaningful to compare model results against observations than vice-versa. Suggest removing this paragraph. For example, in the modeling study by Zhou et al. (2016, ACP), the sensitivity of homo- and heterogeneous ice nucleation to various model parameters and updraft schemes were evaluated. Depending on how these are represented, one can get a broad range of Ni-temperature dependences, including Ni that is relatively insensitive to temperature (similar to the in situ observations of Krämer et al., 2009, ACP), and that modeling result would not support

these DARDAR-LIM findings.

7. Figure 10 and Sec. 6.2: Ni (Dmin = 5 $\mu$m) in the tropics appears contrary to the Ni results in Fig. 1 and Fig. 5 of Part 2 of this study by Gryspeerdt et al. (submitted). Fig. 1a of Gryspeerdt et al. show Ni near cloud top while their Fig. 5 shows that Ni does not change appreciably with distance below cloud top (up to 3 km from cloud top) between -50 and -60 °C. Assuming this result extends to other temperatures, the cloud top results in Fig. 1a of Gryspeerdt et al. should also be approximately valid below cloud top.

Regarding Fig. 1a in Gryspeerdt et al., for T $\geq$ -65 °C, Ni is never higher in the tropics relative to the midlatitudes. Between -55 and -40 °C, where the most optically thick cirrus clouds exist (cirrus defined as clouds having T $\leq$ -38 °C), Ni in the tropics is substantially lower than Ni in the midlatitudes. In Fig. 10 of Part 1 (Sourdeval et al.), Ni increases abruptly in the tropics for T < -40 °C (shown by the dashed curve), with Ni here being typically higher than Ni at similar T in the midlatitudes. This result appears opposite to the findings in Fig. 1a of Gryspeerdt et al. (Part 2). In addition, the CALIPSO Ni retrievals of Mitchell et al. (2016, ACPD) qualitatively support the findings of Gryspeerdt et al. (in terms of relative differences), and the in situ measurements from Muhlbauer et al. (2014) show relatively lower "peak Ni" values in anvil cirrus (vs. frontal, jet stream and ridge-crest cirrus). Finally, several studies (e.g. Jensen et al., 2013, PNAS; Spichtinger and Krämer, 2013, ACP), show that tropical tropopause layer (TTL) cirrus tend to have Ni < 30 L-1. Since the areal coverage of TTL cirrus exceeds that of anvil cirrus, and TTL cirrus tend to be higher than anvil cirrus (Gasparini et al., 2017, J. Climate), the Ni of $\sim$ 200 L-1 in the TTL region in Fig. 10 appears at variance with in situ observations. Please comment on, and, if possible, reconcile these issues.

8. Page 23, lines 1-3 and Fig. 10: Fig. 10 and this text indicate that in the midlatitudes for T < -40 °C, Ni is highest during winter and lowest during summer. This same result was found in Mitchell et al. (2016). One of the ACP review criteria questions is "Do the authors give proper credit to related work and clearly indicate their own new/original

contribution?".

Minor Comments:

1. Page 15, line 9: much => slightly?

2. Page 19, line 6: follows => follow?

3. Page 22, line 13: at => as?

4. Figure 10 caption: Mention the meaning of the dashed curve.

5. Page 20, line 1: an => a?
* * *

---

## Referee Comment (RC3) · Anonymous Referee #2 · 15 Mar 2018

**Review of "Ice crystal number concentration estimates from lidar-radar satellite remote sensing. Part 1: Method and evaluation" by O. Sourdeval et al.**

This paper describes an ice concentration retrieval based on the DARDAR Cloud-Sat/CALIPSO combined lidar-radar retrieval. The extension of DARDAR to retrieve ice concentrations, evaluation by comparison with in situ aircraft measurements, and global distributions are discussed. Although the ice concentration retrieval seems reasonable and potentially useful, I have significant concerns with the paper in its current version. In particular, I think the validity of the retrieval in regions without both lidar extinction and radar reflectivity needs much more discussion and evaluation. Also,

the use of 2D-S measurements for determining concentrations of small ice crystals is suspect at best. These issues (and others) are discussed in detail below.

**1.** The discussion of the retrieval algorithm in section 2 implicitly assumes that both extinction from the lidar and radar reflectivity are available. The authors should make clear early in the paper that the ice concentration retrieval is dubious in cirrus that are not detected by both radar and lidar (i.e., either too optically thin for detection by the CloudSat radar or below optically thick layers where the CALIOP lidar is blocked). When only lidar backscatter or radar reflectivity are available, the ice concentration is entirely dependent on the assumed size distribution. Mean PSDs are shown in the paper, but aircraft data shows that enormous PSD temporal and spatial variability is typically prevalent in cirrus. With only lidar or radar data available, this variability cannot be captured by the retrieval.

**2. Page 6, lines 24-28:** It would be helpful if some formal estimate of the uncertainties in $N_i$ retrievals associated with measurement uncertainties could be provided.

**3. Page 6, lines 26-27:** Further discussion of the the uncertainty in $N_i$ retrieval associated with PSD shape assumption should be included. Perhaps examples could be provided as a guide.

**4.** As noted in the manuscript, only 2D-S data was available from SPARTICUS. The 2D-S ice concentrations are overwhelmingly dominated by the 1st size bin (5–15 $\mu$m). Artifacts and uncertainties render the first bin or two of 2D-S measurements relatively useless. Most 2D-S data users do not use concentrations in the first two bins in their analyses because of the large uncertainties. I would recommend excluding the first two bins in the PSD comparisons shown in Figure 1 for temperature bins for which little or no ATTREX data is available. Also, I think it is inappropriate to use $N_i^{D>5\mu m}$ data from the SPARTICUS 2D-S-only dataset for evaluation of the satellite retrievals. The MACPEX 2D-S data should only be used for $N_i^{D>25\mu m}$ and $N_i^{D>100\mu m}$.

**5. Figure 1:** Indicate in figure or caption which temperature ranges correspond to

ATTREX data (mostly < -70°C) and SPARTICUS data (warmer temperature ranges).

**6. Figure 2:** The authors should note and discuss the D05 overestimate (by factor of 2–3) for small ($D$ <10 $\mu$m) particles in -80 to -70°C bin compared to ATTREX measurements.

**7. Figure 3:** The small sample volume of the FCDP instrument results in discretization of the ice number concentration in steps of about 12 $L^{-1}bin^{-1}$. In other words, the FCDP instrument cannot effectively measure ice concentrations smaller than about 10–20 $L^{-1}$ if the data is used at 1 Hz (as in this study). The CAS data has a similar sample volume issue. Since ice concentrations are often dominated by the small crystals sampled by FCDP and CAS, I would recommend not showing the in situ vs D05 comparisons for concentrations less than 10 $L^{-1}$.

In some of the temperature bins, the data extends to ice concentrations greater than 1000 $L^{-1}$. Extending the upper limit on the Figure 3 axes would be helpful to show how well the retrieval compares with in situ measurements at higher ice concentrations.

**8. Figure 3:** The authors should note that discrepancies up to factors of 2–3 occur but are difficult to see with the log-log axis scales.

**9.** Will the $N_i$ data be made publicly available? If so, data quality flags should be included to indicate when both radar and lidar signals are available as well as when the retrieval is questionable based on in situ comparisons?

**10. Page 14, lines 1-6:** I do not understand what the authors are saying here. I was under the impression that Figures 2 and 3 simply showed statistical comparisons between the in-situ-measured and retrieved PSDs and ice concentrations. The first paragraph of section 4.2 suggests the comparisons in section 4.1 were ideal cases. Perhaps this idealization should be explained and emphasized at the beginning of section 4.1.

**11. Section 4.2:** I am not convinced that the near-coincident in situ/satellite retrieval comparisons are useful given the enormous spatial/temporal variability in cloud properties and the corresponding need for very close time and space coincidences for meaningful comparisons. Not surprisingly, the scatter in the comparisons shown in Figure 4 is very large, spanning 1–2 orders of magnitude.

**12. Page 15, line 9-10:** In contrast to the statement here, the DARDAR-LIM retrieval overestimates $N_i^{D>5\mu m}$ and $N_i^{D>25\mu m}$ compared to SPARTICUS data even in the -60 to -50° C temperature bins.

**13. Figure 5:** The comparisons shown here are very difficult to see, particularly those for lidar-only and radar-only retrievals. The relative agreement between lidar-radar, radar-only, and lidar-only retrievals should be shown in a separate figure, particularly since the lidar-only and radar-only retrievals are suspect.

Also, as discussed above, the SPARTICUS 2D-S-only ice concentrations for D>5 $\mu$m are dominated by the first size bin, with enormous associated uncertainties. The comparisons with SPARTICUS 2D-S-only ice concentrations including the first bin are of little value, possibly misleading, and should be removed.

**14. Page 19, lines 8-10:** the lack of clear transitions in retrieved properties between the lidar-only, lidar-radar, and radar-only regions does not necessarily mean the lidar-only and radar-only $N_i$ retrievals are credible.

**15. Figure 8:** Scatter plots of $N_i^{D>5\mu m}$ and $N_i^{D>100\mu m}$ versus $N_{2D-S}$ would provide much clearer comparisons between the retrievals and measurements. Further, the points could be color coded to indicate whether they are in the lidar-only, lidar-radar, and radar-only regions. In the discussion of Figure 8, the authors claim good agreement between the in situ and retrieved ice concentrations, and they dismiss glaring discrepancies as being caused by the imperfect time coincidence. This argument seems unjustified. The flight track segment has been chosen for good time/space coincidence.

**16. Section 5.3:** The authors describe a cloud formation scenario with air parcels ascending across the -40° C isoline, which suggests that freezing of liquid drops could

be the main ice formation mechanism. Yet they attribute the differences between the high and low ice concentration regions to differences in vertical wind speed and cite the strong sensitivity of $N_i$ to $w$ (citing Krämer et al. 2016; papers showing this sensitivity decades earlier should be cited). However, the strong sensitivity to $w$ occurs primarily when aqueous aerosols freeze, not so much when liquid droplets freeze. Either the description is not clear, or the physical argument made does not make sense.

**17. Figures 9 and 10:** The discrepancy between $N_i^{D>5\mu m}$ and ATTREX FCDP ice concentrations noted above is apparent in the coldest temperature bins and the TTL. Typical values of $N_i^{D>5\mu m}$ are 200-300 L$^{-1}$, whereas ATTREX in situ measurements indicate ice concentrations of about 100 L$^{-1}$ (Jensen et al., 2016). It is also interesting that the ice concentrations are higher over continental and convective regions even in the coldest temperature bins (near the tropical tropopause) where the vast majority of clouds form in situ. Additionally, it might be worth noting that the statistics must be poor in the coldest temperature bin poleward of about 30° latitude since such cold temperatures rarely occur there.

**18. Page 22, line 7:** Simply stating that the spatial distributions agree with the global model predictions is no doubt too strong. A quick examination shows there are some regions of agreement and some glaring discrepancies. I would either omit this statement or qualify it. Perhaps the comparison really shouldn't be discussed without providing much more detail.

**19. Section 6.2:** Most of the speculations about the physical causes of the zonal-height distributions in this section are either not justified or would require much more detail to adequately discuss. It does not look to me like there is a particularly sharp transition at -40° C, nor would one be expected given the importance of sedimentation in cirrus. The retrieval probably doesn't work well in the antarctic wintertime stratosphere since PSCs are typically mixtures of ice crystals, NAT particles, and ternary aerosols.

---

## Referee Comment (RC4) · Anonymous Referee #1 · 20 Mar 2018

Referee #2 has stated that the first two size-bins (5-15 $\mu$m and 15-25 $\mu$m) of the 2D-S probe should not be used to compare against this DARDAR-LIM retrieval since these bins are not reliable (i.e. have too much uncertainty). It should be noted that the FSSP 100 or FSSP 300 sample particles in the size range 3.0–30 or 0.6–40 $\mu$m diameter, respectively, and that typically 90% of the total N (ice particle number concentration) is sampled over this size-range (Krämer et al., 2009, ACP). The smaller size-bins of the 2D-S probe generally measure higher ice particle number concentrations such that $N(D)1 > N(D)2$, where $N(D)1$ corresponds to N between 5-15 $\mu$m and $N(D)2$ corresponds to N between 15-25 $\mu$m. Even if future 2D-S probe improvements show that $N(D)1 < N(D)2$, the concentration $N(D)$ in these two bins is not zero, and $N(D)1$ plus

[Figure]

N(D)2 could easily be higher in number concentration than PSD defined by the larger size-bins of the 2D-S probe. Thus, throwing out the first two 2D-S bins is unlikely to improve the accuracy of 2D-S measurements of N. Furthermore, if the first two 2D-S bins are omitted, what would be the purpose of comparing DARDAR-LIM N retrievals with 2D-S measurements?

On the other hand, such comparisons may be meaningful. In collaboration with SPEC, Inc. (the company that invented the 2D-S probe), our group has analyzed comparisons of FSSP and 2D-S measurements of ice particle size distributions (PSDs) during the SPARTICUS field campaign. PSD sampling times ranged from 1.0 to 5.0 minutes, with a mean sampling time of $2.65 \pm 1.35$ minutes. Ten 2D-S/FSSP paired PSD comparisons were found where the agreement between these probes was relatively good (but far from perfect) for D > 15 $\mu$m; PSD were relatively narrow in 8 of the 10 pairs. The 2D-S ratio N(D)1/N(D)2 was estimated for cirrus clouds based on the FSSP bins that corresponded with 2D-S bins 1 and 2 [i.e. N(D)1 and N(D)2]. On average, this ratio was 0.42, ranging in value from 0.22 to 1.3. While the FSSP generally measured lower concentrations for N(D)1, the FSSP consistently measured higher (or roughly equal) concentrations for N(D)2, relative to the 2D-S probe. Based on this analysis, N(D)2 is not over predicted by the 2D-S probe.

There are uncertainties and limiting factors associated with all of the ice PSD probes, and it can be argued that agreement in the 5-to-35 $\mu$m range cannot be expected to be better than a factor of two, partly due to the small sample volumes and their uncertainties at these sizes (e.g. Cotton et al., 2013, QJRMS). Perhaps expecting satellite retrievals of N to agree with aircraft probe measurements of N within a factor of two is misguided (e.g. comment 17 from Referee #2), given the probe and retrieval uncertainties. Satellite retrievals of N arguably have more uncertainties than the in situ measurements do, and in this referee's opinion, their value is not in capturing the absolute magnitude of N, but it is in revealing how N varies as a function of latitude, season, surface type, etc. Much can be learned from these relative differences in N, and while

it is optimal when retrieved N agrees closely with in situ N, the scientific community should not be denied valuable information concerning these relative differences when differences between retrieved and in situ N exceed standards applied to probe performance.

---

## Author Comment (AC1) · 17 Jul 2018

Response to referee #3 (in RC1)

RC: The article presents a method to be used as an operational retrieval to derive ice crystal number concentrations of pure ice clouds, Ni, (T < -30° C) from combined spaceborne lidar-radar measurements (CALIPSO-CloudSat) and a thorough evaluation using in situ data from five airborne campaigns. An example of application is shown via a case study, including Lagrangian transport modelling. An interesting result is that regions with stronger updraughts show peaks in Ni with particle sizes > 5micron in contrast to regions of mature cloud, as one would expect. At the end, geographical maps and zonal profiles of 10 years of Ni are presented and discussed for particles with sizes > 5micron and > 100micron. A follow-up paper will use these data in the framework of aerosol-cloud interactions. Ni is an essential microphysical parameter, which is recently used as a prognostic variable in climate models, and therefore it is important to have global observational constraints. The variable is also important for process studies. The combination of lidar and radar measurements, being part of the A-Train, allows to determine the vertical structure of clouds such as top and base of the clouds, cloud layering, as well as ice water content and effective ice crystal diameter. The attempt to derive ice crystal number concentration is relatively recent, as its determination depends on several assumptions (in particular a gamma-modified particle size distribution (PSD) and a specific ice crystal mass-maximal diameter relationship). The presented method is based on a direct constraint of the shape of normalized particle size distributions using lidar extinction and radar reflectivity from the operational liDAR-raDAR (DARDAR) products. 40000 in situ PSD's are used for an evaluation, investigating results separately for ice crystal sizes > 5, 25 and 100 micron, first for the prediction of PSD from N0* and Dm and then for retrieved Ni. The article is generally well structured and well written. I strongly recommend the publication of this article, after minor revisions.

AR: We thank the referee for all the insightful comments that have greatly helped us to improve the quality of the manuscript. Detailed responses to each of them are provided below.
* * *
Minor Comments

1. RC: 1) The methodology section 2 gains in clarity by integrating the content of section 2.1 into sections 2.3 and section 3.1, in particular as DARDAR products are data and the retrieved variables such as beta-ext, Ze and beta-ext are not defined. In that way the section on the representation of the size distribution gets section 2.1, in which the advantage of using scaled PSD's is described as well as the necessity to assume a certain m-D relationship and a certain shape of PSD. The new section 2.2 (Extracting Ni from DARDAR products) goes then further into detail how to extract Ni from the DARDAR products $N_0^*$ and IWC. It should be clearly stated in the beginning that from $N_0^*$ and IWC from DARDAR one deduces Dm and finally Ni. New Section 2.1: Be careful of replacing DARDAR by 'DARDAR retrieval (see section 3.1)'. Then a short description of the DARDAR products (like in initial sect 2.1) should be integrated into section 3.1. P4,l17-18 define beta-ext as (lidar) extinction and Ze as (radar) reflectivity

AR: We are grateful for these suggestions and fully agree with them, sections 2 and 3 have been edited accordingly. Sec. 2 now only focuses on describing the methodology and the DARDAR algorithm is described in Sec. 3.1. It can be noted that further technical details on DARDAR are now also provided in Appendix A.

[Figure]

Figure 1: (a) Spatial distribution of the rejection rate associated with the $n_{iter} < 2$ filtering for pure ice clouds with $T_c < $ -30°C. These results correspond to one year of DARDAR retrievals (2008). (b-c) show the corresponding ice water path (IWP) and average number of ice cloud pixels in the vertical column (we recall that the height of a pixel is 60 m). (d) represents the relative difference on $N_i^{5\,\mu m}$ between -60 and -50°C that would be expected if the $n_{iter}$ filtering was not applied.)
* * *
2. RC: 2) p 6, l22 it is stated that DARDAR retrievals of pure ice clouds for which the iterative retrieval converged too quickly are ignored. How many of these retrievals are these and can you explain which category of cases these are?

AR: We thank the referee for this comment as we had not yet looked into the distributions of rejection rates associated with the filtering based on iteration number. We agree that useful information could be contained there. It is reminded that this filtering is used to avoid pixels associated with a too quick convergence of the forward model with the observations, which could indicate a lack of information and therefore a strong reliance on a priori considerations. This is now further discussed in Appendix A of the revised manuscript.

The spatial distribution of this rejection rate for ice clouds with $T_c < $ -30°C is shown in Fig. 1 of this response. A strong latitudinal dependence of the rejection rate is noted, with less than 10% in the mid-latitudes and about 10 to 20% in the tropics. Rejection rates up to 40% are even seen in the north of oceanic subsidence regions of the South hemisphere. A high rejection rate in DARDAR retrievals in the tropics is not surprising as thick clouds with a complex microphysics are likely to be encountered there. However, Fig. 1(b-c) show that the highest rejection rates occur in regions where thin ice clouds with low IWPs are found, most likely retrieved from lidar-only conditions. It could therefore be that, for these thin clouds, a single iteration is sufficient for proper retrievals and we may be over-constraining the dataset filtering. This has never been investigated from DARDAR and would require further analyses to be verified and fully understood. We have nevertheless verified that this filtering actually has a small impact on the overall climatologies. Fig. 1(d) for instance shows the spatial distribution

[Figure]

Figure 2: Similar to Fig. 4 of the revised manuscript. The PSDs have here been subsetted following the classification proposed for SPARTICUS by Jackson et al. [2015]. PSDs for synoptic and convective clouds are shown on the left and right panels, respectively.

of relative differences in $N_i^{5\,\mu m}$ (in the -60 to -50°C bin) between 1-year climatologies obtained without and with applying the $n_{iter}$ filtering. Differences smaller than 10% are typically found in regions where the rejection rate is the most significant. The bias is positive, which seems to indicate that thin cirrus higher $N_i^{5\,\mu m}$ are ignored because of this filtering. It should be noted when comparing these results to Fig. 7(a-c) of the revised manuscript that relatively very low $N_i^{5\,\mu m}$ values are found in regions where this bias is maximum. The $n_{iter}$ filtering therefore does not have any significant influence on the results shown in this study. After careful consideration, we have chosen to keep the filtering as but we will keep in mind these analyses and results when producing future versions of the dataset (based on the next version of DARDAR cloud and mask products, which should soon be available). It can also be mentioned that all these filtering options will be provided together with the $N_i$ dataset, which will hopefully be distributed co-jointly with the publication of this two-part study.
* * *
3. RC: 3) The evaluation of the prediction of PSD's and Ni (using all field campaigns) and later for retrieved Ni (using coincident SPARTICUS measurements) is shown separately for different temperature intervals, which is important as ice crystal particles shapes differ with temperature. It would be very interesting to separate also anvils and synoptic cirrus, as m-D relations might be different. Is there enough statistics of the collocated SPARTICUS campaign measurements to compare Ni distributions of Fig. 5 for anvils and synoptic cirrus?

AR: We thank the referee for this interesting comment. It is a very good point that m-D relations might be different from different cloud types and this could subsequently affect the quality of our evaluation. As mentioned in Sec. 4.1.1, differentiating between different cloud types has not been attempted in this study for reasons of brevity and also because DARDAR does not make any distinction when assuming its PSD shape and m-D relation. It would nevertheless

be interesting, following the referee's comment, to indeed verify if any specific issue occur when applying a basic differentiation, such as convective vs. synoptic clouds.

To do this, we have associated a cloud type to each PSD from the SPARTICUS dataset used in this study, based on the cloud classifications by Muhlbauer et al. [2014] and Jackson et al. [2015]. Fig. 2 of this response shows comparisons between the histograms of collocated SPARTICUS measurements (Fig. 5 in the original manuscript, Fig. 4 in the revised version) when distinguishing between the "convective" and "synoptic" classification by Jackson et al. [2015]. This classification is chosen here as it is more straightforward. Muhlbauer et al. [2014] offer numerous specific cloud classes, which for this application leads to subsets with a lower statistical significance. It can first be observed in Fig. 2 that convective clouds have higher $N_i$ means, but are also much less occurrent than synoptic clouds during SPARTICUS. With respect to the quality of DARDAR-LIM retrievals no obvious bias or other issue can be noted in either cloud class. Differences are noted but it remains difficult to estimate if these are within the noise, considering the small number of statistics. Testing the impact of m-D relations would also require to disentangle the impact of a possible misrepresentation of the PSD shape in either of these two cloud classes. Finally, it should be kept in mind that these cloud classification are often very difficult to obtain and can be associated with large uncertainties as well.

For these reasons, and to avoid substantial additional descriptions and discussions in the manuscript, we have still kept analyses based on cloud types out of the revised manuscript. But we recognize the importance of this point and the strong interest to differentiate between cloud types to test the impact of the m-D relation but also the assumptions made on the PSD shape. This will be done in a future study that will focus on improving the PSD representation used for lidar-radar $N_i$ retrievals.
* * *
4. RC: 4) section 3.2.2: One specific ice crystal mass-maximum diameter (m-D) relationship is used to determine IWC from the PSD. Indeed, Delanoë et al. 2014 show that the uncertainty to the m-D relationship for the normalized PSD is less important when minimizing using lidar extinction and radar reflectivity. The uncertainty seems to increase if only the lidar extinction is used for the minimization (Fig. 9). As both measurements are complementary, there are clouds for which only the first (thin cirrus) or the latter (towards the base of thick cirrus) are available. We also know that the shape of crystals changes with temperature and Heymsfield et al. 2010 showed that the m-D relation for anvil ice clouds yield masses about a factor of 2 larger than for synoptic ice clouds. Erfani and Mitchell 2016 cite this result in their paper and write that their results showing a similarity in m-D expressions between these two cloud types might be an artefact if the ice particle masses for a given projected area are quite different between these types. The L16 m-D relationship was developed for midlatitude cirrus. So for tropical anvils the computed IWC might be biased. Did you test the IWC computed with the L16 m-D relationship with the measured IWC for tropical anvils (using SPARTICUS and ATTREX) ?

AR: We again thank the referee for this very good point. It is absolutely correct that, as shown by Delanoë et al. [2014] and mentioned by the referee here, uncertainties related to the m-D relation used on the normalized PSD are minimum when both lidar and radar are available. This should lead to smaller uncertainties on lidar-radar $N_i$ estimates, as now discussed in the appendix of the revised manuscript.

The consequences on $N_i$ could also be evaluated using the histograms for coincident flights shown in Fig. 2. However, it can be argued that the statistics are for the moment not sufficient

[Figure]

Figure 3: SPARTICUS IWC obtained from Nevzerov (first row) and 2D-S (second row) measurements, as function of L16 predictions based on the 2D-S PSD. The column indicate the cloud category based on Jackson et al. [2015]. A factor of 3 around the one-to-one line is indicated by a dashed line.

to draw any strong conclusions. The impact of m-D assumption will also need to be disentangle from the impact of the PSD shape assumptions, which largely dominate the observed differences. We nevertheless hope to extend this type of evaluation using additional flights coincident with the A-Train, in order to further dig into these issues in the future.

Regarding the use of L16, we have not performed comparison to SPARTICUS or ATTREX measurements in the context of this study, but evaluations of this m-D relation have been made in other studies. Afchine et al. [2018] has for instance shown that this relation should be applicable to tropical clouds, and that the influence of different m-D relations on IWC is small in the temperature range of cirrus. This is now further detailed in Sec. 3.2.2. To provide a more complete response to this comment, we have now analysed the consistency between L16 and SPARTICUS measurements. The classification by Jackson et al. [2015], discussed in the previous response, is used to differentiate between synoptic and convective clouds. IWCs are operationally provided from the 2D-S [based on an assumed area-mass relation; Baker and Lawson, 2006] as well as from bulk measurements from a Nevzerov probe. These comparisons are shown in Fig. 3. It appears that L16 overestimates by a factor of about 2 the IWC measured by the Nevzerov probe. This overestimation seems consistent between synoptic and convective clouds. The 2D-S IWC are in better agreement with L16, for either the synoptic of convective clouds. These results based on SPARTICUS are therefore in agreement with the conclusions by Afchine et al. [2018]. Unfortunately, the ATTREX IWC was not available in the version of the data used for this study and the same analysis could not be repeated. However, it can be noted that Thornberry et al. [2017] showed similarly good agreements between the 2D-S-based IWC and bulk measurements.

5. RC: 5) Figs. 6 c and d of the case study present the trajectories as function of UTC. The relevant variable is the time difference which you show in brackets, and then the position on the map in Fig. 6a. If it is not too complicated, it might be clearer to present instead of UTC longitude.

AR: We thank the referee for this comment, adding the spatial coordinates would indeed add clarity to compare Fig. 6(c-d) to Fig. 6a (Fig. 5 in the revised manuscript). We have changed these figures so that the time difference is now used as the reference variable and the corresponding lat-lon coordinates for trajectories A and B are indicated in brakets.

———————————

6. RC: 6) concerning Fig. 5, is it possible to get also De from DARDAR for this cloud ?

AR: This is a good point, DARDAR $r_{eff}$ retrievals are now added in Fig. 6(d) of the revised manuscript and are briefly described in Sec. 5.2.

———————————

7. RC: 7) The long descriptive text of the case study is sometimes difficult to follow. I suggest for example to move the analysis of the collocated air track comparison (Fig. 8) to a supplement.

AR: We fully agree with this comment, especially considering that the paper already is long and that thorough in situ analyses have extensively been discussed in the previous section. This figure aimed at comforting these results and show that DARDAR-LIM is capable of reproducing the spatial variability of $N_i$ observed by the 2D-S. Following this comment, it has been moved to supplementary materials (see Fig. S7) and the discussion in Sec. 5.2 has been shortened accordingly.

———————————

8. RC: 8) I would rename section 6 'Presentation of global Ni climatologies' and 6.1 'Geographical distributions'. P21l5: 'considered with caution' instead of 'cautiously considered'

AR: We thank the referee for this comment, Sec. 6 has been edited accordingly.

**References:**

A. Afchine, C. Rolf, A. Costa, N. Spelten, M. Riese, B. Buchholz, V. Ebert, R. Heller, S. Kaufmann, A. Minikin, C. Voigt, M. Zöger, J. Smith, P. Lawson, A. Lykov, S. Khaykin, and M. Krämer. Ice particle sampling from aircraft – influence of the probing position on the ice water content. *Atmos. Meas. Tech.*, 11(7):4015–4031, 2018. doi: 10.5194/amt-11-4015-2018.

B. Baker and R. P. Lawson. Improvement in determination of ice water content from two-dimensional particle imagery. part i: Image-to-mass relationships. *J. Appl. Meteor. and Clim.*, 45(9):1282–1290, 2006. doi: 10.1175/JAM2398.1.

J. Delanoë, A. J. Heymsfield, A. Protat, A. Bansemer, and R. J. Hogan. Normalized particle size distribution for remote sensing application. *J. Geophys. Res*, 119(7):4204–4227, 2014. doi: 10.1002/2013JD020700.

R. C. Jackson, G. M. McFarquhar, A. M. Fridlind, and R. Atlas. The dependence of cirrus gamma size distributions expressed as volumes in n0-$\lambda$-$\mu$ phase space and bulk cloud properties on environmental conditions: Results from the small ice particles in cirrus experiment (sparticus). *Journal of Geophysical Research: Atmospheres*, 120(19):10,351–10,377, 2015. doi: 10.1002/2015JD023492.

A. Muhlbauer, T. P. Ackerman, J. M. Comstock, G. S. Diskin, S. M. Evans, R. P. Lawson, and R. T. Marchand. Impact of large-scale dynamics on the microphysical properties of midlatitude cirrus. *Journal of Geophysical Research: Atmospheres*, 119(7):3976–3996, 2014. doi: 10.1002/2013JD020035.

T. D. Thornberry, A. W. Rollins, M. A. Avery, S. Woods, R. P. Lawson, T. V. Bui, and R.-S. Gao. Ice water content-extinction relationships and effective diameter for ttl cirrus derived from in situ measurements during attrex 2014. *J. Geophys. Res. Atmos.*, 122(8):4494–4507, 2017. doi: 10.1002/2016JD025948.

---

## Author Comment (AC2) · 17 Jul 2018

Response to referee #1 (in RC2)

RC: This paper presents a new method for retrieving the ice particle number concentration $N_i$ for glaciated clouds, which should be useful for understanding aerosol interactions with ice clouds and the contribution of homogeneous vs. heterogeneous ice nucleation in cirrus clouds. A satellite remote sensing scheme for $N_i$ is needed since field campaigns cannot adequately inform us how $N_i$ varies with latitude and the seasons. The paper is well organized and well written, and usually cites the relevant literature. The quality of the figures is good. The methods developed in Sec. 5 for testing the retrieval are especially creative and effective.

AR: We are thankful to the referee for all the useful comments that greatly helped us to improve the quality and clarify of this study. In particular concerning the influence of the PSD shape assumptions, the use of 2D-S data and the consistency between the analyses of $N_i$ climatologies presented in the 2 papers of this study. Detailed responses to each comment are provided below.

—————————————

RC: A critical limitation of the retrieval algorithm is the use of a normalized universal ice particle size distribution, or PSD (Delanoë et al., 2005), where it is assumed that all PSD in nature conform to this normalized PSD shape. This normalized PSD is based on a four-parameter gamma function (Eq. 4) where parameters No and k can be deduced through their link with other operationally retrieved properties (IWC and $N_0^*$) while PSD parameters $\alpha$ and $\beta$ need to be fixed as constants. This is of little consequence regarding $\beta$, which affects the largest ice particles having the lowest concentrations. But this is of major consequence regarding $\alpha$, which strongly influences the smallest ice particles that govern $N_i$. This is not mentioned in the paper. The small end of the PSD is sensitive to the rate of ice nucleation which is sensitive to the cloud updraft w (with higher w making $\alpha$ more negative, and $N_i$ higher), as well as the aggregation rate that removes smaller ice particles having higher concentration (Herzegh and Hobbs, 1985, QJRMS; Mitchell, 1991, JAS). Thus, some discussion on this topic is warranted, especially on the errors that may result from "non-standard" conditions where atypical updrafts are common (such as over steep orography).

AR: We thank the referee for pointing out the need for further discussion regarding the impact of non-retrieved shape parameters of the size distributions ($\alpha$ and $\beta$). We completely agree that this was lacking in the original manuscript.

DARDAR unfortunately does not rigorously account for these uncertainties in its operational retrievals, as they are only represented by additional fixed errors considered on the lidar and radar measurements. More rigorous techniques exist to propagate uncertainties on $\alpha$ and $\beta$ through the optimal estimation scheme but they would be too time consuming for an operational algorithm based on active instruments. However, the variability of these two parameters and the subsequent impact on DARDAR has been thoroughly discussed in Delanoë et al. [2014]. It can be noted that, as a result of this study, a revised version of the PSD parameterization has been proposed (notably with a less negative $\alpha$, leading to less small ice crystals and a lower $N_i$) but is not yet implemented in the operational product. The referee is therefore absolutely correct in saying that the fixed $\alpha$ and $\beta$ parameters constitute a strong limitation to our current method that should be further highlighted. These points are now discussed in Sec. 3.1 and in Appendix

A3 of the revised manuscript and are supported by additional figures in the supplementary materials (see Fig. S3).

The impact of the choice of $\alpha$ and $\beta$ on the PSD shape is clearly shown in the upper panel of Fig. S3, and the subsequent impact on $N_i$ because of straying from the selected values ($\alpha = -3$ and $\beta = -1$) is quantified in the lower figure. In order to propose a range of realistic shape parameters, values extracted by Delanoë et al. [2014] from individual in situ campaigns are used (color code in upper figure and shapes in the lower figure). IWC and $N_0^*$ values representative of 3 temperature bins are selected, although it should be kept in mind that each couple of coefficients from the D14 campaigns can realistically applied to only one of these temperature ranges. In agreement with the referee's comment, it can be observed that one D14 campaign displays a more negative $\alpha$, namely the "subvisible" campaign, which corresponds to cirrus measured at temperatures between -80 to -60°C during CRYSTAL-FACE (Cirrus Regional Study of Tropical Anvils and Cirrus-Layers-Florida Area Cirrus Experiment). We recognize that this analysis remains preliminary but it should still allow to provide rough estimates of the uncertainties on $N_i^{5\,\mu m}$, $N_i^{25\,\mu m}$ and $N_i^{100\,\mu m}$ to the reader. This overall uncertainty is here considered to be typically better than about 50% (when considering the variability between all D14 campaigns). This value is now reported in Sec. 3.1 and A3 of the revised manuscript.

––––––––––––––––

RC: The lead author gave a nice talk about this retrieval at the A-Train Symposium in 2017. Henceforth, Ni refers to Ni for ice particle maximum dimension $D > 5\mu m$. Slide 20 of this presentation, showing global distributions of Ni for 10°C intervals, appears almost identical to Fig. 9 of this paper for T < -30°C, except that the Ni legends differ. The Ni values reported in the presentation are higher by a factor of about 1.7 relative to the Ni reported in Fig. 9 of this paper. What is the reason for this difference?

AR: We are grateful that the referee took the time to verify the consistency between this paper and the results presented during the A-Train Symposium. The figure referred to here (slide 20 of this presentation, available on http://atrain2017.org), corresponded to the $N_i$ integrated from $D_{min} = 1\mu m$. This may not have been clearly expressed during the presentation but is suggested by the absence of mention to the size in the label. The $1\,\mu m$ threshold was initially used at early stages of our analyses, but was subsequently changed to $5\,\mu m$ as it is impossible at this point to reasonably evaluate DARDAR-LIM between 1 and $5\,\mu m$. Also, as discussed before, uncertainties related to PSD shape assumptions are likely to be even more important if $D_{min} = 1\,\mu m$.

Fig. 1 shown in this response corresponds to the distribution of $N_i^{1\mu m}$ based on the dataset used for this paper. It can be noted that similar values to those shown during the A-Train Symposium presentation are found, despite small differences in absolute values. These could be due to an error found in the script that converts $D_{min}$ from maximum diameter to an equivalent melted size prior to the numerical integration of the PSD, which led to slightly underestimated $D_{min}$ and thus to higher concentration. This error was corrected before creating the dataset used in these papers. To the best of the first author's knowledge, there should be no other difference between Fig. 1 of this response and the figure in the A-Train Symposium presentation.

––––––––––––––––

Major Comments

[Figure]

Figure 1: Spatial distribution of $N_i^{1\mu m}$ from 2006 to 2016, averaged in a $2 \times 2°$ lat-lon grid and per $10°$ C temperature bin from -80 to -30° C.

1. RC: Page 8, line 25: The 2DS photodiode array length is $1280\,\mu m$, which should be noted. Evidently the "time dimension" is used to size particles up to $3205\,\mu m$; please indicate the particle selection criteria used to size and count particles.

AR: We are very thankful to the referee for this comment that has led us to investigate in greater detail the various selection criteria for particle size and count that are available for the 2D-S instrument.

In the original manuscript, ATTREX-2014 data was processed with the method $M_1$, or $M_7$ method when available. There are important differences between these methods, in particular concerning the size selection, which are for instance extensively described and discussed in Lawson [2011] and Erfani and Mitchell [2016]. The SPARTICUS data was treated with the $M_1$ method only, as $M_7$ isn't operationally available in the ARM database. Comparing concentrations from these 2 methods should not be an issue as Erfani and Mitchell [2016] showed that the number concentration in small particle isn't significantly different between them.

Nevertheless, after further discussion with the 2D-S data providers at SPEC Inc. (P. Lawson and S. Woods), it appeared that using a SPARTICUS dataset based on a $M_4/M_1$ processing could be better adapted to the needs of this study. By $M_4/M_1$ it is meant that the $M_4$ method is used for particles sizes less than $365\,\mu m$ and the $M_1$ is used otherwise. A main differences between these two methods is that $M_4$ resizes out-of-focus particles to equivalent in-focus spheres [Korolev, 2007]. This becomes problematic when the ice particle shapes become strongly non-spherical, and this method can therefore only be applied to small particles. Consequently, it was decided with the SPARTICUS 2D-S data providers that a combined $M_4/M_1$ processing method should be used here.

The differences between the PSDs obtained from $M_1$ alone and $M_4/M_1$ are shown in Fig. 2 of this document. The main difference occur for sizes between about 30 to 100-200 $\mu m$, with typically more particles with $D < 100\mu m$ and less particles larger than this threshold. As a consequence, the bi-modal structure is less pronounced in $M_4/M_1$, but it also is clear from this figure that the results discussed in the original manuscript are not changed by this transition from $M_1$ to $M_4/M_1$ 2D-S data. It can also be mentioned that slightly less flights with $M_4/M_1$ treatments were available on the ARM database.

As a response to this comment, Sec. 3.2.1 was edited to explicitly mention the use of the $M_n$ methods.

Regarding the photodiode specifications, it seems that the 2DS photodiode array length is (if referring to the actual physical size of the array) of about $7.3\,mm$ [Lawson et al., 2006]. However, we fully agree with the referee that, because (i) the equivalent size of each photodiode

[Figure]

Figure 2: Comparison between the SPARTICUS 2D-S PSDs obtained from the M4/M1 (top; as in Fig. 1 of the revised manuscript) and the M1 method (bottom; as in Fig. 1 of original manuscript.)

is about $10\,\mu$m (considering the laser beam magnification) and (ii) the 2D-S being equipped with 128 photodiode, this instrument technically measures particules up to $1280\,\mu$m in size and so an extension to $3205\,\mu$m is only possible by using a time dimension (i.e., by using 2 consecutive measurements a of $1280\,\mu$m particle). This is now mentioned in Sec. 3.2.1 of the revised manuscript.

––––––––––––––

2. RC: Figure 5 and Sec. 4.2: For T > -50C, by what factor is Ni ($D_{min} = 5\mu$m) overestimated, on average? For T $\geq$ -50C?

AR: Based on Fig. 5 of the original manuscript, an overestimation of $N_i^{5\,\mu m}$ by a factor of about 2 to 3 can be considered if looking at the distance between the modes of the 2D-S and DARDAR-LIM distributions. We nevertheless agree that this figure did not provide an easy way to clearly quantify the bias, and visually comparing the modes does not really provide a real statistical estimate of the differences between DARDAR-LIM and the 2D-S. This figure has therefore been edited in order to include the geometric means associated with each histogram (DARDAR-LIM, D05 and 2D-S) for each temperature bin and instrumental condition. These should allow for a more quantitative discussion of the biases, included in the revised Sec. 4.2. For instance, overestimations by about 10 to 30% and 20 to 60% are found in the mean values of $N_i^{25\,\mu m}$ by D05 and DARDAR-LIM, respectively.

Complementarily, a line showing a factor of 3 around to one-to-one line has also been added to Fig. 4 of the original manuscript, now Fig. S4 of the complementary materials.

[Figure]

Figure 3: Left: Conditional density of $N_i^{5\,\mu m}$ as function of the temperature, obtained from the insitu data used in this paper (bottom) and the corresponding D05 predictions (top). Plain red lines indicate the median and dashed lines show the 10th and 90th percentiles. The right panel directly compares the medians and 10th and 90th percentile lines.

––––––––––––––

3. RC: Page 21, lines 9-12: The strong temperature dependence of Ni mentioned here appears at variance with the in situ measurements reported in Krämer et al. (2009). Please mention this.

AR: We thank the referee for this comment, which has encouraged us to further compare our $N_i$ products with the insitu findings in several studies by Krämer et al.

It should first be mentioned that it is very difficult to compare the temperature dependence of $N_i$ obtained from in situ campaigns to those from global results shown in Sec. 6. In situ measurements are rather sparse and it is often difficult to tell what part of the cloud has been sampled. However, the enormous advantage of the dataset by Krämer et al is indeed that it consistently merges numerous in situ campaigns and should therefore tend to being comparable to global satellite data. This dataset is still being improved as airborne campaigns are continuously being added. The Ni(T) relation reported in Krämer et al. [2009] was based on a dataset that was not yet very large and contains some flights in mountain wave clouds that enhanced the frequencies of higher ice concentrations. A new, yet unpublished, dataset called JULIA does not confirm the (slight) dependence of Ni on T shown in Krämer et al. [2009]

[Figure]

Figure 4: Same as 3 but for DARDAR-LIM $N_i^{5\,\mu m}$ compared to co-incident in situ observations during SPARTICUS.

(personal communication, M. Krämer).

This comment has motivated us to compare the Ni(T) obtained in this study with the one from JULIA. Due to the complexity of this task, intermediate steps were taken. First, we have verified that the issues noted with D05 (notably due to its limited shape assumptions) does not create clear biases in the Ni(T) relation. This is shown in Fig. 3 attached to this response, which compares the Ni(T) dependency obtained from the dataset used in this study to that predicted by D05 based on the in situ data. It can be noted that the consistency of the relation found from our dataset to the one from JULIA has been verified, although this cannot be directly demonstrated here due to the latter being unpublished. Fig. 3 clearly shows that D05 is very well capable of reproducing the relation between $N_i^{5\,\mu m}$ and $T_c$ found in the in situ measurements, and so similar results could be expected from DARDAR. This has been verified by looking at the same relations based on the co-incident SPARTICUS flights. Fig. 4 in this response shows that DARDAR-LIM reproduces well the Ni(T) relation observed by the 2D-S. We have checked that these results also hold for $N_i^{25\,\mu m}$ and that they are not sensitive to instrumental conditions.

Consequently, it could be expected that Ni(T) obtained from global DARDAR-LIM estimates are reasonable and that the observations from Sec. 6 are not necessarily at variance with in situ observations. However, the results presented here are preliminary and further analyses are necessary to confirm them. For instance, it would require to subset similar regions, cloud

type or distance from cloud top by comparison the in situ data. Rigorously assessing the consistency between $N_i(T)$ observed from DARDAR-LIM and from in situ measurement would be extremely interesting but unfortunately out of the scope of this paper. These results will be the focus in a following study.

––––––––––––––

4. RC: Figure 9 and Sec. 6.1: For T > -50°C, Ni tends to be lower over regions characterized by extensive marine stratus, like off the west coasts of South America and Africa (from equator and southwards). Is this result real, or is it an artefact of the retrieval? If the latter is true, please explain.

AR: We thank the referee for pointing this out. It is correct that $N_i$ (for all integration thresholds) tend to be relatively lower in marine stratocumulus regions. There does not seem to be any obvious reason to doubt the retrieval method in these regions but it should indeed be kept in mind that there are relatively less ice clouds in these subsidence regions. The spatial distributions of retrieval counts have now been added to supplementary materials (see Fig. S8) to help determining which regions correspond to statistically significant retrievals. Another physical explanation could be that there are no convective clouds in these regions, which seem to drive the high $N_i^{5\,\mu m}$ and $N_i^{25\,\mu m}$ observed in this figure. This is supported by the seasonal variabilities in $N_i$ maps shown in Fig. 8 of the revised manuscript. Consequently, values observed correspond to thin cirrus, perhaps remnants of aged anvils or jet stream cirrus, and $N_i^{5\,\mu m}$ values below $100\,\mathrm{L}^{-1}$ for T > -50°C are thus not surprising, as mentioned in comment #7 of this review. It also means that $N_i^{5\,\mu m}$ in this regions are more comparable to cloud-top values observed in the part 2 paper. This is now noted in Sec. 6.1 of the revised manuscript.

––––––––––––––

5. RC: Page 21, lines 14-19: A similar finding was reported in Mitchell et al. (2016, ACPD), where the highest Ni were associated with mountainous terrain. (Although this paper was rejected since the editor felt the retrieved Ni values were too high, and therefore could not be used to infer nucleation modes, no arguments cast doubt on the spatial and temporal relative differences in Ni, which still appear meaningful.)

AR: We fully agree that further comparisons to existing climatologies would be beneficial to the analyses in Sec. 6.1. A new paragraph discussing comparisons results by Mitchell et al. [2016, 2018] is now included.

––––––––––––––

6. RC: Page 22, lines 7-9: It is more meaningful to compare model results against observations than vice-versa. Suggest removing this paragraph. For example, in the modeling study by Zhou et al. (2016, ACP), the sensitivity of homo- and heterogeneous ice nucleation to various model parameters and updraft schemes were evaluated. Depending on how these are represented, one can get a broad range of Ni-temperature dependences, including Ni that is relatively insensitive to temperature (similar to the in situ observations of Krämer et al., 2009, ACP), and that modeling result would not support these DARDAR-LIM findings.

AR: We agree with this comment, comparisons to modeling would require further analyses that are not in the scope of this paper. This paragraph is now removed
* * *
7. RC: Figure 10 and Sec. 6.2: $N_i$ ($D_{min}$= 5 um) in the tropics appears contrary to the $N_i$ results in Fig. 1 and Fig. 5 of Part 2 of this study by Gryspeerdt et al. (submitted). Fig. 1a of Gryspeerdt et al. show Ni near cloud top while their Fig. 5 shows that $N_i$ does not change appreciably with distance below cloud top (up to $3\,km$ from cloud top) between -50 and -60°C. Assuming this result extends to other temperatures, the cloud top results in Fig. 1a of Gryspeerdt et al. should also be approximately valid below cloud top. Regarding Fig. 1a in Gryspeerdt et al., for $T > $-65°C, $N_i$ is never higher in the tropics relative to the midlatitudes. Between -55 and -40°C, where the most optically thick cirrus clouds exist (cirrus defined as clouds having $T < $-38°C), $N_i$ in the tropics is substantially lower than $N_i$ in the midlatitudes. In Fig. 10 of Part 1 (Sourdeval et al.), Ni increases abruptly in the tropics for $T < $-40°C (shown by the dashed curve), with Ni here being typically higher than Ni at similar T in the midlatitudes. This result appears opposite to the findings in Fig. 1a of Gryspeerdt et al. (Part 2). In addition, the CALIPSO Ni retrievals of Mitchell et al. (2016, ACPD) qualitatively support the findings of Gryspeerdt et al. (in terms of relative differences), and the in situ measurements from Mühlbauer et al. (2014) show relatively lower "peak Ni" values in anvil cirrus (vs. frontal, jet stream and ridge-crest cirrus). Finally, several studies (e.g. Jensen et al., 2013, PNAS; Spichtinger and Krämer, 2013, ACP), show that tropical tropopause layer (TTL) cirrus tend to have $Ni < 30\,L^{-1}$. Since the areal coverage of TTL cirrus exceeds that of anvil cirrus, and TTL cirrus tend to be higher than anvil cirrus (Gasparini et al., 2017, J. Climate), the Ni of $200\,L^{-1}$ in the TTL region in Fig. 10 appears at variance with in situ observations. Please comment on, and, if possible, reconcile these issues.

AR: We thank the referee for this interesting comment. It has motivated us to further compare the spatial distributions obtained from cloud-top $N_i$ ($N_{i(top)}$) (part 2) vs. the "all cloud" maps (part 1).

It is first important to point out that this is not straightforward as these two maps are not necessarily representative of the same cloud types within a given temperature bin. For instance, the $N_{i(top)}$ map between -50 and -60°C (in part 2) only shows concentrations for clouds that have a cloud-top within this temperature bin, whereas the total $N_i$ map (in part 1) also features values that are within deep convective clouds. It is observed that the high values of $N_i^{5\,\mu m}$ and $N_i^{25\,\mu m}$ only appear in convective regions, which is confirmed by the seasonal variabilities showed in Fig. 8. The sampling difference is also clear when comparing retrieval counts between $N_{i(top)}$ and $N_i$ per $T_c$ bin, now showed in Fig. 1 of the revised part 2 paper and in Fig. S8 of the revised part 1 paper, respectively. Nearly no retrievals are present in the tropic for the $N_{i(top)}$ map, whereas convective clouds are present in the $N_i$ map. To support this analysis, it can be noted that high $N_i$ values found between -50 and -60°C within deep convective clouds is in agreement with modeling results by Paukert et al. [2017], who also reports $N_{i(top)}$ lower than $N_i$ for this cloud type.

It could as well be argued that the CALIPSO $N_i$ retrievals presented in Mitchell et al. [2016, 2018] are also more comparable to the $N_{i(top)}$ map as the thermal infrared measurements used in these studies extinguishes within a few optical depth. It is therefore reasonable to expect that retrievals from these studies would not compare exactly to $N_i$ maps presented in part 1 but more to the $N_{i(top)}$ maps presented in part 2, as it is the case (in terms of relative variations of $N_i$).

Regarding the absolute values of $N_i$, we completely agree with the referee that the ones

presented our maps may not be completely exact. An overestimation by a factor of 2, or even 3, could be expected on $N_i^{5\,\mu m}$ considering all uncertainties on the retrievals (especially concerning the assumptions on the PSD shape). These uncertainties should be smaller on $N_i^{25\,\mu m}$, as the impact of the shape is less significant, and the spatial distributions of $N_i^{25\,\mu m}$ are now also included in Fig. 7 of the revised manuscript. The relative variations are similar to those found for $N_i^{5\,\mu m}$, despite a slightly weaker temperature dependence, possibly due to the less directly link between particles with $D > 25\,\mu m$ and homogeneous freezing processes. Maximum $N_i^{25\,\mu m}$ (found at $T_c < $ -70°C in the tropics) are about $100\,L^{-1}$, which is more consistent with values found in the studies referred to here by the referee. Exact comparisons between our results and previous in situ findings would nevertheless require further investigation that are out of the scope of this study.

Sec. 6.1 has been substantially edited to include all the aforementioned discussions, and further explanations on the consistency between $N_i$ and $N_{i(top)}$ maps are now also given in the revised part 2 manuscript

—————————————

8. RC: Page 23, lines 1-3 and Fig. 10: Fig. 10 and this text indicate that in the midlatitudes for T ¡ -40 C, Ni is highest during winter and lowest during summer. This same result was found in Mitchell et al. (2016). One of the ACP review criteria questions is "Do the authors give proper credit to related work and clearly indicate their own new/original

AR: We agree that the consistency between our results and those of Mitchell et al. [2016], especially in the mid-latitude, should have been included. A paragraph is now dedicated to these comparisons in Sec. 6.1.

—————————————

Minor comments

RC: 1. Page 15, line 9: much => slightly?
2. Page 19, line 6: follows => follow?
3. Page 22, line 13: at => as?
4. Figure 10 caption: Mention the meaning of the dashed curve.
5. Page 20, line 1: an => a?

AR: We thank the referee for pointing this out, these typos are corrected in the revised manuscript.

**References:**

J. Delanoë, A. J. Heymsfield, A. Protat, A. Bansemer, and R. J. Hogan. Normalized particle size distribution for remote sensing application. *J. Geophys. Res*, 119(7):4204–4227, 2014. doi: 10.1002/2013JD020700. URL http://dx.doi.org/10.1002/2013JD020700.

E. Erfani and D. L. Mitchell. Developing and bounding ice particle mass- and area-dimension expressions for use in atmospheric models and remote sensing. *Atmos. Chem. Phys.*, 16(7):4379–4400, 2016. doi: 10.5194/acp-16-4379-2016.

A. Korolev. Reconstruction of the sizes of spherical particles from their shadow images. part i: Theoretical considerations. *J. Atmos. Oceanic Technol.*, 24(3):376–389, 2007. doi: 10.1175/JTECH1980.1.

M. Krämer, C. Schiller, A. Afchine, R. Bauer, I. Gensch, A. Mangold, S. Schlicht, N. Spelten, N. Sitnikov, S. Borrmann, M. de Reus, and P. Spichtinger. Ice supersaturations and cirrus cloud crystal numbers. *Atmos. Chem. Phys.*, 9(11):3505–3522, 2009. doi: 10.5194/acp-9-3505-2009.

R. P. Lawson. Effects of ice particles shattering on the 2d-s probe. *Atmos. Meas. Tech.*, 4(7):1361–1381, 2011. doi: 10.5194/amt-4-1361-2011. URL http://www.atmos-meas-tech.net/4/1361/2011/.

R. P. Lawson, D. O'Connor, P. Zmarzly, K. Weaver, B. Baker, Q. Mo, and H. Jonsson. The 2D-S (Stereo) probe: Design and preliminary tests of a new airborne, high-speed, high-resolution particle imaging probe. *J. Atmos. Oceanic Technol.*, 23(11):1462–1477, 2006. doi: 10.1175/JTECH1927.1.

D. L. Mitchell, A. Garnier, M. Avery, and E. Erfani. Calipso observations of the dependence of homo- and heterogeneous ice nucleation in cirrus clouds on latitude, season and surface condition. *Atmos. Chem. Phys. Discuss.*, 2016:1–60, 2016. doi: 10.5194/acp-2016-1062.

D. L. Mitchell, A. Garnier, J. Pelon, and E. Erfani. Calipso (iir-caliop) retrievals of cirrus cloud ice particle concentrations. *Atmospheric Chemistry and Physics Discussions*, 2018:1–60, 2018. doi: 10.5194/acp-2018-526.

M. Paukert, C. Hoose, and M. Simmel. Redistribution of ice nuclei between cloud and rain droplets: Parameterization and application to deep convective clouds. *J. Adv. Model. Earth Sy.*, 9(1):514–535, 2017. doi: 10.1002/2016MS000841.

---

## Author Comment (AC3) · 17 Jul 2018

Response to referee #2 (in RC3)

RC: This paper describes an ice concentration retrieval based on the DARDAR Cloud-Sat/CALIPSO combined lidar-radar retrieval. The extension of DARDAR to retrieve ice concentrations, evaluation by comparison with in situ aircraft measurements, and global distributions are discussed. Although the ice concentration retrieval seems reasonable and potentially useful, I have significant concerns with the paper in its current version. In particular, I think the validity of the retrieval in regions without both lidar extinction and radar reflectivity needs much more discussion and evaluation. Also, the use of 2D-S measurements for determining concentrations of small ice crystals is suspect at best. These issues (and others) are discussed in detail below.

AR: We are thankful to the referee for the insightful comments listed in this review. We fully agree with these two major concerns regarding the behaviour of DARDAR-LIM under single-instrument conditions and the uncertainties related to small ice concentrations by the 2D-S. The manuscript has been substantially edited, with the support of supplementary materials and an appendix, to provide further clarifications and discussions on these two points. Detailed answers to each of the referee's comments are provided below.
* * *
RC: 1. The discussion of the retrieval algorithm in section 2 implicitly assumes that both extinction from the lidar and radar reflectivity are available. The authors should make clear early in the paper that the ice concentration retrieval is dubious in cirrus that are not detected by both radar and lidar (either too optically thin for detection by the CloudSat radar or below optically thick layers where the CALIOP lidar is blocked). When only lidar backscatter or radar reflectivity are available, the ice concentration is entirely dependent on the assumed size distribution. Mean PSDs are shown in the paper, but aircraft data shows that enormous PSD temporal and spatial variability is typically prevalent in cirrus. With only lidar or radar data available, this variability cannot be captured by the retrieval.

AR: The referee is absolutely correct in that DARDAR-LIM can by impacted by the absence of either lidar or radar reflectivity (i.e., referred to as radar- and lidar-only retrievals) and that this issue should explicitly be discussed in the manuscript. Following this comment, the behaviour of the algorithm under such conditions is now discussed in Sec. 3.1 as well as in Appendix A1. These sections clarify that two aspects are important to consider:

- First, it is correct that optimal retrievals should be expected in lidar-radar conditions due to having two pieces of information available to constrain both scaling parameters of the normalized size distribution ($D_\mathrm{m}$ and $N_0^*$). When only one instrument is available, DARDAR must rely on a priori assumptions, and in particular a relation between $N_0^*$ $\alpha_\mathrm{ext}$ and the temperature. Nevertheless, DARDAR also propagates, through its optimal estimation scheme, information vertically by using lidar-radar retrievals within the same column to further constrain this relation. The quality of lidar-only and radar-only $N_\mathrm{i}$ estimates is therefore difficult to predict. A propagation of the operational retrieval uncertainties is now proposed in the revised manuscript (see Sec. A2 and figure S2 of the supplementary materials). Figure S2 in particular shows that errors are indeed minimum in lidar-radar conditions, about 25% against 50% in lidar- and radar-only conditions, at their respective

maximum of occurence. These numbers should nevertheless be carefully accounted for because DARDAR was not designed to retrieve $N_i$ and importance quantities, like the shape of the PSD through the $\alpha$ and $\beta$ parameters, are not rigorously represented.

– However, it should also be pointed out that, despite instrumental sensitivity, it can be reasonable to expect that lidar-only $N_i$ estimates can in certain conditions be more accurate than lidar-radar retrievals. Indeed, lidar-only regions are often met at cloud top where the ice clouds are optically thin. Such conditions are likely to be met by small ice crystals that have not yet aggregated, and therefore display a rather monomodal size distribution that is easier to accurately be reproduced by D05. We recall that D05 assumes a monomodal shape and our study has already shown that this is a major limitation of the current method. Under lidar-radar conditions, i.e. deeper in a thick cloud structures, the PSDs are likely to become more complex and the monomodal-shape approximation followed by D05 will not hold anymore, which leads to more uncertain retrievals. In order to clarify this point, a new figure (Fig. 3) has been added to the revised manuscript. This figure explicitly compares the in situ PSDs measured by the 2D-S (coincident with A-Train overpasses, in black) to the PSD predicted by D05 using $D_m$ and $N_0^*$ from the 2D-S (i.e. the "optimal retrievals"; in red) and the PSD actually retrieved by DARDAR-LIM (i.e. using $D_m$ and $N_0^*$ from DARDAR; in blue). It is clear that in many cases the DARDAR-LIM PSD is close to the D05 PSD, indicating enough sensitivity to properly constrain $D_m$ and $N_0^*$ in all instrumental conditions. It is also interesting to note that good agreements to the 2D-S PSDs are obtained in lidar-only conditions due to their tendencies to be monomodal with less large crystals.

Therefore, deciding on the accuracy of DARDAR-LIM $N_i$ estimates in lidar-only conditions is not trivial, as it depends on the instrumental sensitivity as well as the PSD shape that are met in a given cloud parcel. The manuscript has been revised to make this more clear to the reader.
* * *
RC: 2. Page 6, lines 24-28: It would be helpful if some formal estimate of the uncertainties in Ni retrievals associated with measurement uncertainties could be provided.

AR: We agree that a formal estimates of the uncertainties on $N_i$ due to instrumental error and non-retrieved parameters of the forward model in DARDAR could be useful to the reader. These were not provided in the original manuscript because, as mentioned in the previous point, DARDAR was not designed to estimate $N_i$ and so some non-retrieved parameters in the retrieval algorithm that are important to $N_i$ (such as the PSD small mode shape) have not clearly been considered and included for error calculation. We now propagated the Gaussian uncertainties attached to IWC and $N_0^*$ in order to provide quantitative uncertainties on $N_i^{5\,\mu m}$, $N_i^{25\,\mu m}$ and $N_i^{100\,\mu m}$, which could be considered as lower error bounds. This is now discussed in Sec. 3.1, A2 and Fig. S2 of the supplements. Complementarily, the impact of the shape parameters on $N_i$ is also discussed in this section and in Fig. S3.
* * *
RC: 3. Page 6, lines 26-27: Further discussion of the the uncertainty in Ni retrieval associated with PSD shape assumption should be included. Perhaps examples could be provided as a guide.

AR: We thank the referee for this comment, we fully agree that more discussion on the impact of the PSD shape should be included. This is now discussed in Sec. 3.1, A3 and in Fig. S3 of the supplements. In this figure, several examples of PSD shapes are considered. Values of $N_0^*$ and $D_m$ representative of 3 temperature bins, based on the in situ campaigns used in this study, are considered as well as examples of 9 couples of $\alpha$ and $\beta$ parameters extracted from several in situ campaigns by Delanoë et al. [2014]. It clearly appears in the top figure of S3 that these parameters indeed greatly influence the assumed PSD shape. Consequences on $N_i^{5\,\mu m}$, $N_i^{25\,\mu m}$ and $N_i^{100\,\mu m}$ are shown below. Considering the typical values reported by Delanoë et al. [2014], an overestimation of about 50% can reasonably be expected on $N_i^{5\,\mu m}$, with the exception of one sub-visible (thin cirrus) case representative of a much higher concentration in small crystals than D05. Lower uncertainties due to the PSD shape are expected on $N_i^{25\,\mu m}$ due to a lesser influence of the $\alpha$ parameter.
* * *
RC: 4. As noted in the manuscript, only 2D-S data was available from SPARTICUS. The 2D-S ice concentrations are overwhelmingly dominated by the 1st size bin (5-15 $\mu m$). Artifacts and uncertainties render the first bin or two of 2D-S measurements relatively useless. Most 2D-S data users do not use concentrations in the first two bins in their analyses because of the large uncertainties. I would recommend excluding the first two bins in the PSD comparisons shown in Figure 1 for temperature bins for which little or no ATTREX data is available. Also, I think it is inappropriate to use $N_i^{5\,\mu m}$ data from the SPARTICUS 2D-S-only dataset for evaluation of the satellite retrievals. The MACPEX 2D-S data should only be used for $N_i^{25\,\mu m}$ and $N_i^{100\,\mu m}$.

AR: We agree with the referee concerning the high degree of uncertainties of the ice concentrations measured in the 2D-S first bin (5-15 $\mu m$). Despite that this matter is still actively discussed, as the response to this review provided by Referee #1 (RC4) illustrates very well, it is important to be careful when dealing with this data. That is why several $D_{min}$ thresholds were used in this study, including a $D_{min}=25\,\mu m$ that allowed for excluding the first two size bins of the 2D-S. The reader can then decide on the degree of trust they put on the 2D-S data and, subsequently, on the DARDAR-LIM evaluation. It should be noted that all three thresholds investigated here ($D_{min}=5$ 25 and $100\,\mu m$) are part of the product that will soon be made publicly available. After careful consideration, we have decided to not completely remove $N_i^{5\,\mu m}$ analyses from the manuscript but the discussions were instead largely edited throughout the manuscript to remind that $N_i^{25\,\mu m}$ is a more trustworthy reference when it comes to 2D-S data. Further discussions on issues with 2D-S measurements in its first 2 bins have also been included in Sec. 3.2 and 3.3. Finally, all analyses (including for the case study and geographic distributions) are now extended to $N_i^{25\,\mu m}$. The conclusion was also edited to stress the need for more evaluation of the DARDAR-LIM $N_i^{5\,\mu m}$ because of these issues. We hope that this should provide enough information to the reader to make an educated choice regarding its use of the $N_i$ dataset that will be provided co-jointly with these papers.

However, it should be noted that $N_i^{5\,\mu m}$ predictions by D05 agree fairly well with the FCDP and NIXE-CAPS measurements (with a possible overestimation by less than a factor of 2 but a good correlation). This gives in further confidence in that $N_i^{5\,\mu m}$ by DARDAR-LIM are useful and contain physical meaning, even if further investigation based on coincident flights will be required to assess the accuracy of their absolute value.

Following the referee's advice Fig. 1 was edited to exclude the two first bins of 2D-S where little SPARTICUS data is available by comparison to ATTREX (i.e., for $T_c <$-60°C). However, the MACPEX 2D-S data has not been added to this study as it would not add much more information by comparison to the dataset already used, especially considering the uncertainties on the 2D-S.

Finally, it should be mentioned that, following a comment in by Referee #1 (in RC2), the SPARTICUS dataset has been updated. The new dataset is now based on 2D-S data treated with a different processing method for $D < 365\,\mu$m (see response to RC2 or edits in Sec. 3.2.1 of the revised manuscript). This does not change in any way the conclusions of this study but slights differences in the figures will be noted.

––––––––––––––––

RC: 5. Figure 1: Indicate in figure or caption which temperature ranges correspond to AT-TREX data (mostly <-70 ° C) and SPARTICUS data (warmer temperature ranges).

AR: We agree that this would be a useful information to the readers. In order to avoid including too much information in Fig. 1, an histogram showing the temperature distribution for all included campaigns is added in the supplements (Fig. S1) and is referred to in the caption of Fig. 1.

––––––––––––––––

RC: 6. Figure 2: The authors should note and discuss the D05 overestimate (by factor of 2-3) for small (D $<10\,\mu$m) particles in -80 to -70°C bin compared to ATTREX measurements.

AR: Thank you for noting this important point. It is true that D05 seems to overestimate the concentration in small ice crystals (smaller than about 15 to 25 $\mu$m) due to a too steep representation of the PSD small mode (too negative $\alpha$ coefficient). This is now noted and discussed in the analysis of this figure (now Fig. 1 in revised manuscript) and the now Fig. 2 of the revised manuscript, which indicates a subsequent overestimation by a factor less than 2 on $N_i^{5\,\mu m}$. This point is also now mentioned in the conclusion as it is an aspect that should be improved in future parameterizations used for $N_i$ retrievals. It can be mentioned that the PSD parameterization planned for the next versions of DARDAR possesses a less steep representation of the concentration in small particles (i.e., a less negative $\alpha$, as can be noted in Fig. S3, the new parameterization being "all (DARDAR)"). Improvements are therefore expected in future DARDAR-LIM versions, but discussing them at this stage is out of the scope of this paper.

––––––––––––––––

RC: 7. Figure 3: The small sample volume of the FCDP instrument results in discretization of the ice number concentration in steps of about 12 conc.bin$^{-1}$. In other words, the FCDP instrument cannot effectively measure ice concentrations smaller than about 10-20 L$^{-1}$ if the data is used at 1 Hz (as in this study). The CAS data has a similar sample volume issue. Since ice concentrations are often dominated by the small crystals sampled by FCDP and CAS, I would recommend not showing the in situ vs D05 comparisons for concentrations less than 10

$\text{L}^{-1}$. In some of the temperature bins, the data extends to ice concentrations greater than 1000 $\text{L}^{-1}$. Extending the upper limit on the Figure 3 axes would be helpful to show how well the retrieval compares with in situ measurements at higher ice concentrations.

AR: We completely agree with the referee that the FCDP and CAS instruments are not optimal for measuring small concentrations less than about 15 $\text{L}^{-1}$. However, it can be argued that in the occurence of such small $\text{N}_\text{i}$, the overall concentration is likely to be dominated by large particles that are not measured by these two instruments. Additionally, a minimum detection limit is used in the treatment of the CAS and FCDP data so that concentration smaller than that threshold leads to no signal. Therefore, concentrations less than 15 $\text{L}^{-1}$ only originate from ice crystals larger than $25\,\mu m$ in the 1-Hz dataset. This is not necessarily the case in our dataset as 10 1-Hz PSDs are here merged to create PSDs representative of a 10-s sampling (comparable to the CloudSat overpass, as discussed in Sec. 3.2.2). We therefore kept $1\,\text{L}^{-1}$ as the lowest value for these analyses.

Nevertheless, following this comment, the concentrations axes in Fig. 3 (now Fig. 2 of the revised manuscript) have been extended to 5000 $\text{L}^{-1}$ to encompass all measured concentrations.

––––––––––––––––

RC: 8. Figure 3: The authors should note that discrepancies up to factors of 2-3 occur but are difficult to see with the log-log axis scales.

AR: We thank the referee for pointing this out. Additional lines have been added to this figure in order to explicitly show a factor of 2 and 3 around the one-to-one line.

––––––––––––––––

RC: 9. Will the Ni data be made publicly available? If so, data quality flags should be included to indicate when both radar and lidar signals are available as well as when the retrieval is questionable based on in situ comparisons?

AR: Yes, we think that it is extremely important that this dataset is made publicly available as soon as possible, hopefully together with these papers. A procedure has been initiated to distribute this $\text{N}_\text{i}$ dataset via the ICARE data center (http://www.icare.univ-lille1.fr), next to the operational DARDAR product. Level-2 (orbital retrievals) as well as Level-3 (daily/monthly gridded means) are currently being produced and we hope to be able to announce a DOI together with the final version of the manuscript. A choice was made to wait for the end of the reviews in case of significant changes to the methodology were requested.

The L2 dataset will include $\text{N}_\text{i}^{5\,\mu m}$, $\text{N}_\text{i}^{25\,\mu m}$, $\text{N}_\text{i}^{100\,\mu m}$, as well as numerous flags that will allows to filter for instrumental conditions, cloud types and iteration numbers. It is difficult to create an additional quantitative flag that will reflect the conclusions of the in situ comparison made in this paper but the temperature (from ECMWF reanalyses) is included in the L2 dataset to provide some flexibility to the users in that direction. A filtering following what has been done for Sec. 6 of this study will be applied for the L3 climatologies.

––––––––––––––––

 Page 14, lines 1-6: I do not understand what the authors are saying here. I was under the impression that Figures 2 and 3 simply showed statistical comparisons between the in-situ-measured and retrieved PSDs and ice concentrations. The first paragraph of section 4.2 suggests the comparisons in section 4.1 were ideal cases. Perhaps this idealization should be explained and emphasized at the beginning of section 4.1.

AR: We thank the referee for noting that this point was not very clear. As indicated in the introduction and the beginning of Sec. 3.2, two main questions are investigated in this in situ evaluation. First it is determined if D05, which predicts PSDs on the basis of IWC and $N_0^*$, is capable of accurately predicting the concentration in small particle and therefore $N_i^{5\,\mu m}$ and $N_i^{25\,\mu m}$. Second, it is checked that enough sensitivity is available in lidar and radar measurements to properly constrain these two input parameters. Fig. 1-3 of the original manuscript responded to the first question by comparing in situ measurements of PSDs and $N_i$ to equivalent predictions by D05 (obtained on the basis of IWC and $N_0^*$ extracted from the same in situ data). In other terms, these correspond to optimal $N_i$ estimates from DARDAR-LIM since we assume that IWC and $N_0^*$ perfectly fit the in situ measurements, as if they were perfectly retrieved by DARDAR. This allows to disentangle the errors originating from PSD shape assumptions, which are tested here, from errors related with a lack of sensitivity in lidar-radar measurements, which are investigated later using co-incident flights. Therefore, these comparisons allow to clearly conclude on the limitations of the D05 parameterization and what needs to be improved (e.g. a better representation of the bi-modality) to obtain better $N_i$ estimates.

This point was clarified by editing the first paragraphs of Sec. 4.1 and 4.2 in the revised manuscript.
* * *
 Section 4.2: I am not convinced that the near-coincident in situ/satellite retrieval comparisons are useful given the enormous spatial/temporal variability in cloud properties and the corresponding need for very close time and space coincidences for meaningful comparisons. Not surprisingly, the scatter in the comparisons shown in Figure 4 is very large, spanning 1-2 orders of magnitude.

AR: We completely acknowledge that it is extremely difficult to colocate and compare aircraft and satellite measurements. Such attempts are still common to evaluated satellite products, including DARDAR [Deng et al., 2012], and always show a strong scatter in direct comparisons. However, even if 2D-S and DARDAR-LIM and not comparable one-to-one, the constraints taken here on the time and space collocation (i.e., 5 km and 30 min) should at least allows them to be statistically representative of similar cloud samples.

Fig. 4 of the original manuscript did not provide very good quantitative comparisons and so it is now moved to the supplementary materials (see Fig. S4 and S5). It can still be noted that the average agreement (around the center of the 1-$\sigma$ isoline in dashed white) agrees well with the one-to-one line for $N_i^{100\,\mu m}$ and shows an expected overestimation by a factor of about 2 to 3 for $N_i^{25\,\mu m}$ and $N_i^{5\,\mu m}$. This overestimation is consistent with expectations from the limited representation of the PSD shape by D05, as can be observed by comparing Fig. S4 and S5.

Instead, comparisons of the PSDs measured by the 2D-S, predicted by D05 based on 2D-S data, and retrieved by DARDAR-LIM are shown in the new Fig. 3 of the revised manuscript. This figure shows that DARDAR-LIM PSDs are very consistent with the D05 predictions,

meaning that the cloud volumes sampled by the 2D-S and CloudSat/CALIPSO are statistically comparable most each temperature bins and instrumental conditions. The comparisons in terms of histograms, shown in Fig. 4 of the revised manuscript, now also include mean $N_i$ values to allow for a more quantitative statistical comparisons.

Section 4.2 has therefore been substantially edited to include and adapt to these new analyses, which should provide more convincing evidence of the statistical comparability of 2D-S and coincident DARDAR-LIM products.

—————————————

RC: 12. Page 15, line 9-10: In contrast to the statement here, the DARDAR-LIM retrieval overestimates $N_{D>5\mu m}$ and $N_{D>25\mu m}$ compared to SPARTICUS data even in the -60 to -50° C temperature bins.

AR: We thank the referee for pointing this out. The corresponding sentence has now been removed from the revised manuscript because this section has been substantially edited, following the response to the previous point. As a response to this comment, lines have been added in Fig. 4 of the original manuscript (now Fig. S4) in order to explicitly show a factor of 3 around the one-to-one line and identify more clearly these overestimations.

—————————————

RC: 13. Figure 5: The comparisons shown here are very difficult to see, particularly those for lidar-only and radar-only retrievals. The relative agreement between lidar-radar, radar-only, and lidar-only retrievals should be shown in a separate figure, particularly since the lidar-only and radar-only retrievals are suspect. Also, as discussed above, the SPARTICUS 2D-S-only ice concentrations for D>5 $\mu m$ are dominated by the first size bin, with enormous associated uncertainties. The comparisons with SPARTICUS 2D-S-only ice concentrations including the first bin are of little value, possibly misleading, and should be removed.

AR: We acknowledge that Fig. 5 of the original manuscript (now also Fig. 5) did not provide clear quantitative messages regarding the differences between 2D-S, D05 and DARDAR-LIM. These were even more difficult to observe for lidar-only and radar-only conditions as often less retrievals are available. We have responded to this issue by included the values of geometric means of $N_i^{5\,\mu m}$, $N_i^{25\,\mu m}$ and $N_i^{100\,\mu m}$ for 2D-S, D05 and DARDAR-LIM. The overall mean values are shown, as well as the values corresponding to $N_i$ estimates obtained in lidar-, radar-only and lidar-radar conditions. Also, as advised by the referee, individual histograms for each of these conditions are shown in Fig. S6 to provide a more clear idea concerning the influence of different instrumental conditions on the retrievals.

Regarding the evaluation of $N_i^{5\,\mu m}$, we have chosen to keep the corresponding plots in this analyses for the reasons discussed in the response to point 4. However, we fully agree with the referee that great care should be taken when presenting results using concentration from the first size bin of the 2D-S. Sec. 4.2 has therefore been edited so that its analyses are more centered on $N_i^{25\,\mu m}$ and to contain an explicit warning to the reader that $N_i^{25\,\mu m}$ constitutes the better reference concentration for the 2D-S. The revised conclusion also repeats this message as a reminded that more evaluation of $N_i^{5\,\mu m}$ remains necessary.

—————————————

RC: 14. Page 19, lines 8-10: the lack of clear transitions in retrieved properties between the lidar-only, lidar-radar, and radar-only regions does not necessarily mean the lidar- only and radar-only Ni retrievals are credible.

AR: We agree with this comment, a lack of transition between lidar/radar-only and lidar-radar regions does not necessarily prove the quality of $N_i$ estimates obtained in single-instrument conditions. This sentence was more meant as an observation rather than a definite proof, and has therefore been toned down in the revised manuscript. This sentence is also more justified now that the quality of lidar- and radar-only $N_i$ estimates is further discussed in the revised Sec. 4.2. Nevertheless, this lack of transition is still worth commenting on as it represents a very impressive feature of the DARDAR algorithm, which allows for a real multispectral consistency between it's lidar and radar retrievals. It also shows that some information is used to constrain the $N_i$ estimates as there do not seem to jump back to some a priori value.

―――――――――――――

RC: 15. Figure 8: Scatter plots of $N_{D>5\mu m}$ and $N_{D>100\mu m}$ versus $N_{2D-S}$ would provide much clearer comparisons between the retrievals and measurements. Further, the points could be color coded to indicate whether they are in the lidar-only, lidar-radar, and radar-only regions. In the discussion of Figure 8, the authors claim good agreement between the in situ and retrieved ice concentrations, and they dismiss glaring discrepancies as being caused by the imperfect time coincidence. This argument seems unjustified. The flight track segment has been chosen for good time/space coincidence.

AR: We thank the referee for this comment, it has allowed us to realize that the in situ analyses included in the case study was perhaps not clearly explained. Because of this, and following a comment in RC1 this figure has been moved to supplementary materials (see Fig. S7) and is now only briefly mentioned. This is also justified since this figure mainly supported the previous conclusions, with redundant results. We however think that this figure can in this format still provide good insights on the quality of DARDAR-LIM as it shows that the satellite estimates are to some extent well capable of reproducing the spatial variability in $N_i^{5\,\mu m}$, $N_i^{25\,\mu m}$ and $N_i^{100\,\mu m}$ measured by the 2D-S. This is why scatterplots are not proposed for this figure, especially since scatterplots have already been widely examined before. The added-value of figure is to show a consistency in the spatial distribution along the aircraft fight.

We nevertheless agree with the referee's comments and have added information on the instrumental conditions met in DARDAR-LIM in the new figure (see color background in Fig. S7). The $N_i^{25\,\mu m}$ is also included in this study, as it now represents the new reference for small ice concentration from the 2D-S. Finally, the part of the flight leg that was further away from the satellite overpass (more than 10 km) has been removed and the coordinates have been changed so that the reader can easy grasp the temporal and spacial distance between the aircraft track and the satellite overpass.

We agree that discrepancies appear, in particular in the $N_i^{25\,\mu m}$ comparisons and during the descending leg, which we attributed to the distance (about 10 km) from the track. The hypothesis of different cloud sampling between the satellite and the aircraft appears reasonable especially since different $N_i^{100\,\mu m}$ values are noted during this descending leg. We recall that no serious issues are expected in the DARDAR-LIM $N_i^{100\,\mu m}$ estimates based on the previous in situ evaluations and note that similar increases of $N_i^{100\,\mu m}$ can be observed in Fig. 6(h) (of the

revised manuscript) right next to the descending part of the track. It is therefore reasonable to attribute issues in $N_i^{100\,\mu m}$ to the sampling different cloud volumes, which means that differences in $N_i^{5\,\mu m}$ and $N_i^{25\,\mu m}$ could as well be expected in this area.

––––––––––––––––

RC: 16. Section 5.3: The authors describe a cloud formation scenario with air parcels ascending across the -40°C isoline, which suggests that freezing of liquid drops could be the main ice formation mechanism. Yet they attribute the differences between the high and low ice concentration regions to differences in vertical wind speed and cite the strong sensitivity of Ni to w (citing Krämer et al. 2016; papers showing this sensitivity decades earlier should be cited). However, the strong sensitivity to w occurs primarily when aqueous aerosols freeze, not so much when liquid droplets freeze. Either the description is not clear, or the physical argument made does not make sense.

AR: We are thankful to the reviewer for pointing this out. It appears that the explanation proposed in the original manuscript was misleading. We indeed meant that the relation between $N_i$ and $w$, which we show via the analysis of back-trajectories, is the result of homogeneous freezing events of aqueous aerosols. These events are however likely to occur on top of existing ice crystals formed from liquid droplets, as it is clear that supercooled layers appear close to the region of high $N_i^{5\,\mu m}$ and $N_i^{25\,\mu m}$ (seen in the $\beta_{ext}$ profile). We have edited this paragraph and added references and comparisons to studies that analysed these processes [e.g. Kärcher and Ström, 2003, Kärcher et al., 2006].

––––––––––––––––

RC: 17. Figures 9 and 10: The discrepancy between $N_{D>5\mu m}$ and ATTREX FCDP ice concentrations noted above is apparent in the coldest temperature bins and the TTL. Typical values of ND>5 $\mu m$ are 200-300 $L^{-1}$, whereas ATTREX in situ measurements indicate ice concentrations of about $100\,L^{-1}$ (Jensen et al., 2016). It is also interesting that the ice concentrations are higher over continental and convective regions even in the coldest temperature bins (near the tropical tropopause) where the vast majority of clouds form in situ. Additionally, it might be worth noting that the statistics must be poor in the coldest temperature bin poleward of about 30° latitude since such cold temperatures rarely occur there.

AR: We are grateful to the referee for pointing this out and relating these analyses to the observed $N_i^{5\,\mu m}$ discrepancies between D05 and the FCDP at very low temperatures. The section has been edited to note these, as well as the disagreements between concentrations observed in these figures and the findings of Jensen et al. [2013, 2016] for TTL cirrus. These could indeed be caused by a misrepresentation of the small ice concentration in D05, which seems to overestimate the steepness of the concentration towards small particles at low temperatures and therefore overestimates of $N_i^{5\,\mu m}$ (by a factor less than 2). Interestingly, the spatial distributions of $N_i^{25\,\mu m}$ (less impacted by shape assumptions), now added to Fig. 7 of the revised manuscript, indicates concentrations of about $100\,L^{-1}$ in TTL regions. It can also be noted that $N_i^{5\,\mu m}$ in the same regions and during winter months (see Fig. 8 of revised manuscript), i.e. when in situ clouds should be even more dominant, are about 50% lower. It therefore appears difficult to strongly conclude on disagreements by comparison to in situ observations without an exact knowledge of what cloud type is present in each lat-lon-$T_c$ bin, and further analyses will be

needed to fully assess this disagreement with Jensen et al. [2013, 2016]. This discussion has now been summarized in Sec. 6.1 of the revised manuscript.

We also agree that additional information on the statistical significance of the results provided here would be useful. Fig. S8 and S9 of the supplementary material now indicate the pixel counts corresponding to the spatial and zonal distributions analysed in Sec. 6.

—————————————

RC: 18. Page 22, line 7: Simply stating that the spatial distributions agree with the global model predictions is no doubt too strong. A quick examination shows there are some regions of agreement and some glaring discrepancies. I would either omit this statement or qualify it. Perhaps the comparison really shouldn't be discussed without providing much more detail.

AR: We thank the referee for this comment and fully agree with it. Further comparisons to modeling is beyond the scope of this study and this sentence has been deleted from the manuscript.

—————————————

RC: 19. Section 6.2: Most of the speculations about the physical causes of the zonal-height distributions in this section are either not justified or would require much more detail to adequately discuss. It does not look to me like there is a particularly sharp transition at -40°C, nor would one be expected given the importance of sedimentation in cirrus. The retrieval probably doesn't work well in the antarctic wintertime stratosphere since PSCs are typically mixtures of ice crystals, NAT particles, and ternary aerosols.

AR: We agree with this point, Sec. 6.2 contained some analyses that remained too hypothetical, such as the attribution of some $N_i$ patterns to PSCs. After further investigation, it appears that DARDAR retrievals in these regions are highly uncertain due to potential failures in the cloud mask and wrong categorizations of cloud pixels. This does not mean that DARDAR retrievals are always wrong in these regions but further investigation based on the new version of the DARDAR mask should be performed and are out of the scope of this study. It is now clearly stated in the manuscript that this feature in $N_i$ should not be trusted. Regarding the vertical transition at -40°C, it could still be argued that $N_i^{5\,\mu m}$ and $N_i^{25\,\mu m}$ values quickly change around this temperature, especially in the tropics. $N_i^{25\,\mu m}$ for instance increases from about $50\,L^{-1}$ to above $100\,L^{-1}$. We nevertheless agree that this transition cannot be considered sharp and have toned down this analysis.

Following this comment, Sec. 6.2 has been edited to provide further discussion and remove analyses that seemed too far-fetched. The analysis of seasonal variations of $N_i$ are now also supported by Fig. 8 of the revised manscript, which shows seasonal variability in spatial distributions. It should also be noted that this section also comes as a natural transition between part 1 and part 2, which further investigates some of the observations made in Sec. 6.2. This is now also reminded.

**References:**

J. Delanoë, A. J. Heymsfield, A. Protat, A. Bansemer, and R. J. Hogan. Normalized particle size distribution for remote sensing application. *J. Geophys. Res*, 119(7):4204–4227, 2014. doi: 10.1002/2013JD020700. URL http://dx.doi.org/10.1002/2013JD020700.

M. Deng, G. G. Mace, Z. Wang, and R. P. Lawson. Evaluation of several a-train ice cloud retrieval products with in situ measurements collected during the sparticus campaign. *J. Appl. Meteor. and Clim.*, 52(4):1014–1030, 2012. doi: 10.1175/JAMC-D-12-054.1.

E. J. Jensen, G. Diskin, R. P. Lawson, S. Lance, T. P. Bui, D. Hlavka, M. McGill, L. Pfister, O. B. Toon, and R. Gao. Ice nucleation and dehydration in the tropical tropopause layer. *Proceedings of the National Academy of Sciences*, 110(6):2041–2046, 2013. doi: 10.1073/pnas.1217104110.

E. J. Jensen, R. Ueyama, L. Pfister, T. V. Bui, R. P. Lawson, S. Woods, T. Thornberry, A. W. Rollins, G. S. Diskin, J. P. DiGangi, and M. A. Avery. On the susceptibility of cold tropical cirrus to ice nuclei abundance. *J. Atmos. Sci.*, 73(6):2445–2464, 2016. doi: 10.1175/JAS-D-15-0274.1.

B. Kärcher and J. Ström. The roles of dynamical variability and aerosols in cirrus cloud formation. *Atmos. Chem. Phys.*, 3(3):823–838, 2003. doi: 10.5194/acp-3-823-2003.

B. Kärcher, J. Hendricks, and U. Lohmann. Physically based parameterization of cirrus cloud formation for use in global atmospheric models. *J. Geophys. Res. Atmos.*, 111(D1), 2006. doi: 10.1029/2005JD006219.

---

## Author Comment (AC4) · 17 Jul 2018

Response to referee #1 (in RC4)

AR: We thank Referee #1 for taking the time to read comments from other reviews and provide such insightful comments. This was very helpful to the discussions that have taken place in the context of this overall review process.

This response by Referee #1 very well illustrates that the uncertainties expected on in situ measurements of the concentration in small ice crystals are still not completely understood and require further efforts and investigations. It is clear that large uncertainties occur in the two first size bins of the 2D-S but arguments such as provided here by Referee #1 seem to indicate that measurements in these bins may not be completely meaningless. This is exactly what has initially motivated us to provide the total ice particle concentrations integrated from multiple minimum size thresholds: $5\,\mu$m, $25\,\mu$m and $100\,\mu$m. As discussed in the manuscript, the absolute values of $N_i^{5\,\mu m}$ are associated with larger uncertainties than those of $N_i^{25\,\mu m}$. However, providing that these issues are clearly stated and discussed, it seems reasonable for $N_i^{5\,\mu m}$ to still be provided to users, who can then make an educated choice on whether or not this quantity is of interest for their studies. Looking at spatial variations in $N_i^{5\,\mu m}$, as mentioned here by Referee #1, are a perfect example of analyses that doesn't require the absolute value to be correct but only the relative changes to be physically meaningful. The results shown in the two parts of this study seem to provide good confidence in the latter point. This comment by Referee #1 has therefore further convinced us that $N_i^{5\,\mu m}$ should not be completely removed from the revised version of the manuscript.

Finally, it is important to emphasize that satellite products of $N_i$ are still at an early stage. They have too rarely been attempted and, most importantly, rigorously evaluated before. It is worth noting that, for this reason, "estimates" has been preferred to "retrievals" to describe $N_i$ in this manuscript. The results presented here, together with the recent studies by Mitchell et al. [2016, 2018], constitute first encourageing steps towards providing more accurate and well understood datasets of $N_i$ to the community. It is evident that DARDAR-LIM can still largely benefit from further improvements - the evaluation presented in this paper has identified several of them and work is in progress in that direction - but this two-part study also presents evidence that realistic and useful $N_i$ values can already be reached. The conclusions drawn here hopefully will serve as motivations for further developments of $N_i$ retrievals from remote sensing measurements.

**References:**

D. L. Mitchell, A. Garnier, M. Avery, and E. Erfani. Calipso observations of the dependence of homo- and heterogeneous ice nucleation in cirrus clouds on latitude, season and surface condition. *Atmos. Chem. Phys. Discuss.*, 2016:1–60, 2016. doi: 10.5194/acp-2016-1062.

D. L. Mitchell, A. Garnier, J. Pelon, and E. Erfani. Calipso (iir-caliop) retrievals of cirrus cloud ice particle concentrations. *Atmospheric Chemistry and Physics Discussions*, 2018:1–60, 2018. doi: 10.5194/acp-2018-526.

---

## Author Comment (AC5) · 17 Jul 2018

Response to referee #1 (in RC2)

RC: This paper presents a new method for retrieving the ice particle number concentration Ni for glaciated clouds, which should be useful for understanding aerosol interactions with ice clouds and the contribution of homogeneous vs. heterogeneous ice nucleation in cirrus clouds. A satellite remote sensing scheme for Ni is needed since field campaigns cannot adequately inform us how Ni varies with latitude and the seasons. The paper is well organized and well written, and usually cites the relevant literature. The quality of the figures is good. The methods developed in Sec. 5 for testing the retrieval are especially creative and effective.

AR: We are thankful to the referee for all the useful comments that greatly helped us to improve the quality and clarify of this study. In particular concerning the influence of the PSD shape assumptions, the use of 2D-S data and the consistency between the analyses of $N_i$ climatologies presented in the 2 papers of this study. Detailed responses to each comment are provided below.

—————————————

RC: A critical limitation of the retrieval algorithm is the use of a normalized universal ice particle size distribution, or PSD (Delanoë et al., 2005), where it is assumed that all PSD in nature conform to this normalized PSD shape. This normalized PSD is based on a four-parameter gamma function (Eq. 4) where parameters No and k can be deduced through their link with other operationally retrieved properties (IWC and $N_0^*$) while PSD parameters $\alpha$ and $\beta$ need to be fixed as constants. This is of little consequence regarding $\beta$, which affects the largest ice particles having the lowest concentrations. But this is of major consequence regarding $\alpha$, which strongly influences the smallest ice particles that govern Ni. This is not mentioned in the paper. The small end of the PSD is sensitive to the rate of ice nucleation which is sensitive to the cloud updraft w (with higher w making $\alpha$ more negative, and Ni higher), as well as the aggregation rate that removes smaller ice particles having higher concentration (Herzegh and Hobbs, 1985, QJRMS; Mitchell, 1991, JAS). Thus, some discussion on this topic is warranted, especially on the errors that may result from "non-standard" conditions where atypical updrafts are common (such as over steep orography).

AR: We thank the referee for pointing out the need for further discussion regarding the impact of non-retrieved shape parameters of the size distributions ($\alpha$ and $\beta$). We completely agree that this was lacking in the original manuscript.

DARDAR unfortunately does not rigorously account for these uncertainties in its operational retrievals, as they are only represented by additional fixed errors considered on the lidar and radar measurements. More rigorous techniques exist to propagate uncertainties on $\alpha$ and $\beta$ through the optimal estimation scheme but they would be too time consuming for an operational algorithm based on active instruments. However, the variability of these two parameters and the subsequent impact on DARDAR has been thoroughly discussed in Delanoë et al. [2014]. It can be noted that, as a result of this study, a revised version of the PSD parameterization has been proposed (notably with a less negative $\alpha$, leading to less small ice crystals and a lower $N_i$) but is not yet implemented in the operational product. The referee is therefore absolutely correct in saying that the fixed $\alpha$ and $\beta$ parameters constitute a strong limitation to our current method that should be further highlighted. These points are now discussed in Sec. 3.1 and in Appendix

A3 of the revised manuscript and are supported by additional figures in the supplementary materials (see Fig. S3).

The impact of the choice of $\alpha$ and $\beta$ on the PSD shape is clearly shown in the upper panel of Fig. S3, and the subsequent impact on $N_i$ because of straying from the selected values ($\alpha = -3$ and $\beta = -1$) is quantified in the lower figure. In order to propose a range of realistic shape parameters, values extracted by Delanoë et al. [2014] from individual in situ campaigns are used (color code in upper figure and shapes in the lower figure). IWC and $N_0^*$ values representative of 3 temperature bins are selected, although it should be kept in mind that each couple of coefficients from the D14 campaigns can realistically applied to only one of these temperature ranges. In agreement with the referee's comment, it can be observed that one D14 campaign displays a more negative $\alpha$, namely the "subvisible" campaign, which corresponds to cirrus measured at temperatures between -80 to -60°C during CRYSTAL-FACE (Cirrus Regional Study of Tropical Anvils and Cirrus-Layers-Florida Area Cirrus Experiment). We recognize that this analysis remains preliminary but it should still allow to provide rough estimates of the uncertainties on $N_i^{5\,\mu m}$, $N_i^{25\,\mu m}$ and $N_i^{100\,\mu m}$ to the reader. This overall uncertainty is here considered to be typically better than about 50% (when considering the variability between all D14 campaigns). This value is now reported in Sec. 3.1 and A3 of the revised manuscript.

———————————

RC: The lead author gave a nice talk about this retrieval at the A-Train Symposium in 2017. Henceforth, Ni refers to Ni for ice particle maximum dimension $D > 5\mu m$. Slide 20 of this presentation, showing global distributions of Ni for 10°C intervals, appears almost identical to Fig. 9 of this paper for T < -30°C, except that the Ni legends differ. The Ni values reported in the presentation are higher by a factor of about 1.7 relative to the Ni reported in Fig. 9 of this paper. What is the reason for this difference?

AR: We are grateful that the referee took the time to verify the consistency between this paper and the results presented during the A-Train Symposium. The figure referred to here (slide 20 of this presentation, available on http://atrain2017.org), corresponded to the $N_i$ integrated from $D_{min}= 1\mu m$. This may not have been clearly expressed during the presentation but is suggested by the absence of mention to the size in the label. The $1\,\mu m$ threshold was initially used at early stages of our analyses, but was subsequently changed to $5\,\mu m$ as it is impossible at this point to reasonably evaluate DARDAR-LIM between 1 and $5\,\mu m$. Also, as discussed before, uncertainties related to PSD shape assumptions are likely to be even more important if $D_{min} = 1\,\mu m$.

Fig. 1 shown in this response corresponds to the distribution of $N_i^{1\mu m}$ based on the dataset used for this paper. It can be noted that similar values to those shown during the A-Train Symposium presentation are found, despite small differences in absolute values. These could be due to an error found in the script that converts $D_{min}$ from maximum diameter to an equivalent melted size prior to the numerical integration of the PSD, which led to slightly underestimated $D_{min}$ and thus to higher concentration. This error was corrected before creating the dataset used in these papers. To the best of the first author's knowledge, there should be no other difference between Fig. 1 of this response and the figure in the A-Train Symposium presentation.

———————————

Major Comments

[Figure]

Figure 1: Spatial distribution of $N_i^{1\mu m}$ from 2006 to 2016, averaged in a $2 \times 2°$ lat-lon grid and per $10°$ C temperature bin from -80 to -30° C.

1. RC: Page 8, line 25: The 2DS photodiode array length is $1280\,\mu m$, which should be noted. Evidently the "time dimension" is used to size particles up to $3205\,\mu m$; please indicate the particle selection criteria used to size and count particles.

AR: We are very thankful to the referee for this comment that has led us to investigate in greater detail the various selection criteria for particle size and count that are available for the 2D-S instrument.

In the original manuscript, ATTREX-2014 data was processed with the method $M_1$, or $M_7$ method when available. There are important differences between these methods, in particular concerning the size selection, which are for instance extensively described and discussed in Lawson [2011] and Erfani and Mitchell [2016]. The SPARTICUS data was treated with the $M_1$ method only, as $M_7$ isn't operationally available in the ARM database. Comparing concentrations from these 2 methods should not be an issue as Erfani and Mitchell [2016] showed that the number concentration in small particle isn't significantly different between them.

Nevertheless, after further discussion with the 2D-S data providers at SPEC Inc. (P. Lawson and S. Woods), it appeared that using a SPARTICUS dataset based on a $M_4/M_1$ processing could be better adapted to the needs of this study. By $M_4/M_1$ it is meant that the $M_4$ method is used for particles sizes less than $365\,\mu m$ and the $M_1$ is used otherwise. A main differences between these two methods is that $M_4$ resizes out-of-focus particles to equivalent in-focus spheres [Korolev, 2007]. This becomes problematic when the ice particle shapes become strongly non-spherical, and this method can therefore only be applied to small particles. Consequently, it was decided with the SPARTICUS 2D-S data providers that a combined $M_4/M_1$ processing method should be used here.

The differences between the PSDs obtained from $M_1$ alone and $M_4/M_1$ are shown in Fig. 2 of this document. The main difference occur for sizes between about 30 to 100-200 $\mu m$, with typically more particles with $D < 100\mu m$ and less particles larger than this threshold. As a consequence, the bi-modal structure is less pronounced in $M_4/M_1$, but it also is clear from this figure that the results discussed in the original manuscript are not changed by this transition from $M_1$ to $M_4/M_1$ 2D-S data. It can also be mentioned that slightly less flights with $M_4/M_1$ treatments were available on the ARM database.

As a response to this comment, Sec. 3.2.1 was edited to explicitly mention the use of the $M_n$ methods.

Regarding the photodiode specifications, it seems that the 2DS photodiode array length is (if referring to the actual physical size of the array) of about $7.3\,mm$ [Lawson et al., 2006]. However, we fully agree with the referee that, because (i) the equivalent size of each photodiode

[Figure]

Figure 2: Comparison between the SPARTICUS 2D-S PSDs obtained from the M4/M1 (top; as in Fig. 1 of the revised manuscript) and the M1 method (bottom; as in Fig. 1 of original manuscript.)

is about $10\,\mu$m (considering the laser beam magnification) and (ii) the 2D-S being equipped with 128 photodiode, this instrument technically measures particules up to $1280\,\mu$m in size and so an extension to $3205\,\mu$m is only possible by using a time dimension (i.e., by using 2 consecutive measurements a of $1280\,\mu$m particle). This is now mentioned in Sec. 3.2.1 of the revised manuscript.
* * *
2. RC: Figure 5 and Sec. 4.2: For T > -50C, by what factor is Ni ($D_{min} = 5\mu$m) overestimated, on average? For T $\geq$ -50C?

AR: Based on Fig. 5 of the original manuscript, an overestimation of $N_i^{5\,\mu m}$ by a factor of about 2 to 3 can be considered if looking at the distance between the modes of the 2D-S and DARDAR-LIM distributions. We nevertheless agree that this figure did not provide an easy way to clearly quantify the bias, and visually comparing the modes does not really provide a real statistical estimate of the differences between DARDAR-LIM and the 2D-S. This figure has therefore been edited in order to include the geometric means associated with each histogram (DARDAR-LIM, D05 and 2D-S) for each temperature bin and instrumental condition. These should allow for a more quantitative discussion of the biases, included in the revised Sec. 4.2. For instance, overestimations by about 10 to 30% and 20 to 60% are found in the mean values of $N_i^{25\,\mu m}$ by D05 and DARDAR-LIM, respectively.

Complementarily, a line showing a factor of 3 around to one-to-one line has also been added to Fig. 4 of the original manuscript, now Fig. S4 of the complementary materials.

[Figure]

Figure 3: Left: Conditional density of $N_i^{5\,\mu m}$ as function of the temperature, obtained from the insitu data used in this paper (bottom) and the corresponding D05 predictions (top). Plain red lines indicate the median and dashed lines show the 10th and 90th percentiles. The right panel directly compares the medians and 10th and 90th percentile lines.
* * *
3. RC: Page 21, lines 9-12: The strong temperature dependence of Ni mentioned here appears at variance with the in situ measurements reported in Krämer et al. (2009). Please mention this.

 AR: We thank the referee for this comment, which has encouraged us to further compare our $N_i$ products with the insitu findings in several studies by Krämer et al.

 It should first be mentioned that it is very difficult to compare the temperature dependence of $N_i$ obtained from in situ campaigns to those from global results shown in Sec. 6. In situ measurements are rather sparse and it is often difficult to tell what part of the cloud has been sampled. However, the enormous advantage of the dataset by Krämer et al is indeed that it consistently merges numerous in situ campaigns and should therefore tend to being comparable to global satellite data. This dataset is still being improved as airborne campaigns are continuously being added. The Ni(T) relation reported in Krämer et al. [2009] was based on a dataset that was not yet very large and contains some flights in mountain wave clouds that enhanced the frequencies of higher ice concentrations. A new, yet unpublished, dataset called JULIA does not confirm the (slight) dependence of Ni on T shown in Krämer et al. [2009]

[Figure]

Figure 4: Same as 3 but for DARDAR-LIM $N_i^{5\,\mu m}$ compared to co-incident in situ observations during SPARTICUS.

(personal communication, M. Krämer).

This comment has motivated us to compare the Ni(T) obtained in this study with the one from JULIA. Due to the complexity of this task, intermediate steps were taken. First, we have verified that the issues noted with D05 (notably due to its limited shape assumptions) does not create clear biases in the Ni(T) relation. This is shown in Fig. 3 attached to this response, which compares the Ni(T) dependency obtained from the dataset used in this study to that predicted by D05 based on the in situ data. It can be noted that the consistency of the relation found from our dataset to the one from JULIA has been verified, although this cannot be directly demonstrated here due to the latter being unpublished. Fig. 3 clearly shows that D05 is very well capable of reproducing the relation between $N_i^{5\,\mu m}$ and $T_c$ found in the in situ measurements, and so similar results could be expected from DARDAR. This has been verified by looking at the same relations based on the co-incident SPARTICUS flights. Fig. 4 in this response shows that DARDAR-LIM reproduces well the Ni(T) relation observed by the 2D-S. We have checked that these results also hold for $N_i^{25\,\mu m}$ and that they are not sensitive to instrumental conditions.

Consequently, it could be expected that Ni(T) obtained from global DARDAR-LIM estimates are reasonable and that the observations from Sec. 6 are not necessarily at variance with in situ observations. However, the results presented here are preliminary and further analyses are necessary to confirm them. For instance, it would require to subset similar regions, cloud

type or distance from cloud top by comparison the in situ data. Rigorously assessing the consistency between Ni(T) observed from DARDAR-LIM and from in situ measurement would be extremely interesting but unfortunately out of the scope of this paper. These results will be the focus in a following study.

––––––––––––

4. RC: Figure 9 and Sec. 6.1: For T > -50°C, Ni tends to be lower over regions characterized by extensive marine stratus, like off the west coasts of South America and Africa (from equator and southwards). Is this result real, or is it an artefact of the retrieval? If the latter is true, please explain.

AR: We thank the referee for pointing this out. It is correct that $N_i$ (for all integration thresholds) tend to be relatively lower in marine stratocumulus regions. There does not seem to be any obvious reason to doubt the retrieval method in these regions but it should indeed be kept in mind that there are relatively less ice clouds in these subsidence regions. The spatial distributions of retrieval counts have now been added to supplementary materials (see Fig. S8) to help determining which regions correspond to statistically significant retrievals. Another physical explanation could be that there are no convective clouds in these regions, which seem to drive the high $N_i^{5\,\mu m}$ and $N_i^{25\,\mu m}$ observed in this figure. This is supported by the seasonal variabilities in $N_i$ maps shown in Fig. 8 of the revised manuscript. Consequently, values observed correspond to thin cirrus, perhaps remnants of aged anvils or jet stream cirrus, and $N_i^{5\,\mu m}$ values below $100\,L^{-1}$ for T > -50°C are thus not surprising, as mentioned in comment #7 of this review. It also means that $N_i^{5\,\mu m}$ in this regions are more comparable to cloud-top values observed in the part 2 paper. This is now noted in Sec. 6.1 of the revised manuscript.

––––––––––––

5. RC: Page 21, lines 14-19: A similar finding was reported in Mitchell et al. (2016, ACPD), where the highest Ni were associated with mountainous terrain. (Although this paper was rejected since the editor felt the retrieved Ni values were too high, and therefore could not be used to infer nucleation modes, no arguments cast doubt on the spatial and temporal relative differences in Ni, which still appear meaningful.)

AR: We fully agree that further comparisons to existing climatologies would be beneficial to the analyses in Sec. 6.1. A new paragraph discussing comparisons results by Mitchell et al. [2016, 2018] is now included.

––––––––––––

6. RC: Page 22, lines 7-9: It is more meaningful to compare model results against observations than vice-versa. Suggest removing this paragraph. For example, in the modeling study by Zhou et al. (2016, ACP), the sensitivity of homo- and heterogeneous ice nucleation to various model parameters and updraft schemes were evaluated. Depending on how these are represented, one can get a broad range of Ni-temperature dependences, including Ni that is relatively insensitive to temperature (similar to the in situ observations of Krämer et al., 2009, ACP), and that modeling result would not support these DARDAR-LIM findings.

AR: We agree with this comment, comparisons to modeling would require further analyses that are not in the scope of this paper. This paragraph is now removed

7. RC: Figure 10 and Sec. 6.2: $N_i$ ($D_{min}$= 5 um) in the tropics appears contrary to the $N_i$ results in Fig. 1 and Fig. 5 of Part 2 of this study by Gryspeerdt et al. (submitted). Fig. 1a of Gryspeerdt et al. show Ni near cloud top while their Fig. 5 shows that $N_i$ does not change appreciably with distance below cloud top (up to 3 km from cloud top) between -50 and -60°C. Assuming this result extends to other temperatures, the cloud top results in Fig. 1a of Gryspeerdt et al. should also be approximately valid below cloud top. Regarding Fig. 1a in Gryspeerdt et al., for T > -65°C, $N_i$ is never higher in the tropics relative to the midlatitudes. Between -55 and -40°C, where the most optically thick cirrus clouds exist (cirrus defined as clouds having T < -38°C), $N_i$ in the tropics is substantially lower than $N_i$ in the midlatitudes. In Fig. 10 of Part 1 (Sourdeval et al.), Ni increases abruptly in the tropics for T < -40°C (shown by the dashed curve), with Ni here being typically higher than Ni at similar T in the midlatitudes. This result appears opposite to the findings in Fig. 1a of Gryspeerdt et al. (Part 2). In addition, the CALIPSO Ni retrievals of Mitchell et al. (2016, ACPD) qualitatively support the findings of Gryspeerdt et al. (in terms of relative differences), and the in situ measurements from Mühlbauer et al. (2014) show relatively lower "peak Ni" values in anvil cirrus (vs. frontal, jet stream and ridge-crest cirrus). Finally, several studies (e.g. Jensen et al., 2013, PNAS; Spichtinger and Krämer, 2013, ACP), show that tropical tropopause layer (TTL) cirrus tend to have Ni < 30 L$^{-1}$. Since the areal coverage of TTL cirrus exceeds that of anvil cirrus, and TTL cirrus tend to be higher than anvil cirrus (Gasparini et al., 2017, J. Climate), the Ni of   200 L$^{-1}$ in the TTL region in Fig. 10 appears at variance with in situ observations. Please comment on, and, if possible, reconcile these issues.

AR: We thank the referee for this interesting comment. It has motivated us to further compare the spatial distributions obtained from cloud-top $N_i$ ($N_{i(top)}$) (part 2) vs. the "all cloud" maps (part 1).

It is first important to point out that this is not straightforward as these two maps are not necessarily representative of the same cloud types within a given temperature bin. For instance, the $N_{i(top)}$ map between -50 and -60°C (in part 2) only shows concentrations for clouds that have a cloud-top within this temperature bin, whereas the total $N_i$ map (in part 1) also features values that are within deep convective clouds. It is observed that the high values of $N_i^{5\,\mu m}$ and $N_i^{25\,\mu m}$ only appear in convective regions, which is confirmed by the seasonal variabilities showed in Fig. 8. The sampling difference is also clear when comparing retrieval counts between $N_{i(top)}$ and $N_i$ per $T_c$ bin, now showed in Fig. 1 of the revised part 2 paper and in Fig. S8 of the revised part 1 paper, respectively. Nearly no retrievals are present in the tropic for the $N_{i(top)}$ map, whereas convective clouds are present in the $N_i$ map. To support this analysis, it can be noted that high $N_i$ values found between -50 and -60°C within deep convective clouds is in agreement with modeling results by Paukert et al. [2017], who also reports $N_{i(top)}$ lower than $N_i$ for this cloud type.

It could as well be argued that the CALIPSO $N_i$ retrievals presented in Mitchell et al. [2016, 2018] are also more comparable to the $N_{i(top)}$ map as the thermal infrared measurements used in these studies extinguishes within a few optical depth. It is therefore reasonable to expect that retrievals from these studies would not compare exactly to $N_i$ maps presented in part 1 but more to the $N_{i(top)}$ maps presented in part 2, as it is the case (in terms of relative variations of $N_i$).

Regarding the absolute values of $N_i$, we completely agree with the referee that the ones

presented our maps may not be completely exact. An overestimation by a factor of 2, or even 3, could be expected on $N_i^{5\,\mu m}$ considering all uncertainties on the retrievals (especially concerning the assumptions on the PSD shape). These uncertainties should be smaller on $N_i^{25\,\mu m}$, as the impact of the shape is less significant, and the spatial distributions of $N_i^{25\,\mu m}$ are now also included in Fig. 7 of the revised manuscript. The relative variations are similar to those found for $N_i^{5\,\mu m}$, despite a slightly weaker temperature dependence, possibly due to the less directly link between particles with $D > 25\,\mu m$ and homogeneous freezing processes. Maximum $N_i^{25\,\mu m}$ (found at $T_c < $-70°C in the tropics) are about $100\,L^{-1}$, which is more consistent with values found in the studies referred to here by the referee. Exact comparisons between our results and previous in situ findings would nevertheless require further investigation that are out of the scope of this study.

Sec. 6.1 has been substantially edited to include all the aforementioned discussions, and further explanations on the consistency between $N_i$ and $N_{i(top)}$ maps are now also given in the revised part 2 manuscript

––––––––––––––––

8. RC: Page 23, lines 1-3 and Fig. 10: Fig. 10 and this text indicate that in the midlatitudes for T ¡ -40 C, Ni is highest during winter and lowest during summer. This same result was found in Mitchell et al. (2016). One of the ACP review criteria questions is "Do the authors give proper credit to related work and clearly indicate their own new/original

AR: We agree that the consistency between our results and those of Mitchell et al. [2016], especially in the mid-latitude, should have been included. A paragraph is now dedicated to these comparisons in Sec. 6.1.

––––––––––––––––

Minor comments

RC: 1. Page 15, line 9: much => slightly?
2. Page 19, line 6: follows => follow?
3. Page 22, line 13: at => as?
4. Figure 10 caption: Mention the meaning of the dashed curve.
5. Page 20, line 1: an => a?

AR: We thank the referee for pointing this out, these typos are corrected in the revised manuscript.

Response to referee #1 (in RC4)

AR: We thank Referee #1 for taking the time to read comments from other reviews and provide such insightful comments. This was very helpful to the discussions that have taken place in the context of this overall review process.

This response by Referee #1 very well illustrates that the uncertainties expected on in situ measurements of the concentration in small ice crystals are still not completely understood and require further efforts and investigations. It is clear that large uncertainties occur in the two first size bins of the 2D-S but arguments such as provided here by Referee #1 seem to indicate that measurements in these bins may not be completely meaningless. This is exactly what has initially motivated us to provide the total ice particle concentrations integrated from multiple minimum size thresholds: $5\,\mu$m, $25\,\mu$m and $100\,\mu$m. As discussed in the manuscript, the absolute values of $N_i^{5\,\mu m}$ are associated with larger uncertainties than those of $N_i^{25\,\mu m}$. However, providing that these issues are clearly stated and discussed, it seems reasonable for $N_i^{5\,\mu m}$ to still be provided to users, who can then make an educated choice on whether or not this quantity is of interest for their studies. Looking at spatial variations in $N_i^{5\,\mu m}$, as mentioned here by Referee #1, are a perfect example of analyses that doesn't require the absolute value to be correct but only the relative changes to be physically meaningful. The results shown in the two parts of this study seem to provide good confidence in the latter point. This comment by Referee #1 has therefore further convinced us that $N_i^{5\,\mu m}$ should not be completely removed from the revised version of the manuscript.

Finally, it is important to emphasize that satellite products of $N_i$ are still at an early stage. They have too rarely been attempted and, most importantly, rigorously evaluated before. It is worth noting that, for this reason, "estimates" has been preferred to "retrievals" to describe $N_i$ in this manuscript. The results presented here, together with the recent studies by Mitchell et al. [2016, 2018], constitute first encourageing steps towards providing more accurate and well understood datasets of $N_i$ to the community. It is evident that DARDAR-LIM can still largely benefit from further improvements - the evaluation presented in this paper has identified several of them and work is in progress in that direction - but this two-part study also presents evidence that realistic and useful $N_i$ values can already be reached. The conclusions drawn here hopefully will serve as motivations for further developments of $N_i$ retrievals from remote sensing measurements.

Response to referee #2 (in RC3)

RC: This paper describes an ice concentration retrieval based on the DARDAR Cloud-Sat/CALIPSO combined lidar-radar retrieval. The extension of DARDAR to retrieve ice concentrations, evaluation by comparison with in situ aircraft measurements, and global distributions are discussed. Although the ice concentration retrieval seems reasonable and potentially useful, I have significant concerns with the paper in its current version. In particular, I think the validity of the retrieval in regions without both lidar extinction and radar reflectivity needs much more discussion and evaluation. Also, the use of 2D-S measurements for determining concentrations of small ice crystals is suspect at best. These issues (and others) are discussed in detail below.

AR: We are thankful to the referee for the insightful comments listed in this review. We fully agree with these two major concerns regarding the behaviour of DARDAR-LIM under single-instrument conditions and the uncertainties related to small ice concentrations by the 2D-S. The manuscript has been substantially edited, with the support of supplementary materials and an appendix, to provide further clarifications and discussions on these two points. Detailed answers to each of the referee's comments are provided below.

———————————

RC: 1. The discussion of the retrieval algorithm in section 2 implicitly assumes that both extinction from the lidar and radar reflectivity are available. The authors should make clear early in the paper that the ice concentration retrieval is dubious in cirrus that are not detected by both radar and lidar (either too optically thin for detection by the CloudSat radar or below optically thick layers where the CALIOP lidar is blocked). When only lidar backscatter or radar reflectivity are available, the ice concentration is entirely dependent on the assumed size distribution. Mean PSDs are shown in the paper, but aircraft data shows that enormous PSD temporal and spatial variability is typically prevalent in cirrus. With only lidar or radar data available, this variability cannot be captured by the retrieval.

AR: The referee is absolutely correct in that DARDAR-LIM can by impacted by the absence of either lidar or radar reflectivity (i.e., referred to as radar- and lidar-only retrievals) and that this issue should explicitly be discussed in the manuscript. Following this comment, the behaviour of the algorithm under such conditions is now discussed in Sec. 3.1 as well as in Appendix A1. These sections clarify that two aspects are important to consider:

- First, it is correct that optimal retrievals should be expected in lidar-radar conditions due to having two pieces of information available to constrain both scaling parameters of the normalized size distribution ($D_{\mathrm{m}}$ and $N_0^*$). When only one instrument is available, DARDAR must rely on a priori assumptions, and in particular a relation between $N_0^*$ $\alpha_{\mathrm{ext}}$ and the temperature. Nevertheless, DARDAR also propagates, through its optimal estimation scheme, information vertically by using lidar-radar retrievals within the same column to further constrain this relation. The quality of lidar-only and radar-only $N_{\mathrm{i}}$ estimates is therefore difficult to predict. A propagation of the operational retrieval uncertainties is now proposed in the revised manuscript (see Sec. A2 and figure S2 of the supplementary materials). Figure S2 in particular shows that errors are indeed minimum in lidar-radar conditions, about 25% against 50% in lidar- and radar-only conditions, at their respective

maximum of occurence. These numbers should nevertheless be carefully accounted for because DARDAR was not designed to retrieve $N_i$ and importance quantities, like the shape of the PSD through the $\alpha$ and $\beta$ parameters, are not rigorously represented.

– However, it should also be pointed out that, despite instrumental sensitivity, it can be reasonable to expect that lidar-only $N_i$ estimates can in certain conditions be more accurate than lidar-radar retrievals. Indeed, lidar-only regions are often met at cloud top where the ice clouds are optically thin. Such conditions are likely to be met by small ice crystals that have not yet aggregated, and therefore display a rather monomodal size distribution that is easier to accurately be reproduced by D05. We recall that D05 assumes a monomodal shape and our study has already shown that this is a major limitation of the current method. Under lidar-radar conditions, i.e. deeper in a thick cloud structures, the PSDs are likely to become more complex and the monomodal-shape approximation followed by D05 will not hold anymore, which leads to more uncertain retrievals. In order to clarify this point, a new figure (Fig. 3) has been added to the revised manuscript. This figure explicitly compares the in situ PSDs measured by the 2D-S (coincident with A-Train overpasses, in black) to the PSD predicted by D05 using $D_m$ and $N_0^*$ from the 2D-S (i.e. the "optimal retrievals"; in red) and the PSD actually retrieved by DARDAR-LIM (i.e. using $D_m$ and $N_0^*$ from DARDAR; in blue). It is clear that in many cases the DARDAR-LIM PSD is close to the D05 PSD, indicating enough sensitivity to properly constrain $D_m$ and $N_0^*$ in all instrumental conditions. It is also interesting to note that good agreements to the 2D-S PSDs are obtained in lidar-only conditions due to their tendencies to be monomodal with less large crystals.

Therefore, deciding on the accuracy of DARDAR-LIM $N_i$ estimates in lidar-only conditions is not trivial, as it depends on the instrumental sensitivity as well as the PSD shape that are met in a given cloud parcel. The manuscript has been revised to make this more clear to the reader.

–––––––––––––––

RC: 2. Page 6, lines 24-28: It would be helpful if some formal estimate of the uncertainties in Ni retrievals associated with measurement uncertainties could be provided.

AR: We agree that a formal estimates of the uncertainties on $N_i$ due to instrumental error and non-retrieved parameters of the forward model in DARDAR could be useful to the reader. These were not provided in the original manuscript because, as mentioned in the previous point, DARDAR was not designed to estimate $N_i$ and so some non-retrieved parameters in the retrieval algorithm that are important to $N_i$ (such as the PSD small mode shape) have not clearly been considered and included for error calculation. We now propagated the Gaussian uncertainties attached to IWC and $N_0^*$ in order to provide quantitative uncertainties on $N_i^{5\,\mu m}$, $N_i^{25\,\mu m}$ and $N_i^{100\,\mu m}$, which could be considered as lower error bounds. This is now discussed in Sec. 3.1, A2 and Fig. S2 of the supplements. Complementarily, the impact of the shape parameters on $N_i$ is also discussed in this section and in Fig. S3.

–––––––––––––––

RC: 3. Page 6, lines 26-27: Further discussion of the the uncertainty in Ni retrieval associated with PSD shape assumption should be included. Perhaps examples could be provided as a guide.

AR: We thank the referee for this comment, we fully agree that more discussion on the impact of the PSD shape should be included. This is now discussed in Sec. 3.1, A3 and in Fig. S3 of the supplements. In this figure, several examples of PSD shapes are considered. Values of $N_0^*$ and $D_m$ representative of 3 temperature bins, based on the in situ campaigns used in this study, are considered as well as examples of 9 couples of $\alpha$ and $\beta$ parameters extracted from several in situ campaigns by Delanoë et al. [2014]. It clearly appears in the top figure of S3 that these parameters indeed greatly influence the assumed PSD shape. Consequences on $N_i^{5\,\mu m}$, $N_i^{25\,\mu m}$ and $N_i^{100\,\mu m}$ are shown below. Considering the typical values reported by Delanoë et al. [2014], an overestimation of about 50% can reasonably be expected on $N_i^{5\,\mu m}$, with the exception of one sub-visible (thin cirrus) case representative of a much higher concentration in small crystals than D05. Lower uncertainties due to the PSD shape are expected on $N_i^{25\,\mu m}$ due to a lesser influence of the $\alpha$ parameter.
* * *
RC: 4. As noted in the manuscript, only 2D-S data was available from SPARTICUS. The 2D-S ice concentrations are overwhelmingly dominated by the 1st size bin (5-15 $\mu m$). Artifacts and uncertainties render the first bin or two of 2D-S measurements relatively useless. Most 2D-S data users do not use concentrations in the first two bins in their analyses because of the large uncertainties. I would recommend excluding the first two bins in the PSD comparisons shown in Figure 1 for temperature bins for which little or no ATTREX data is available. Also, I think it is inappropriate to use $N_i^{5\,\mu m}$ data from the SPARTICUS 2D-S-only dataset for evaluation of the satellite retrievals. The MACPEX 2D-S data should only be used for $N_i^{25\,\mu m}$ and $N_i^{100\,\mu m}$.

AR: We agree with the referee concerning the high degree of uncertainties of the ice concentrations measured in the 2D-S first bin (5-15 $\mu m$). Despite that this matter is still actively discussed, as the response to this review provided by Referee #1 (RC4) illustrates very well, it is important to be careful when dealing with this data. That is why several $D_{min}$ thresholds were used in this study, including a $D_{min}$=25 $\mu m$ that allowed for excluding the first two size bins of the 2D-S. The reader can then decide on the degree of trust they put on the 2D-S data and, subsequently, on the DARDAR-LIM evaluation. It should be noted that all three thresholds investigated here ($D_{min}$=5 25 and 100 $\mu m$) are part of the product that will soon be made publicly available. After careful consideration, we have decided to not completely remove $N_i^{5\,\mu m}$ analyses from the manuscript but the discussions were instead largely edited throughout the manuscript to remind that $N_i^{25\,\mu m}$ is a more trustworthy reference when it comes to 2D-S data. Further discussions on issues with 2D-S measurements in its first 2 bins have also been included in Sec. 3.2 and 3.3. Finally, all analyses (including for the case study and geographic distributions) are now extended to $N_i^{25\,\mu m}$. The conclusion was also edited to stress the need for more evaluation of the DARDAR-LIM $N_i^{5\,\mu m}$ because of these issues. We hope that this should provide enough information to the reader to make an educated choice regarding its use of the $N_i$ dataset that will be provided co-jointly with these papers.

However, it should be noted that $N_i^{5\,\mu m}$ predictions by D05 agree fairly well with the FCDP and NIXE-CAPS measurements (with a possible overestimation by less than a factor of 2 but a good correlation). This gives in further confidence in that $N_i^{5\,\mu m}$ by DARDAR-LIM are useful and contain physical meaning, even if further investigation based on coincident flights will be required to assess the accuracy of their absolute value.

Following the referee's advice Fig. 1 was edited to exclude the two first bins of 2D-S where little SPARTICUS data is available by comparison to ATTREX (i.e., for $T_c$ <-60°C). However, the MACPEX 2D-S data has not been added to this study as it would not add much more information by comparison to the dataset already used, especially considering the uncertainties on the 2D-S.

Finally, it should be mentioned that, following a comment in by Referee #1 (in RC2), the SPARTICUS dataset has been updated. The new dataset is now based on 2D-S data treated with a different processing method for $D < 365\,\mu m$ (see response to RC2 or edits in Sec. 3.2.1 of the revised manuscript). This does not change in any way the conclusions of this study but slights differences in the figures will be noted.

––––––––––––––

RC: 5. Figure 1: Indicate in figure or caption which temperature ranges correspond to AT-TREX data (mostly <-70 ° C) and SPARTICUS data (warmer temperature ranges).

AR: We agree that this would be a useful information to the readers. In order to avoid including too much information in Fig. 1, an histogram showing the temperature distribution for all included campaigns is added in the supplements (Fig. S1) and is referred to in the caption of Fig. 1.

––––––––––––––

RC: 6. Figure 2: The authors should note and discuss the D05 overestimate (by factor of 2-3) for small (D <10 $\mu m$) particles in -80 to -70°C bin compared to ATTREX measurements.

AR: Thank you for noting this important point. It is true that D05 seems to overestimate the concentration in small ice crystals (smaller than about 15 to 25 $\mu m$) due to a too steep representation of the PSD small mode (too negative $\alpha$ coefficient). This is now noted and discussed in the analysis of this figure (now Fig. 1 in revised manuscript) and the now Fig. 2 of the revised manuscript, which indicates a subsequent overestimation by a factor less than 2 on $N_i^{5\,\mu m}$. This point is also now mentioned in the conclusion as it is an aspect that should be improved in future parameterizations used for $N_i$ retrievals. It can be mentioned that the PSD parameterization planned for the next versions of DARDAR possesses a less steep representation of the concentration in small particles (i.e., a less negative $\alpha$, as can be noted in Fig. S3, the new parameterization being "all (DARDAR)"). Improvements are therefore expected in future DARDAR-LIM versions, but discussing them at this stage is out of the scope of this paper.

––––––––––––––

RC: 7. Figure 3: The small sample volume of the FCDP instrument results in discretization of the ice number concentration in steps of about 12 conc.bin$^{-1}$. In other words, the FCDP instrument cannot effectively measure ice concentrations smaller than about 10-20 L$^{-1}$ if the data is used at 1 Hz (as in this study). The CAS data has a similar sample volume issue. Since ice concentrations are often dominated by the small crystals sampled by FCDP and CAS, I would recommend not showing the in situ vs D05 comparisons for concentrations less than 10

L$^{-1}$. In some of the temperature bins, the data extends to ice concentrations greater than 1000 L$^{-1}$. Extending the upper limit on the Figure 3 axes would be helpful to show how well the retrieval compares with in situ measurements at higher ice concentrations.

AR: We completely agree with the referee that the FCDP and CAS instruments are not optimal for measuring small concentrations less than about 15 L$^{-1}$. However, it can be argued that in the occurence of such small N$_i$, the overall concentration is likely to be dominated by large particles that are not measured by these two instruments. Additionally, a minimum detection limit is used in the treatment of the CAS and FCDP data so that concentration smaller than that threshold leads to no signal. Therefore, concentrations less than 15 L$^{-1}$ only originate from ice crystals larger than 25 $\mu$m in the 1-Hz dataset. This is not necessarily the case in our dataset as 10 1-Hz PSDs are here merged to create PSDs representative of a 10-s sampling (comparable to the CloudSat overpass, as discussed in Sec. 3.2.2). We therefore kept 1 L$^{-1}$ as the lowest value for these analyses.

Nevertheless, following this comment, the concentrations axes in Fig. 3 (now Fig. 2 of the revised manuscript) have been extended to 5000 L$^{-1}$ to encompass all measured concentrations.

————————————————

RC: 8. Figure 3: The authors should note that discrepancies up to factors of 2-3 occur but are difficult to see with the log-log axis scales.

AR: We thank the referee for pointing this out. Additional lines have been added to this figure in order to explicitly show a factor of 2 and 3 around the one-to-one line.

————————————————

RC: 9. Will the Ni data be made publicly available? If so, data quality flags should be included to indicate when both radar and lidar signals are available as well as when the retrieval is questionable based on in situ comparisons?

    AR: Yes, we think that it is extremely important that this dataset is made publicly available as soon as possible, hopefully together with these papers. A procedure has been initiated to distribute this N$_i$ dataset via the ICARE data center (http://www.icare.univ-lille1.fr), next to the operational DARDAR product. Level-2 (orbital retrievals) as well as Level-3 (daily/monthly gridded means) are currently being produced and we hope to be able to announce a DOI together with the final version of the manuscript. A choice was made to wait for the end of the reviews in case of significant changes to the methodology were requested.

    The L2 dataset will include N$_i^{5\,\mu m}$, N$_i^{25\,\mu m}$, N$_i^{100\,\mu m}$, as well as numerous flags that will allows to filter for instrumental conditions, cloud types and iteration numbers. It is difficult to create an additional quantitative flag that will reflect the conclusions of the in situ comparison made in this paper but the temperature (from ECMWF reanalyses) is included in the L2 dataset to provide some flexibility to the users in that direction. A filtering following what has been done for Sec. 6 of this study will be applied for the L3 climatologies.

————————————————

 I do not understand what the authors are saying here. I was under the impression that Figures 2 and 3 simply showed statistical comparisons between the in-situ-measured and retrieved PSDs and ice concentrations. The first paragraph of section 4.2 suggests the comparisons in section 4.1 were ideal cases. Perhaps this idealization should be explained and emphasized at the beginning of section 4.1.

AR: We thank the referee for noting that this point was not very clear. As indicated in the introduction and the beginning of Sec. 3.2, two main questions are investigated in this in situ evaluation. First it is determined if D05, which predicts PSDs on the basis of IWC and $N_0^*$, is capable of accurately predicting the concentration in small particle and therefore $N_i^{5\,\mu m}$ and $N_i^{25\,\mu m}$. Second, it is checked that enough sensitivity is available in lidar and radar measurements to properly constrain these two input parameters. Fig. 1-3 of the original manuscript responded to the first question by comparing in situ measurements of PSDs and $N_i$ to equivalent predictions by D05 (obtained on the basis of IWC and $N_0^*$ extracted from the same in situ data). In other terms, these correspond to optimal $N_i$ estimates from DARDAR-LIM since we assume that IWC and $N_0^*$ perfectly fit the in situ measurements, as if they were perfectly retrieved by DARDAR. This allows to disentangle the errors originating from PSD shape assumptions, which are tested here, from errors related with a lack of sensitivity in lidar-radar measurements, which are investigated later using co-incident flights. Therefore, these comparisons allow to clearly conclude on the limitations of the D05 parameterization and what needs to be improved (e.g. a better representation of the bi-modality) to obtain better $N_i$ estimates.

This point was clarified by editing the first paragraphs of Sec. 4.1 and 4.2 in the revised manuscript.
* * *
 I am not convinced that the near-coincident in situ/satellite retrieval comparisons are useful given the enormous spatial/temporal variability in cloud properties and the corresponding need for very close time and space coincidences for meaningful comparisons. Not surprisingly, the scatter in the comparisons shown in Figure 4 is very large, spanning 1-2 orders of magnitude.

AR: We completely acknowledge that it is extremely difficult to colocate and compare aircraft and satellite measurements. Such attempts are still common to evaluated satellite products, including DARDAR [Deng et al., 2012], and always show a strong scatter in direct comparisons. However, even if 2D-S and DARDAR-LIM and not comparable one-to-one, the constraints taken here on the time and space collocation (i.e., 5 km and 30 min) should at least allows them to be statistically representative of similar cloud samples.

Fig. 4 of the original manuscript did not provide very good quantitative comparisons and so it is now moved to the supplementary materials (see Fig. S4 and S5). It can still be noted that the average agreement (around the center of the 1-$\sigma$ isoline in dashed white) agrees well with the one-to-one line for $N_i^{100\,\mu m}$ and shows an expected overestimation by a factor of about 2 to 3 for $N_i^{25\,\mu m}$ and $N_i^{5\,\mu m}$. This overestimation is consistent with expectations from the limited representation of the PSD shape by D05, as can be observed by comparing Fig. S4 and S5.

Instead, comparisons of the PSDs measured by the 2D-S, predicted by D05 based on 2D-S data, and retrieved by DARDAR-LIM are shown in the new Fig. 3 of the revised manuscript. This figure shows that DARDAR-LIM PSDs are very consistent with the D05 predictions,

meaning that the cloud volumes sampled by the 2D-S and CloudSat/CALIPSO are statistically comparable most each temperature bins and instrumental conditions. The comparisons in terms of histograms, shown in Fig. 4 of the revised manuscript, now also include mean $N_i$ values to allow for a more quantitative statistical comparisons.

Section 4.2 has therefore been substantially edited to include and adapt to these new analyses, which should provide more convincing evidence of the statistical comparability of 2D-S and coincident DARDAR-LIM products.

––––––––––––––––––––

RC: 12. Page 15, line 9-10: In contrast to the statement here, the DARDAR-LIM retrieval overestimates $N_{D>5\mu m}$ and $N_{D>25\mu m}$ compared to SPARTICUS data even in the -60 to -50° C temperature bins.

AR: We thank the referee for pointing this out. The corresponding sentence has now been removed from the revised manuscript because this section has been substantially edited, following the response to the previous point. As a response to this comment, lines have been added in Fig. 4 of the original manuscript (now Fig. S4) in order to explicitly show a factor of 3 around the one-to-one line and identify more clearly these overestimations.

––––––––––––––––––––

RC: 13. Figure 5: The comparisons shown here are very difficult to see, particularly those for lidar-only and radar-only retrievals. The relative agreement between lidar-radar, radar-only, and lidar-only retrievals should be shown in a separate figure, particularly since the lidar-only and radar-only retrievals are suspect. Also, as discussed above, the SPARTICUS 2D-S-only ice concentrations for D>5 $\mu m$ are dominated by the first size bin, with enormous associated uncertainties. The comparisons with SPARTICUS 2D-S-only ice concentrations including the first bin are of little value, possibly misleading, and should be removed.

AR: We acknowledge that Fig. 5 of the original manuscript (now also Fig. 5) did not provide clear quantitative messages regarding the differences between 2D-S, D05 and DARDAR-LIM. These were even more difficult to observe for lidar-only and radar-only conditions as often less retrievals are available. We have responded to this issue by included the values of geometric means of $N_i^{5\,\mu m}$, $N_i^{25\,\mu m}$ and $N_i^{100\,\mu m}$ for 2D-S, D05 and DARDAR-LIM. The overall mean values are shown, as well as the values corresponding to $N_i$ estimates obtained in lidar-, radar-only and lidar-radar conditions. Also, as advised by the referee, individual histograms for each of these conditions are shown in Fig. S6 to provide a more clear idea concerning the influence of different instrumental conditions on the retrievals.

Regarding the evaluation of $N_i^{5\,\mu m}$, we have chosen to keep the corresponding plots in this analyses for the reasons discussed in the response to point 4. However, we fully agree with the referee that great care should be taken when presenting results using concentration from the first size bin of the 2D-S. Sec. 4.2 has therefore been edited so that its analyses are more centered on $N_i^{25\,\mu m}$ and to contain an explicit warning to the reader that $N_i^{25\,\mu m}$ constitutes the better reference concentration for the 2D-S. The revised conclusion also repeats this message as a reminded that more evaluation of $N_i^{5\,\mu m}$ remains necessary.

––––––––––––––––––––

RC: 14. Page 19, lines 8-10: the lack of clear transitions in retrieved properties between the lidar-only, lidar-radar, and radar-only regions does not necessarily mean the lidar- only and radar-only Ni retrievals are credible.

AR: We agree with this comment, a lack of transition between lidar/radar-only and lidar-radar regions does not necessarily prove the quality of $N_i$ estimates obtained in single-instrument conditions. This sentence was more meant as an observation rather than a definite proof, and has therefore been toned down in the revised manuscript. This sentence is also more justified now that the quality of lidar- and radar-only $N_i$ estimates is further discussed in the revised Sec. 4.2. Nevertheless, this lack of transition is still worth commenting on as it represents a very impressive feature of the DARDAR algorithm, which allows for a real multispectral consistency between it's lidar and radar retrievals. It also shows that some information is used to constrain the $N_i$ estimates as there do not seem to jump back to some a priori value.

––––––––––––––––––––

RC: 15. Figure 8: Scatter plots of $N_{D>5\mu m}$ and $N_{D>100\mu m}$ versus $N_{2D-S}$ would provide much clearer comparisons between the retrievals and measurements. Further, the points could be color coded to indicate whether they are in the lidar-only, lidar-radar, and radar-only regions. In the discussion of Figure 8, the authors claim good agreement between the in situ and retrieved ice concentrations, and they dismiss glaring discrepancies as being caused by the imperfect time coincidence. This argument seems unjustified. The flight track segment has been chosen for good time/space coincidence.

AR: We thank the referee for this comment, it has allowed us to realize that the in situ analyses included in the case study was perhaps not clearly explained. Because of this, and following a comment in RC1 this figure has been moved to supplementary materials (see Fig. S7) and is now only briefly mentioned. This is also justified since this figure mainly supported the previous conclusions, with redundant results. We however think that this figure can in this format still provide good insights on the quality of DARDAR-LIM as it shows that the satellite estimates are to some extent well capable of reproducing the spatial variability in $N_i^{5\,\mu m}$, $N_i^{25\,\mu m}$ and $N_i^{100\,\mu m}$ measured by the 2D-S. This is why scatterplots are not proposed for this figure, especially since scatterplots have already been widely examined before. The added-value of figure is to show a consistency in the spatial distribution along the aircraft fight.

We nevertheless agree with the referee's comments and have added information on the instrumental conditions met in DARDAR-LIM in the new figure (see color background in Fig. S7). The $N_i^{25\,\mu m}$ is also included in this study, as it now represents the new reference for small ice concentration from the 2D-S. Finally, the part of the flight leg that was further away from the satellite overpass (more than 10 km) has been removed and the coordinates have been changed so that the reader can easy grasp the temporal and spacial distance between the aircraft track and the satellite overpass.

We agree that discrepancies appear, in particular in the $N_i^{25\,\mu m}$ comparisons and during the descending leg, which we attributed to the distance (about 10 km) from the track. The hypothesis of different cloud sampling between the satellite and the aircraft appears reasonable especially since different $N_i^{100\,\mu m}$ values are noted during this descending leg. We recall that no serious issues are expected in the DARDAR-LIM $N_i^{100\,\mu m}$ estimates based on the previous in situ evaluations and note that similar increases of $N_i^{100\,\mu m}$ can be observed in Fig. 6(h) (of the

revised manuscript) right next to the descending part of the track. It is therefore reasonable to attribute issues in $N_i^{100\,\mu m}$ to the sampling different cloud volumes, which means that differences in $N_i^{5\,\mu m}$ and $N_i^{25\,\mu m}$ could as well be expected in this area.

––––––––––

RC: 16. Section 5.3: The authors describe a cloud formation scenario with air parcels ascending across the -40°C isoline, which suggests that freezing of liquid drops could be the main ice formation mechanism. Yet they attribute the differences between the high and low ice concentration regions to differences in vertical wind speed and cite the strong sensitivity of Ni to w (citing Krämer et al. 2016; papers showing this sensitivity decades earlier should be cited). However, the strong sensitivity to w occurs primarily when aqueous aerosols freeze, not so much when liquid droplets freeze. Either the description is not clear, or the physical argument made does not make sense.

AR: We are thankful to the reviewer for pointing this out. It appears that the explanation proposed in the original manuscript was misleading. We indeed meant that the relation between $N_i$ and $w$, which we show via the analysis of back-trajectories, is the result of homogeneous freezing events of aqueous aerosols. These events are however likely to occur on top of existing ice crystals formed from liquid droplets, as it is clear that supercooled layers appear close to the region of high $N_i^{5\,\mu m}$ and $N_i^{25\,\mu m}$ (seen in the $\beta_{ext}$ profile). We have edited this paragraph and added references and comparisons to studies that analysed these processes [e.g. Kärcher and Ström, 2003, Kärcher et al., 2006].

––––––––––

RC: 17. Figures 9 and 10: The discrepancy between $N_{D>5\mu m}$ and ATTREX FCDP ice concentrations noted above is apparent in the coldest temperature bins and the TTL. Typical values of ND>5 $\mu$m are 200-300 $L^{-1}$, whereas ATTREX in situ measurements indicate ice concentrations of about $100\,L^{-1}$ (Jensen et al., 2016). It is also interesting that the ice concentrations are higher over continental and convective regions even in the coldest temperature bins (near the tropical tropopause) where the vast majority of clouds form in situ. Additionally, it might be worth noting that the statistics must be poor in the coldest temperature bin poleward of about 30° latitude since such cold temperatures rarely occur there.

AR: We are grateful to the referee for pointing this out and relating these analyses to the observed $N_i^{5\,\mu m}$ discrepancies between D05 and the FCDP at very low temperatures. The section has been edited to note these, as well as the disagreements between concentrations observed in these figures and the findings of Jensen et al. [2013, 2016] for TTL cirrus. These could indeed be caused by a misrepresentation of the small ice concentration in D05, which seems to overestimate the steepness of the concentration towards small particles at low temperatures and therefore overestimates of $N_i^{5\,\mu m}$ (by a factor less than 2). Interestingly, the spatial distributions of $N_i^{25\,\mu m}$ (less impacted by shape assumptions), now added to Fig. 7 of the revised manuscript, indicates concentrations of about $100\,L^{-1}$ in TTL regions. It can also be noted that $N_i^{5\,\mu m}$ in the same regions and during winter months (see Fig. 8 of revised manuscript), i.e. when in situ clouds should be even more dominant, are about 50% lower. It therefore appears difficult to strongly conclude on disagreements by comparison to in situ observations without an exact knowledge of what cloud type is present in each lat-lon-$T_c$ bin, and further analyses will be

needed to fully assess this disagreement with Jensen et al. [2013, 2016]. This discussion has now been summarized in Sec. 6.1 of the revised manuscript.

We also agree that additional information on the statistical significance of the results provided here would be useful. Fig. S8 and S9 of the supplementary material now indicate the pixel counts corresponding to the spatial and zonal distributions analysed in Sec. 6.

———————————

RC: 18. Page 22, line 7: Simply stating that the spatial distributions agree with the global model predictions is no doubt too strong. A quick examination shows there are some regions of agreement and some glaring discrepancies. I would either omit this statement or qualify it. Perhaps the comparison really shouldn't be discussed without providing much more detail.

AR: We thank the referee for this comment and fully agree with it. Further comparisons to modeling is beyond the scope of this study and this sentence has been deleted from the manuscript.

———————————

RC: 19. Section 6.2: Most of the speculations about the physical causes of the zonal-height distributions in this section are either not justified or would require much more detail to adequately discuss. It does not look to me like there is a particularly sharp transition at -40°C, nor would one be expected given the importance of sedimentation in cirrus. The retrieval probably doesn't work well in the antarctic wintertime stratosphere since PSCs are typically mixtures of ice crystals, NAT particles, and ternary aerosols.

AR: We agree with this point, Sec. 6.2 contained some analyses that remained too hypothetical, such as the attribution of some $N_i$ patterns to PSCs. After further investigation, it appears that DARDAR retrievals in these regions are highly uncertain due to potential failures in the cloud mask and wrong categorizations of cloud pixels. This does not mean that DARDAR retrievals are always wrong in these regions but further investigation based on the new version of the DARDAR mask should be performed and are out of the scope of this study. It is now clearly stated in the manuscript that this feature in $N_i$ should not be trusted. Regarding the vertical transition at -40°C, it could still be argued that $N_i^{5\,\mu m}$ and $N_i^{25\,\mu m}$ values quickly change around this temperature, especially in the tropics. $N_i^{25\,\mu m}$ for instance increases from about $50\,L^{-1}$ to above $100\,L^{-1}$. We nevertheless agree that this transition cannot be considered sharp and have toned down this analysis.

Following this comment, Sec. 6.2 has been edited to provide further discussion and remove analyses that seemed too far-fetched. The analysis of seasonal variations of $N_i$ are now also supported by Fig. 8 of the revised manscript, which shows seasonal variability in spatial distributions. It should also be noted that this section also comes as a natural transition between part 1 and part 2, which further investigates some of the observations made in Sec. 6.2. This is now also reminded.

AR: We are grateful for these suggestions and fully agree with them, sections 2 and 3 have been edited accordingly. Sec. 2 now only focuses on describing the methodology and the DARDAR algorithm is described in Sec. 3.1. It can be noted that further technical details on DARDAR are now also provided in Appendix A.

[Figure]

Figure 1: (a) Spatial distribution of the rejection rate associated with the $n_{\text{iter}} < 2$ filtering for pure ice clouds with $T_c < -30$°C. These results correspond to one year of DARDAR retrievals (2008). (b-c) show the corresponding ice water path (IWP) and average number of ice cloud pixels in the vertical column (we recall that the height of a pixel is 60 m). (d) represents the relative difference on $N_i^{5\,\mu\text{m}}$ between -60 and -50°C that would be expected if the $n_{\text{iter}}$ filtering was not applied.)
* * *
2. RC: 2) p 6, l22 it is stated that DARDAR retrievals of pure ice clouds for which the iterative retrieval converged too quickly are ignored. How many of these retrievals are these and can you explain which category of cases these are?

AR: We thank the referee for this comment as we had not yet looked into the distributions of rejection rates associated with the filtering based on iteration number. We agree that useful information could be contained there. It is reminded that this filtering is used to avoid pixels associated with a too quick convergence of the forward model with the observations, which could indicate a lack of information and therefore a strong reliance on a priori considerations. This is now further discussed in Appendix A of the revised manuscript.

The spatial distribution of this rejection rate for ice clouds with $T_c < -30$°C is shown in Fig. 1 of this response. A strong latitudinal dependence of the rejection rate is noted, with less than 10% in the mid-latitudes and about 10 to 20% in the tropics. Rejection rates up to 40% are even seen in the north of oceanic subsidence regions of the South hemisphere. A high rejection rate in DARDAR retrievals in the tropics is not surprising as thick clouds with a complex microphysics are likely to be encountered there. However, Fig. 1(b-c) show that the highest rejection rates occur in regions where thin ice clouds with low IWPs are found, most likely retrieved from lidar-only conditions. It could therefore be that, for these thin clouds, a single iteration is sufficient for proper retrievals and we may be over-constraining the dataset filtering. This has never been investigated from DARDAR and would require further analyses to be verified and fully understood. We have nevertheless verified that this filtering actually has a small impact on the overall climatologies. Fig. 1(d) for instance shows the spatial distribution

[Figure]

Figure 2: Similar to Fig. 4 of the revised manuscript. The PSDs have here been subsetted following the classification proposed for SPARTICUS by Jackson et al. [2015]. PSDs for synoptic and convective clouds are shown on the left and right panels, respectively.

of relative differences in $N_i^{5\,\mu m}$ (in the -60 to -50°C bin) between 1-year climatologies obtained without and with applying the $n_{iter}$ filtering. Differences smaller than 10% are typically found in regions where the rejection rate is the most significant. The bias is positive, which seems to indicate that thin cirrus higher $N_i^{5\,\mu m}$ are ignored because of this filtering. It should be noted when comparing these results to Fig. 7(a-c) of the revised manuscript that relatively very low $N_i^{5\,\mu m}$ values are found in regions where this bias is maximum. The $n_{iter}$ filtering therefore does not have any significant influence on the results shown in this study. After careful consideration, we have chosen to keep the filtering as but we will keep in mind these analyses and results when producing future versions of the dataset (based on the next version of DARDAR cloud and mask products, which should soon be available). It can also be mentioned that all these filtering options will be provided together with the $N_i$ dataset, which will hopefully be distributed co-jointly with the publication of this two-part study.
* * *
3. RC: 3) The evaluation of the prediction of PSD's and Ni (using all field campaigns) and later for retrieved Ni (using coincident SPARTICUS measurements) is shown separately for different temperature intervals, which is important as ice crystal particles shapes differ with temperature. It would be very interesting to separate also anvils and synoptic cirrus, as m-D relations might be different. Is there enough statistics of the collocated SPARTICUS campaign measurements to compare Ni distributions of Fig. 5 for anvils and synoptic cirrus?

AR: We thank the referee for this interesting comment. It is a very good point that m-D relations might be different from different cloud types and this could subsequently affect the quality of our evaluation. As mentioned in Sec. 4.1.1, differentiating between different cloud types has not been attempted in this study for reasons of brevity and also because DARDAR does not make any distinction when assuming its PSD shape and m-D relation. It would nevertheless

be interesting, following the referee's comment, to indeed verify if any specific issue occur when applying a basic differentiation, such as convective vs. synoptic clouds.

To do this, we have associated a cloud type to each PSD from the SPARTICUS dataset used in this study, based on the cloud classifications by Muhlbauer et al. [2014] and Jackson et al. [2015]. Fig. 2 of this response shows comparisons between the histograms of collocated SPARTICUS measurements (Fig. 5 in the original manuscript, Fig. 4 in the revised version) when distinguishing between the "convective" and "synoptic" classification by Jackson et al. [2015]. This classification is chosen here as it is more straightforward. Muhlbauer et al. [2014] offer numerous specific cloud classes, which for this application leads to subsets with a lower statistical significance. It can first be observed in Fig. 2 that convective clouds have higher $N_i$ means, but are also much less occurrent than synoptic clouds during SPARTICUS. With respect to the quality of DARDAR-LIM retrievals no obvious bias or other issue can be noted in either cloud class. Differences are noted but it remains difficult to estimate if these are within the noise, considering the small number of statistics. Testing the impact of m-D relations would also require to disentangle the impact of a possible misrepresentation of the PSD shape in either of these two cloud classes. Finally, it should be kept in mind that these cloud classification are often very difficult to obtain and can be associated with large uncertainties as well.

For these reasons, and to avoid substantial additional descriptions and discussions in the manuscript, we have still kept analyses based on cloud types out of the revised manuscript. But we recognize the importance of this point and the strong interest to differentiate between cloud types to test the impact of the m-D relation but also the assumptions made on the PSD shape. This will be done in a future study that will focus on improving the PSD representation used for lidar-radar $N_i$ retrievals.

———————

4. RC: 4) section 3.2.2: One specific ice crystal mass-maximum diameter (m-D) relationship is used to determine IWC from the PSD. Indeed, Delanoë et al. 2014 show that the uncertainty to the m-D relationship for the normalized PSD is less important when minimizing using lidar extinction and radar reflectivity. The uncertainty seems to increase if only the lidar extinction is used for the minimization (Fig. 9). As both measurements are complementary, there are clouds for which only the first (thin cirrus) or the latter (towards the base of thick cirrus) are available. We also know that the shape of crystals changes with temperature and Heymsfield et al. 2010 showed that the m-D relation for anvil ice clouds yield masses about a factor of 2 larger than for synoptic ice clouds. Erfani and Mitchell 2016 cite this result in their paper and write that their results showing a similarity in m-D expressions between these two cloud types might be an artefact if the ice particle masses for a given projected area are quite different between these types. The L16 m-D relationship was developed for midlatitude cirrus. So for tropical anvils the computed IWC might be biased. Did you test the IWC computed with the L16 m-D relationship with the measured IWC for tropical anvils (using SPARTICUS and ATTREX) ?

AR: We again thank the referee for this very good point. It is absolutely correct that, as shown by Delanoë et al. [2014] and mentioned by the referee here, uncertainties related to the m-D relation used on the normalized PSD are minimum when both lidar and radar are available. This should lead to smaller uncertainties on lidar-radar $N_i$ estimates, as now discussed in the appendix of the revised manuscript.

The consequences on $N_i$ could also be evaluated using the histograms for coincident flights shown in Fig. 2. However, it can be argued that the statistics are for the moment not sufficient

[Figure]

Figure 3: SPARTICUS IWC obtained from Nevzerov (first row) and 2D-S (second row) measurements, as function of L16 predictions based on the 2D-S PSD. The column indicate the cloud category based on Jackson et al. [2015]. A factor of 3 around the one-to-one line is indicated by a dashed line.

to draw any strong conclusions. The impact of m-D assumption will also need to be disentangle from the impact of the PSD shape assumptions, which largely dominate the observed differences. We nevertheless hope to extend this type of evaluation using additional flights coincident with the A-Train, in order to further dig into these issues in the future.

Regarding the use of L16, we have not performed comparison to SPARTICUS or ATTREX measurements in the context of this study, but evaluations of this m-D relation have been made in other studies. Afchine et al. [2018] has for instance shown that this relation should be applicable to tropical clouds, and that the influence of different m-D relations on IWC is small in the temperature range of cirrus. This is now further detailed in Sec. 3.2.2. To provide a more complete response to this comment, we have now analysed the consistency between L16 and SPARTICUS measurements. The classification by Jackson et al. [2015], discussed in the previous response, is used to differentiate between synoptic and convective clouds. IWCs are operationally provided from the 2D-S [based on an assumed area-mass relation; Baker and Lawson, 2006] as well as from bulk measurements from a Nevzerov probe. These comparisons are shown in Fig. 3. It appears that L16 overestimates by a factor of about 2 the IWC measured by the Nevzerov probe. This overestimation seems consistent between synoptic and convective clouds. The 2D-S IWC are in better agreement with L16, for either the synoptic of convective clouds. These results based on SPARTICUS are therefore in agreement with the conclusions by Afchine et al. [2018]. Unfortunately, the ATTREX IWC was not available in the version of the data used for this study and the same analysis could not be repeated. However, it can be noted that Thornberry et al. [2017] showed similarly good agreements between the 2D-S-based IWC and bulk measurements.
* * *
5. RC: 5) Figs. 6 c and d of the case study present the trajectories as function of UTC. The relevant variable is the time difference which you show in brackets, and then the position on the map in Fig. 6a. If it is not too complicated, it might be clearer to present instead of UTC longitude.

AR: We thank the referee for this comment, adding the spatial coordinates would indeed add clarity to compare Fig. 6(c-d) to Fig. 6a (Fig. 5 in the revised manuscript). We have changed these figures so that the time difference is now used as the reference variable and the corresponding lat-lon coordinates for trajectories A and B are indicated in brakets.
* * *
6. RC: 6) concerning Fig. 5, is it possible to get also De from DARDAR for this cloud ?

AR: This is a good point, DARDAR $r_{eff}$ retrievals are now added in Fig. 6(d) of the revised manuscript and are briefly described in Sec. 5.2.
* * *
7. RC: 7) The long descriptive text of the case study is sometimes difficult to follow. I suggest for example to move the analysis of the collocated air track comparison (Fig. 8) to a supplement.

AR: We fully agree with this comment, especially considering that the paper already is long and that thorough in situ analyses have extensively been discussed in the previous section. This figure aimed at comforting these results and show that DARDAR-LIM is capable of reproducing the spatial variability of $N_i$ observed by the 2D-S. Following this comment, it has been moved to supplementary materials (see Fig. S7) and the discussion in Sec. 5.2 has been shortened accordingly.
* * *
8. RC: 8) I would rename section 6 'Presentation of global Ni climatologies' and 6.1 'Geographical distributions'. P21l5: 'considered with caution' instead of 'cautiously considered'

AR: We thank the referee for this comment, Sec. 6 has been edited accordingly.

[revised manuscript text omitted]

Figure S2: The left figure shows relative errors on DARDAR-LIM $N_i$ estimates obtained by propagating the Gaussian standard-deviations on IWC and $N_0^*$ provided in DARDAR operational retrievals (here based on 500 orbits; Jan-Feb 2008). The propagation method is mentioned in Sec. A2. These errors are provided as function of $N_i$ per temperature bins (color lines), minimum diameter threshold bins for the $N_i$ integration (rows; see Sec. 3.3), and instrumental conditions (columns; see Sec. A1). To clearly identify the dominating errors under different temperature and instrumental conditions, the right figure similarly indicates the distribution of $N_i$ values in each panel.

[Figure]

Figure S3: Analysis of the sensitivity of DARDAR-LIM $N_i$ estimates to prior PSD shape assumptions ($\alpha$ and $\beta$ parameters in Eq. (4)). The top figure shows PSD predictions by the D05 parameterization in 3 temperature bins. For each bin, representative $N_0^*$ and $D_m$ values (indicated in the legend) have been selected based on all in situ campaigns described in Sec. 3.2. The D05 PSD is shown in blue and other colors correspond to PSDs computed using $\alpha$ and $\beta$ values extracted by Delanoë et al. [2014, D14] from multiple in situ campaigns (see Table 4 of that study or figure below). Vertical plain, dashed and dotted lines indicate the position of $D_{min} = 5$, 25 and $100 \mu$m, respectively. The bottom figure indicates relative biases $\Delta N_i$ between predictions by D05 ($N_{iD05}$) and the $N_i$ obtained from a wide range of $\alpha$ (x-axis) and $\beta$ (y-axis) values. Brown and blue color therefore indicate an overestimation and underestimation of $N_i$ by D05. Specific $\alpha$ and $\beta$ values from each campaign used in D14 are indicated by various point shapes. D05 is represented by a black dot. $\Delta N_i$ is computed for each selected $D_{min}$ threshold (rows; see Sec. 3.3) and per temperature bins (columns) similarly to the top figures. The $N_{iD05}$ values are indicated in each panel.

[Figure]

Figure S4: Similar to Fig. 2 of the paper, scatterplots of $N_i$ retrieved by DARDAR-LIM as function of the co-incident 2D-S measurements during SPARTICUS. Red, green and blue dots indicate that $N_i$ estimates were obtained in lidar-, radar-only and lidar-radar conditions. White isolines show the overall 68% confidence interval per $D_{min}$ and $T_c$ bins.

[Figure]

Figure S5: Similar to Fig. S4 but for D05 predictions (based on in situ IWC and $N_0^*$) as function of 2D-S measurements.

[Figure]

Figure S6: Similar to Fig. 4 of the paper, but the histograms are separated per instrumental conditions.

[Figure]

Figure S7: $N_i$ measured by the 2D-S (first column) and retrieved by DARDAR-LIM (second column) along a projection of the Learjet-25 track on the A-Train overpass, show in Fig. 5(a-b) and Fig. 6 of the paper. The $N_i$ is provided as function of the aircraft flight time (x-axis), with the distance to the satellite overpass track indicated in brackets. The overpass time (about 19:56 UTC) is shown by a vertical green line. The color background indicates if the DARDAR-LIM $N_i$ has been estimated under lidar-only (black), radar-only (white) or lidar-radar (grey) conditions.

[Figure]

Figure S8: Spatial distribution of the count of $N_i$ retrievals by DARDAR-LIM per temperature bin, corresponding to Fig. 7 of the paper.

[Figure]

Figure S9: Spatial distribution of the count of $N_i$ retrievals by DARDAR-LIM per temperature bin, corresponding to Fig. 9 of the paper.

**References:**

J. Delanoë, A. J. Heymsfield, A. Protat, A. Bansemer, and R. J. Hogan. Normalized particle size distribution for remote sensing application. *J. Geophys. Res*, 119(7):4204–4227, 2014. doi: 10.1002/2013JD020700. URL `http://dx.doi.org/10.1002/2013JD020700`.

---

## Referee Report (RR1)

Review of MS No.: acp-2018-20
Title: Ice crystal number concentration estimates from lidar-radar satellite remote sensing. Part 1: Method and evaluation
Author(s): Odran Sourdeval, E. Gryspeerdt, M. Krämer, T. Goren, J. Delanoë, A. Afchine, F. Hemmer, and J. Quaas
MS Type: Research article

General Comments:
The authors have provided thorough replies and explanations to all my comments/questions. I appreciate all the work they have invested into this process and which has provided the research community with a valuable tool. I have only a few specific comments.

Specific Comments:

1.  Regarding the 2[nd] author comment and Fig. S3, please include SPARTICUS data in the analysis for Fig. S3. SPARTICUS (dedicated to sampling cirrus clouds) was conducted over a 6-month period with several flights per month. Some flights were downwind of the Rocky Mountains of North America while many others were over the ARM Southern Great Plains facility. This should provide a diversity of cirrus conditions and should thus provide representative sampling over continental regions. The other campaigns listed do not provide the long-term and spatially extensive sampling that SPARTICUS provides. I'm assuming that the ARM data indicated is for the year 2000 ARM campaign.

2.  Regarding major comment/response #5 and Sec. 6.1, the cirrus sampled by the lidar-IR method in Mitchell et al. (2016; 2018) roughly correspond to visible optical depths (ODs) between 0.3 and 3.0. Most would probably agree that cirrus clouds that completely block the Sun's image are not thin cirrus, and one can just make out the orb of the Sun at a cirrus OD of 1.0 (Peter Francis, private communication). That is, the Sun's orb is not visible for OD > 1.0 approximately. Therefore, it is suggested that in Sec. 6.1, that the sentence appearing as "Lidar and thermal-infrared measurements indeed only provide the concentrations of thin cirrus or at cloud-top found at this temperature range, ..." be changed to "Lidar and thermal-infrared measurements indeed only provide the concentrations of **thin-to-moderately thick** cirrus or **near** cloud-top found at this temperature range, …". This revised sentence appears consistent with the findings of Hong and Liu (2015, J. Climate).

This reviewer agrees that OD differences in the tropical cirrus clouds sampled by the lidar-IR method vs. the DARDAR-LIM method is the most logical explanation for the relative differences in tropical N retrieved by these two methods. The fact that the two retrievals exhibit similar relative differences in N in the mid-to-high latitudes is indeed encouraging.